# Extending the planar theory of anyons to quantum wire networks

T. Maciazek[1], A. Conlon[2,3,*], G. Vercleyen[2], J. K. Slingerland[2,3]

**1** Department of Mathematics, University of Bristol, England
**2** Department of Theoretical Physics, Maynooth University, Ireland
**3** Dublin Institute for Advanced Studies, School of Theoretical Physics, 10 Burlington Rd, Dublin, Ireland
* aaron@stp.dias.ie

April 11, 2023

## Abstract

The braiding of the worldlines of particles restricted to move on a network (graph) is governed by the graph braid group, which can be strikingly different from the standard braid group known from two-dimensional physics. It has been recently shown that imposing the compatibility of graph braiding with anyon fusion for anyons exchanging at a single wire junction leads to new types of anyon models with the braiding exchange operators stemming from solutions of certain generalised hexagon equations. In this work, we establish these graph-braided anyon fusion models for general wire networks. We show that the character of braiding strongly depends on the graph-theoretic connectivity of the given network. In particular, we prove that triconnected networks yield the same braiding exchange operators as the planar anyon models. In contrast, modular biconnected networks support independent braiding exchange operators in different modules. Consequently, such modular networks may lead to more efficient topological quantum computer circuits. Finally, we conjecture that the graph-braided anyon fusion models will possess the (generalised) coherence property where certain polygon equations determine the braiding exchange operators for an arbitrary number of anyons. We also extensively study solutions to these polygon equations for chosen low-rank fusion rings, including the Ising theory, quantum double of groups, and Tambara-Yamagami models. We find numerous solutions that do not appear in the planar theory of anyons.

# 1   Introduction

A topological quantum computer performs its computations using anyons, quantum quasi-particles that obey exotic types of quantum statistics which make such a computer intrinsically robust against errors arising from decoherence [55, 40, 28]. Crucially, performing computations on a topological quantum computer requires the ability to move the anyons around and exchange them. This is a great technological challenge which is currently being addressed by considering architectures for quantum computers that have the structure of a network, where anyons are moved along the edges of the network and exchanged at the junctions [38, 5, 37]. Of particular importance in this context is Kitaev's superconducting quantum wire model that supports Majorana edge modes [38]. Such a system can be realised in semiconductor nanowires coupled to a superconductor [5, 30], as well as other solid state [57, 47, 20, 52, 53], and photonic systems [74]. However, there also exist numerous proposals for realising other types of anyonic excitations on networks. This includes parafermionic excitations [27, 4, 75, 37, 16, 42, 41], and Fibonacci excitations [46], although topological protection may be a problem [14]. Such network-based proposals have been recognised as some of the most robust candidates for an architecture of a topological quantum computer. However, anyon braiding, a crucial ingredient, is still in early development. There have been proposals for braiding Majorana modes [5, 17, 30], accompanied by studies of the resulting errors and qubit fidelity e.g. [10, 65]. There has also been work addressing the scalability of network-based topological quantum computers [36, 37]. Finally, we mention recent experimental evidence of Majorana measurement [3].

Notably, the above mentioned substantial body of literature also shows that, in contrast to anyon theory in two dimensions (2D), there is no uniform theory describing anyons on quantum wire networks. In other words, for every type of anyons which is known from 2D physics (Ising/Majorana, Fibonacci, etc.) a new microscopic model for a quantum wire has to be proposed and the existence of well-defined topological anyonic exchange operators has to be proved. In this work, we aim to establish a universal anyon theory for quantum wire networks which is analogous to the braided fusion theory of anyons in 2D [55, 39]. Braiding describes all the possible ways the anyons can be exchanged (up to continuous deformations of anyons' worldlines). This information is encoded in a mathematical object called the fundamental group of the appropriate configuration space, also called the braid group (see the seminal work [45] for more details). For 2D anyon theory the relevant mathematical object is Artin's well-known braid group [8]. However, according to the general theory laid out in [45], *Artin's braid group does not describe anyon exchange on a network*. The correct mathematical object describing anyon braiding on a network (graph) is its graph braid group [9, 2]. This crucial observation has led to the necessity of developing new physical and mathematical models for anyons constrained to exchange on a network [9, 34, 32].

Intuitively, a full description of braiding of many anyons on arbitrary networks can be considerably complicated and requires using advanced mathematical tools [19, 43, 44]. However, recently a tractable and physically intuitive description of graph braid groups has been accomplished [6, 48]. Importantly, it shows that graph braids have strikingly different properties than planar braids. One such property is the lack of the standard Yang-Baxter relation between the graph braids. Moreover, a fixed pair of anyons can typically be exchanged in several topologically independent ways on a given graph. Since graph braid groups govern the anyon exchange on networks, this raises the importance of understanding what exchange statistics are possible. This question can be answered only when one takes into account anyon fusion, i.e. processes where anyons are not only braided with each other, but also where a group of anyons behaves as one composite anyon. Importantly, braiding and fusion of anyons must be compatible with each other, a fact which for planar anyon theories is guaranteed by the hexagon equations [55, 39]. Only recently, the compatibility of braiding and fusion on a single junction (i.e. a network that consists of multiple edges incident to a single vertex) has been considered in [18], by two of the authors of this paper. The results show numerous important differences between 2D anyon models and anyon models defined on a network. In particular, the planar hexagon equations are replaced by the more general $P$- and $Q$-hexagon equations that lead to Abelian and non-Abelian quantum exchange statistics which do not appear in the planar theory. In this work, we follow the programme set out in [18], and work towards a complete anyon fusion theory where anyon fusion and braiding are compatible on arbitrary networks (composed of multiple junctions and also containing loops) and for arbitrary numbers of anyons. These compatibility conditions are encoded in a finite set of certain polygon equations. By solving the polygon equations, we show numerous possibilities for the existence of quantum exchange statistics which are not present in 2D anyon theories. Besides emphasising the fundamental importance of this fact, we also show that the new possibilities can be utilised in topological quantum computers to build more efficient quantum circuits.

We aim for the presented material to be self-contained. Thus, we include an introduction to the relevant notions from the planar anyon fusion theory in Section 2 as well as a brief recap of the key features of the graph braid groups in Appendix A. Because our work builds on the results of [18] concerning the trijunction, we review the main points of this work at the beginning of Section 3. In Section 3.1 we take the first steps towards generalising the results of [18] and build anyon models for higher numbers of particles on a trijunction. This is subsequently generalised to anyons constrained to move on tree graphs in Section 3.3. Our general methodology is to build anyon models for certain small canonical graphs which are the building blocks of larger networks. Consequently, in Sections 4 and 5 we study anyon models on a circle and on a lollipop graph respectively. Section 6 contains a discussion about the solutions to the graph-braid equations for several fusion rings. In Section 7 we study anyon models on a $\Theta$-graph in order to show that the exchange operators in our anyon models on any triconnected network are identical to the exchange operators from the corresponding planar anyon model. In other words, sufficiently highly connected networks can host only planar exchange statistics. Thus, the new exchange statistics may appear within our presented framework only on one-connected (also known as separable) networks (e.g. star graphs or trees) or biconnected networks, a possibility which may be useful for generating larger sets of topological quantum gates (Section 8). We also extensively study solutions to our polygon equations (encoding the compatibility of fusion and braiding on a given network) for chosen low-rank anyon models such as the Ising model (Appendix G), Tabmara-Yamagami models (Appendices F and I) and the quantum double of $\mathbb{Z}_2$ (Appendix H). Some key general features of the solutions are also collected in tables which are distributed throughout the main body of the paper. Finally, we conjecture that our anyon models will possess a generalised coherence property. Our reasoning is outlined in Appendix C.

## 2  Planar braiding of anyons

In this section, we will provide a brief overview of planar braiding of anyons. For further detail, we refer the reader to [39, 51], as well as recent papers such as [12]. By an anyon model we mean the following data; a fusion algebra, labelling the topological charges and their fusion rules, the $F$- symbols, giving consistent recoupling rules, and the $R-$ symbols, which give the exchange statistics of the anyons in the model. We shall review each of these in order.

A fusion algebra consists of a finite set of particles, labelled by their topological charge with fusion rules written as,

$$a \times b = \sum_c N_c^{ab} c. \tag{1}$$

The coefficients $N_c^{ab} \in \mathbb{Z}_{\geq 0}$ are the dimension of the fusion space $V_c^{ab}$ of ground states with two particles of charges $a$ and $b$ and with overall charge $c$. There is a unique anyon, 1 called the vacuum, such that $a \times 1 = 1 \times a = a$. Each anyon has a unique antiparticle such that, $a \times \bar{a} = 1 + \dots$. Anyon $a$ is called abelian if there is only the vacuum charge on the right-hand side, i.e. $a \times \bar{a} = 1$. In this paper we will focus on multiplicity-free fusion algebras which means the coefficients $N_c^{ab}$ are either 0 or 1 and consider only *commutative* product which means that $a \times b = b \times a$. Considering non-commutative products would be a natural extension of the work presented in this paper. Intuitively, it would be a very suitable extension, as placing anyons on an edge of a network naturally imposes their linear ordering.

We choose an orthonormal basis for each nontrivial fusion space $V_c^{ab}$. This choice introduces a gauge freedom $u_c^{ab}$, a unitary matrix of dimension $N_c^{ab}$. In the multiplicity free case, $u_c^{ab} \in U(1)$. The two isomorphic ways to fuse three anyons to get a total topological charge $d$ are related by a change of basis given by the matrix elements of the $F$-symbols,

$$\left[F_d^{abc}\right] : \bigoplus_e V_e^{ab} \otimes V_d^{ec} \rightarrow \bigoplus_f V_d^{af} \otimes V_f^{bc}. \tag{2}$$

The action of the $F$-symbols are graphically represented as,

The $F$-symbols are required to satisfy the *pentagon equations*,

$$\left[F_e^{fcd}\right]_{gl}\left[F_e^{abl}\right]_{fk} = \sum_h \left[F_g^{abc}\right]_{fh}\left[F_e^{ahd}\right]_{gk}\left[F_k^{bcd}\right]_{hl}. \tag{3}$$

A solution of the pentagon equations gives a set of $F$ symbols which can be used to rearrange the compositional order of fusion locally [49, 23].

The other important ingredient for anyon models is the exchange statistics and the resulting braiding operators. The exchange statistics in planar anyon models are governed by the $R$- symbols which, for multiplicity-free fusion rules, are $U(1)$ matrices acting on the fusion space;

$$R_c^{ba} : V_c^{ba} \rightarrow V_c^{ab}. \tag{4}$$

The action of the $R$-symbols is graphically represented as

This action allows one to resolve a simple braid in spacetime diagrams by introducing an $R$-symbol acting on the states in the fusion space. We will frequently use graphical depictions later in this work when we discuss graph braiding of anyons. A change of basis of $V_c^{ab}$ introduces a gauge transformation of the $R$- symbols and $F$-symbols, which is discussed in Section E. The compatibility of fusion and braiding is implemented by enforcing that we can slide a fusion vertex through a braid in spacetime history;

This is implemented by the *hexagon equations* which come from hexagonal commutative diagrams [39, 23, 63, 58]. There are four hexagon equations corresponding to four topologically inequivalent ways that fusion can commute with braiding. However, only two of them are independent. Here we show one of them:

The other independent diagram is obtained when the worldlines of anyons $a$ and $b$ braid over the worldline of anyon $c$. The resulting hexagon equations read as follows;

$$R_g^{ca} \left[F_d^{acb}\right]_{gf} R_f^{cb} = \sum_e \left[F_d^{cab}\right]_{ge} R_d^{ce} \left[F_d^{abc}\right]_{ef},$$

$$R_g^{ca} \left[(F_d^{bac})^{-1}\right]_{ge} R_e^{ba} = \sum_f \left[(F_d^{bca})^{-1}\right]_{gf} R_d^{fa} \left[(F_d^{abc})^{-1}\right]_{fe}.$$

(5)

Satisfying the above consistency relations describing the compatibility of fusion and braiding of $N = 3$ anyons implies the compatibility of fusion and braiding for any number of anyons, a result known as the *braided coherence theorem* [60]. Furthermore, there are only a finite number of solutions to the planar hexagon equations up to gauge equivalence. This property is known as Oceanu rigidity [39, 24]. One particular gauge invariant quantity we will discuss on the circle graph in Section 4 is the topological twist. In the planar case, this is represented by the following spacetime diagram;

The twist factors $\theta_a$ can be expressed in terms of the $R$-symbols as,

$$\theta_a = \theta_{\bar{a}} = \sum_c \frac{d_c}{d_a} R_c^{aa},$$

(6)

where $d_a$ is the quantum dimension of anyon $a$. By Vafa's theorem [72], the twist factors are constrained to be roots of unity. The twist factors are related to changing an anyon's so-called "framing" [61]. Another relation between the twist factors and the $R$-symbols is the ribbon property,

$$R_c^{ab} R_c^{ba} = \frac{\theta_c}{\theta_a \theta_b}.$$

(7)

The ribbon property comes from considering the worldlines of anyons as world-ribbons which get twisted when an anyon's worldline is wrapped around itself. In other words, the twist factors represent full twists of the world-ribbons. Similarly to the full twists, one can also consider half-twists (also called the $\pi$-twists). Interestingly, the full and the half-twists come up naturally in the graph setting (see Appendix D), and are necessary for proofs of our results in the later sections.

Recall that any planar braid is a composition of simple braids exchanging pairs of neighbouring anyons. Thus, using the $R$-symbols which satisfy the hexagon equations, one can construct a representation of the planar braid group. In particular, for $N = 3$ identical anyons of topological charge $a$ and the total charge of the system $c$, we get the following representation of $B_3 \to U(d)$, [63, 22, 54],

$$\rho(\sigma_1) = \text{diag}\left(R^{aa}_{b_1}, \ldots, R^{aa}_{b_k}\right), \quad \rho(\sigma_2) = \left(F^{aaa}_c\right)^{-1} \rho(\sigma_1) F^{aaa}_c, \tag{8}$$

where are the $b_k$ fusion outcomes of $a \times a = b_1 + \cdots + b_k$. The $k \times k$ unitary matrices $\rho(\sigma_1)$ and $\rho(\sigma_2)$ are called the braiding exchange operators. Crucially, the braiding exchange operators satisfy the *Yang-Baxter* relation, i.e.

$$\rho(\sigma_1)\rho(\sigma_2)\rho(\sigma_1) = \rho(\sigma_2)\rho(\sigma_1)\rho(\sigma_2).$$

In other words, the braiding exchange operators form a representation of Artin's braid group [8]. In the quantum computing context, the braiding exchange operators have the interpretation of topological quantum gates acting on a single qudit. In Section 8 we consider similar topological quantum gates coming from graph-braided anyon models. We also address computational universality and the circuit depth of a graph-based topological quantum computer. In particular, we show that the graph-based architecture may allow one to build quantum circuits of a lower depth.

To end this review section, we will collect results for a few concrete anyon models we intend to reference later in the paper. For any finite Abelian group, $H$, one can construct a fusion algebra where the group multiplication gives the fusion rules. There is always guaranteed at least one solution to the pentagon equation given by the trivial $F$- symbols. However, often more solutions exist. One interesting family of models is provided by $H = \mathbb{Z}_N$. The anyons are labelled by $[a]_N$, the least reside of $a$ modulo $N$ and the $F$-symbols are given by $U(1)$ valued three-cocycles in the group cohomology of $H$ [13]. Here the family splits into two cases depending on whether $N$ is even or odd. The situation becomes interesting if one tries to introduce a non-trivial cocycle for the $F$- symbols. Then, there is only a solution to the hexagon equations if $N$ is even [51, 23, 29].

Another family of interest is the Tambara-Yamagami models (TY($G$)) [70], which are constructed over a finite Abelian group, $G$ with the addition of a non-Abelian anyon; $\sigma$. The fusion rules are given by

$$\sigma \times \sigma = \sum_i g_i, \qquad g_i \times \sigma = \sigma \times g_i = \sigma, \qquad g_i \times g_j = g_i g_j. \tag{9}$$

The corresponding non-trivial $F$-symbols are,

$$\left[F^{g_i \sigma g_j}_\sigma\right]_{\sigma\sigma} = \left[F^{\sigma g_i \sigma}_{g_j}\right]_{\sigma\sigma} = \chi(g_i, g_j),$$
$$\left[F^{\sigma\sigma\sigma}_\sigma\right]_{g_i g_j} = \kappa\,\tau^{-1}\overline{\chi}(g_i, g_j), \tag{10}$$

where $\chi$ is a symmetric non degenerate bicharacter, $\kappa$ is the Frobenius Schur indicator and $\tau = |G|^{-1/2}$. It has been proven that unless $G$ is a direct product of $\mathbb{Z}_2$ factors, there are no solution to the hexagon equations, [67]. For graph braided particles with Tambara-Yamagami fusion rules this result remains the same for graphs with junctions. We provide a proof of this result in Appendix F. For the circle graph, however, there are solutions to the graph-braid hexagon equations. Specific solutions can be found in Appendix I. In the case $G = \mathbb{Z}_2$, the anyons are usually denoted $(1, \psi, \sigma)$, where $\psi$ is the non trivial element of $\mathbb{Z}_2$. There are two solutions to the pentagon equations, which are related by the choice of the Frobenius-Schur indicator $\kappa = \pm 1$, [39, 58]. The solutions to the hexagon equations are,

$$R^{\psi\psi}_1 = -1, \qquad R^{\sigma\psi}_\sigma = R^{\psi\sigma}_\sigma = \pm i, \qquad R^{\sigma\sigma}_1 = e^{\pm i(2k+1)\pi/8}, \qquad R^{\sigma\sigma}_\psi = \pm i R^{\sigma\sigma}_1, \tag{11}$$

where $k \in \{0, 1, 2, 3\}$. The particular values of $k = 0$ and $k = 3$ are for the choice of $\kappa = 1$ and the other values of $k$ are for the choice $\kappa = -1$. The choice of $\kappa = +1$ is often called the Ising model, [39]. The topological twists for the Ising solutions to the hexagon equations are,

$$\theta_\psi = -1, \qquad \theta_\sigma = e^{\frac{i\kappa\pi}{8}}. \tag{12}$$

There are many other notable anyon models such as; the Fibonacci anyon model, [71], quantum groups with a truncated tensor product [7, 11], Rep($G$), the category of representations of $G$ any finite group [23, 25], and a quantum double of a finite-dimensional semisimple Hopf algebra [40]. Recently there have been experimental measurements of the exchange statistics for the case $D(\mathbb{Z}_2)$ [64, 69, 76].

# 3   Anyon models on star graphs and tree graphs

Let us briefly recall the fundamental differences between graph braided anyon models and the planar braided anyon models. Following the formalism introduced in [18], we associate $V_c^{ab}$ with the fusion space of anyon states, where fusion takes place on the edges of the graph $\Gamma$. The $n$-strand braid group [2, 6] of the graph $\Gamma$ will be denoted by $B_n(\Gamma)$.

   The general strategy is to assign different braiding exchange operators (acting on the states of the fusion space) to different (i.e. topologically inequivalent) elements of the graph braid group. In particular, on a tri-junction, we can represent the exchange of the two particles closest to the junction by a $U(1)$ matrix, analogous to planar anyon models. The braiding exchange operators corresponding to the exchange of the two particles closest to the junction are associated with $R$-symbols as shown in Figure 1. The corresponding simple braid is denoted $\sigma_1^{(1,2)}$, where 1 and 2 refer to the edge assignment as can be seen in Figure 1. The particle closest to the junction is sent to edge with label (1), which we identify with the back plane and the second particle is sent to edge with label (2), which we identify with the front plane. Consequently, $\sigma_1^{(2,1)}$ is the inverse of the braid $\sigma_1^{(1,2)}$. We can view the action of a $\sigma_1$ graph braid as a spacetime process where particles initially placed on an edge of the graph are transported through a junction point to other edges and then returned to the initial edge in a different order. From now on, all the $R$-symbols will be associated with such a $\sigma_1^{(1,2)}$ graph braid and should not be confused with the $R$-symbols used in Section 2 in the context of 2D anyon models. In particular, they may not solve the heaxgon equations (5).

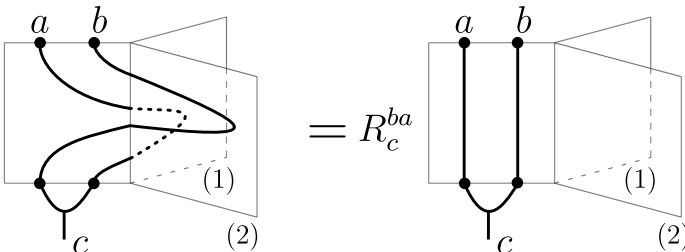

Figure 1: The simple braid $\sigma_1^{(1,2)}$ and the associated $R$-symbol. The superscript in $\sigma_1^{(1,2)}$ refers to the edge assignment the particles are sent to under the graph braid.

   To incorporate the commutation of fusion and braiding processes, we need to consider at least three anyons. Here we find the first clear differences between braiding in the plane and braiding on a graph. Firstly, there are two topologically inequivalent ways of realising the simple braid $\sigma_2$ on a trijunction [48, 6]. Namely, the two realisations are distinguished by the edge visited by the anyon closest to the junction. These simple braids are denoted by $\sigma_2^{(1,1,2)}$ and $\sigma_2^{(2,1,2)}$ (see Figure 2 and Appendix A for more explanation). Despite these differences, it is possible to construct a graph anyon model on a trijunction which reflects the properties of the respective graph braid group in the sense that different unitary operators represent inequivalent simple braids on the Hilbert space, [18]. The key idea relies on introducing $P$-symbols and $Q$-symbols associated with the simple braids $\sigma_2^{(1,1,2)}$ and $\sigma_2^{(2,1,2)}$ respectively, We display the action of these in Figure 2.

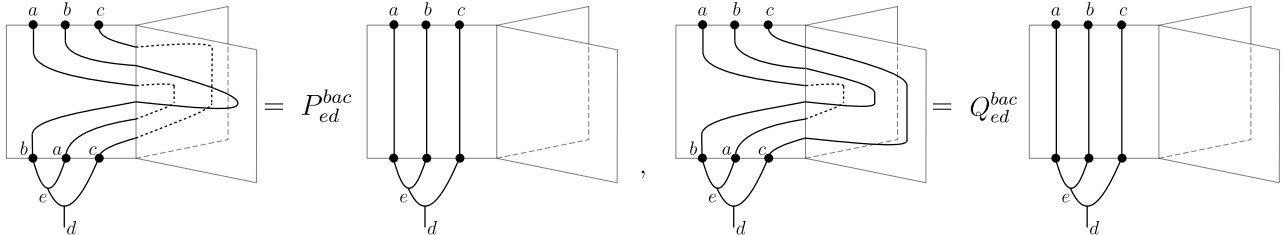

Figure 2: The $P$- and $Q$-symbols associated with the simple braids $\sigma_2^{(1,1,2)}$ and $\sigma_2^{(2,1,2)}$ on a trijunction.

The gauge transformations of the $R$-, $P$- and $Q$- graph braid symbols have the same structure as the gauge transformations of the planar $R$-symbols, since the topological charge $e$ is conserved. We discuss the gauge equivalence of solutions in Appendix E. However, so defined simple braids do not satisfy the Yang-Baxter relation, i.e. the composite braid $\sigma_1^{(1,2)}\sigma_2^{(1,1,2)}\sigma_1^{(1,2)}$ is topologically inequivalent to the composite braid $\sigma_2^{(1,1,2)}\sigma_1^{(1,2)}\sigma_2^{(1,1,2)}$. In fact, the three-strand braid group of a trijunction is a free group generated by the above-defined three simple braids [6]. In other words, the corresponding braiding exchange operators determine some particular unitary representations of the graph braid group.

Next, let us revisit the derivation of the generalised hexagons containing the $P$- and $Q$-symbols as it contains ideas which are key for the remaining parts of this paper. We will study only the $Q$-hexagon in detail. The derivation of the $P$-hexagon is completely analogous and has been done in detail in [18]. The key idea is to incorporate the commutation of fusion and graph braiding of anyons into the spacetime histories. This is done by considering compositions of the simple braids where the spacetime configuration of worldlines of two anyons is such that the two worldlines stay next to each other throughout the process, and their fusion vertex can be pulled through the entire exchange. An example of such a braid is $\sigma_1^{(1,2)}\sigma_2^{(2,1,2)}$ whose relevant deformations are shown in Figure 3.

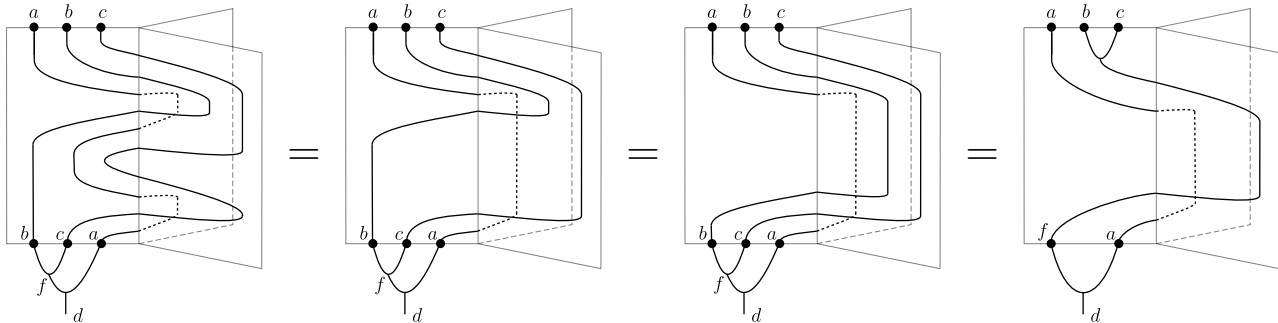

Figure 3: The fusion vertex of anyons $b$ and $c$ can be pulled through the entire braid $\sigma_1^{(1_a,2_c)}\sigma_2^{(2_c,1_a,2_b)}$ so that the resulting process is just a simple braid of the composite anyon $f = b \times c$ with anyon $a$, i.e. $\sigma_1^{(1_a,2_{b\times c})}$. The diagram on the furthest right expresses the $\sigma_1^{(1_a,2_{b\times c})}$ graph braid.

We can observe that the diagrams on the far left and far right of Figure 3 can be related by sequences of $F$ symbols and resolving the graph braids analogous to the derivation of the planar hexagon equations. This leads to the $Q$-hexagon diagram shown in Figure 4.

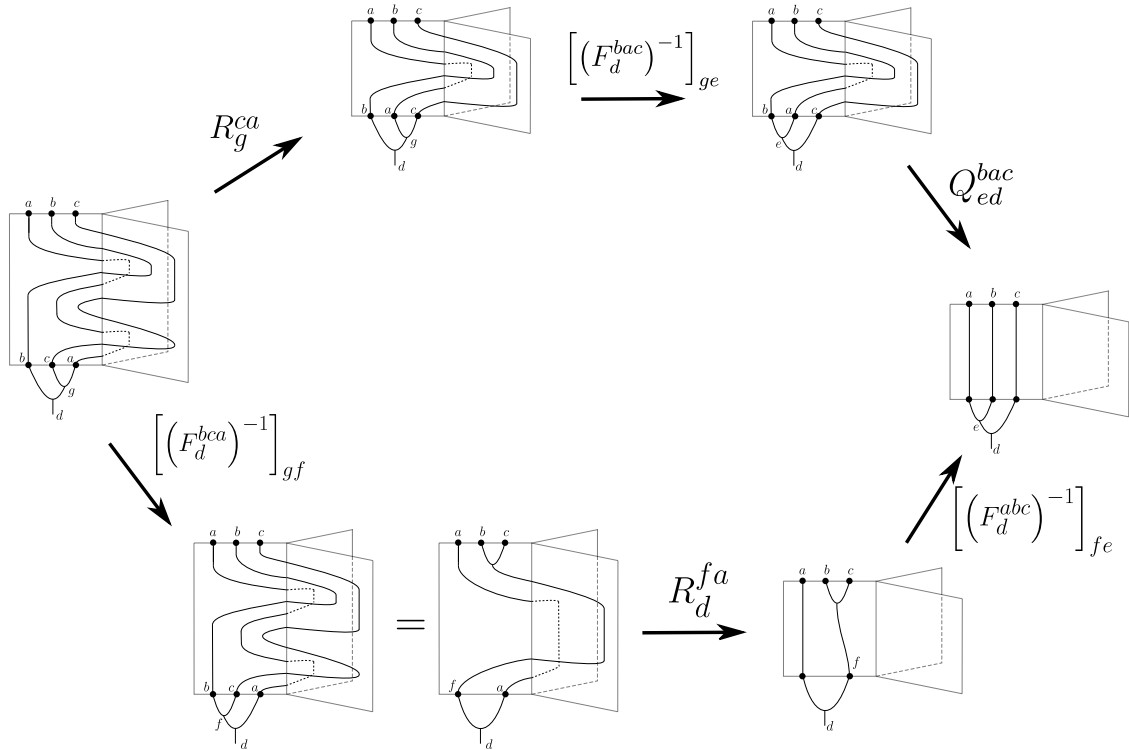

Figure 4: The hexagon diagram which we call the $Q$-hexagon. It is derived from the identity $\sigma_1^{(1_a,2_c)}\sigma_2^{(2_c,1_a,2_b)} = \sigma_1^{(1_a,2_{b\times c})}$ shown in Figure 3 and applied in the bottom-left corner of the hexagon. The hexagon diagram provides a set of hexagon Equations (13) which allow one to express the $Q$-symbols via the $R$- and $F$-symbols.

Equating the upper and lower path in Figure 4 leads to the $Q$-hexagon equations,

$$R_g^{ca}\left[\left(F_d^{bac}\right)^{-1}\right]_{ge} Q_{ed}^{bac} = \sum_f \left[\left(F_d^{bca}\right)^{-1}\right]_{gf} R_d^{fa}\left[\left(F_d^{abc}\right)^{-1}\right]_{fe}. \tag{13}$$

There are three more hexagon diagrams coming from the braids $\sigma_2^{(1,1,2)}\sigma_1^{(1,2)}$, $\sigma_2^{(2,2,1)}\sigma_1^{(2,1)}$ and $\sigma_1^{(2,1)}\sigma_2^{(1,2,1)}$, but only two of the four lead to independent hexagon equations. The other independent set of hexagon equations is the following,

$$P_{gd}^{cab}\left[F_d^{acb}\right]_{gf} R_f^{cb} = \sum_e \left[F_d^{cab}\right]_{ge} R_d^{ce}\left[F_d^{abc}\right]_{ef}. \tag{14}$$

In [18], it is shown that solving the graph hexagon equations for $N = 3$ particles with $\mathrm{TY}(\mathbb{Z}_2)$ (also known as Ising) fusion rules and $F$-symbols on a trijunction leads to a two parameter family of solutions for the $R$-symbols;

$$R_1^{\sigma\sigma} = \pm i R_\psi^{\sigma\sigma}, \qquad R_\sigma^{\sigma\psi} = R_\sigma^{\psi\sigma} = \pm i, \qquad R_1^{\psi\psi}, R_1^{\sigma\sigma} \in U(1). \tag{15}$$

The only constraints on $R_1^{\psi\psi}$ and $R_1^{\sigma\sigma}$ is that they are elements of $U(1)$. As we explain in Appendix G, further consistency equations for $N = 4$ Ising anyons on a trijunction will fix $R_1^{\psi\psi} = -1$ and only $R_1^{\sigma\sigma}$ will remain the free parameter of the theory. The corresponding expressions for $P$ and $Q$ symbols are contained in Section 2 of the Supplementary Material of [18]. Analogous to the planar braiding of anyons, there is no solution to the graph braiding hexagon equations for $\mathrm{TY}(G)$ on a trijunction, unless $G = \mathbb{Z}_2$ to some power, we provide a proof of this in Section F.

Although we have focused on a trijunction, the analysis generalises to junctions of arbitrary order. See, for instance, the Supplementary Material of [18] where the tetrajunction is studied. Although increasing the

valence introduces additional topologically inequivalent ways to exchange particles at the junction, in particular, a valence $d$ star graph will have $(d-1)(d-2)/2$ inequivalent $\sigma_1$ generators and $(d-1)^2(d-2)/2$ inequivalent $\sigma_2$ generators. The introduction of fusion commuting with braiding effectively "splits" the star graph into a collection of trijunctions. On each trijunction, one has two independent sets of hexagon equations, while on a valence $d$ star graph, one has $(d-1)(d-2)$ independent hexagon equations. However, there are no consistency relations mixing exchanges on different trijunctions (see [18] for more explanation).

## 3.1 Greater particle number

In this section, we will discuss graph braided anyon models with four or more particles. For planar braided anyon models, this situation is covered by MacLane coherence theorem [49] and the braided coherence theorem, [60]. The implication of these theorems is that the solutions of the pentagon and hexagon equations are sufficient for the description of any number of anyons. Explicitly, if one constructed some braiding polygon for $N > 3$ particles, one could use the pentagon and hexagon equations iteratively to satisfy this polygon and find no new constraint equations on the $R$ and $F$ symbols of the theory. However on a graph, since there are multiple topologically inequivalent choices for $\sigma_j$ with $j > 1$ (as we discussed earlier), satisfying the 3-particle $P$- and $Q$-hexagons does not guarantee that we have a full description for any number of particles. In this section we will focus on a trijunction, the simplest graph permitting particle exchange and discuss later how the analysis translates to higher valence graphs.

The new generators of the graph braid group for $N = 4$ (see Appendix A for an exhaustive definition of the generators) and their corresponding symbols are

$$\rho(\sigma_3^{(1,1,1,2)}) = X, \quad \rho(\sigma_3^{(2,2,1,2)}) = Y, \quad \rho(\sigma_3^{(2,1,1,2)}) = B, \quad \rho(\sigma_3^{(1,2,1,2)}) = A. \tag{16}$$

The gauge transformation of the four particle graph braid symbols is given in Appendix E where we discuss removing the gauge symmetry from the obtained solutions. There, we also list our convention for the four particle anyon labels in Equation (59). We will use this convention in the present section.

The first step is to resolve how the $\sigma_3$-graph braids from (16) act in the fusion space of four anyons $V_e^{abcd}$. For the $\sigma_2$-braids represented on a three-particle fusion space this is unambiguous – the two particles being exchanged are joined by a fusion vertex (as we can see in Figure 2). Thus, the respective braiding exchange operators are necessarily diagonal in the left-fused basis where the second and third particle away from the junction point are joined by a common fusion channel. However, for the $\sigma_3$-braids acting on the fusion space of four anyons, the choice of the appropriate fusion tree is not clear at the first sight. Clearly, the two particles being exchanged must be joined by a common fusion vertex. This leaves two choices for the fusion tree structure of the other two particles – the fully left associated (left-fused) basis or the pairwise associated basis. Crucially, there are important physical arguments that dictate the correct choice of the fusion tree. Namely, if a braiding exchange operator is diagonal in a certain basis, then all the anyon charges appearing in the chosen fusion tree have to be conserved throughout the corresponding braiding exchange process. The total charge of a set of anyons is conserved if one can bound this set of anyons by a disk which remains sufficiently separated from the anyons outside the disk throughout the entire exchange process. These disks are associated with the choice of the fusion tree. For instance, the fusion tree with anyons $a, b, c, d$ being fused pairwise implies two separate disks containing the pairs $a, b$ and $c, d$ respectively and one disk containing all the anyons $a, b, c, d$ (note that the disks cannot leave the graph as this is the actual space where the anyons move) – see Figure 5a. Importantly, the pairwise-associated fusion tree is not a correct basis for representing the braid $\sigma_3^{(1_d, 2_c, 1_b, 2_a)}$ diagonally as anyons $b$ and $a$ will necessarily enter the disk containing anyons $d$ and $c$ during the exchange, hence the total charge of $c$ and $d$ may not be conserved. This is shown in Figure 5b.

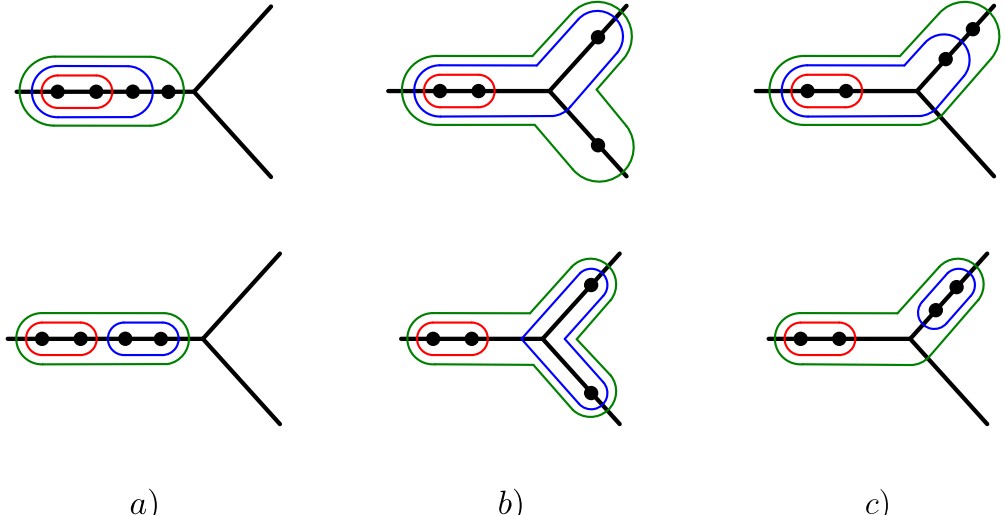

<p align="center"><i>a)</i>                <i>b)</i>                <i>c)</i></p>

Figure 5: a) The disks associated with the left-fused basis (top panel) and the pairwise-fused basis (bottom panel). b) The configuration of particles during the $\sigma_3^{(2,1,1,2)}$-exchange right before the particles bounded by the red disk exchange. The disks in the left-fused basis remain separated during the exchange (top panel) whereas in the pairwise-fused basis (bottom panel) the red disk has to necessarily intersect the blue disk because the blue disk is stretched across the junction. c) A configuration of particles during the $\sigma_3^{(1,1,1,2)}$-exchange right before the particles bounded by the red disk exchange. The disks in both bases remain separated throughout the entire exchange.

Figure 5 also explains that the left-fused basis is a good basis for representing diagonally any $\sigma_3$-braid. However, the braids $\sigma_3^{(1_d,1_c,1_b,2_a)}$ and $\sigma_3^{(2_d,2_c,1_b,2_a)}$ must be represented diagonally both in the left-fused basis and the pairwise-fused basis. This is because anyons $d$ and $c$ visit the same edge during the exchange and thus can also be bounded by a well-separated disk (see Figure 5c). The braid $\sigma_3^{(1,1,1,2)}$ is represented in the left-fused basis by the $X$-symbols as shown in Figure 6.

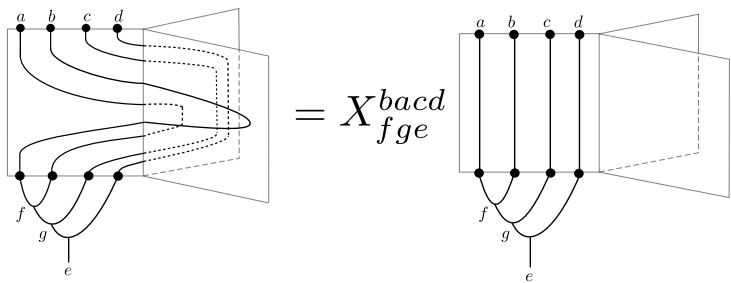

Figure 6: Here we display the symbol $X_{fge}^{bacd}$ resolving the graph braid $\sigma_3^{(1,1,1,2)}$.

The $\sigma_3$ generators can of course be expressed in the pairwise associated basis, given by conjugation by the appropriate $F$-symbols;

$$\left[\tilde{W}_{fe}^{bacd}\right]_{l,l'} = \sum_g \left[(F_e^{fcd})^{-1}\right]_{lg} W_{fge}^{bacd} \left[F_e^{fcd}\right]_{gl'}, \qquad W \in \{X,Y,A,B\}, \qquad (17)$$

where $f$ is the total charge of $a$ and $b$, $g$ is the total charge of $a,b,c$ and $l,l'$ are the total charges of $c,d$. It is generally not guaranteed that a graph braiding exchange operator which is diagonal in the left associated basis is diagonal in the pairwise associated basis (the total charge of the anyons $c$ and $d$ may change), hence we use the matrix notation for the $\tilde{W}$ symbols acting in the pairwise associated basis.

Let us next proceed with an analysis of the equations involving the four particle symbols. The $\sigma_3^{(1,1,1,2)}$ sends the two particles closest to the junction to the back plane as displayed in Figure 6. Using the $F$-moves to join the two particles $c$ and $d$ closest to the junction by a fusion vertex, we can slide the $c \times d = l$, fusion vertex

through the graph braid. Thus, in the pairwise associated basis the braiding exchange operator corresponding to $\sigma_3^{(1,1,1,2)}$ is effectively represented via $\rho(\sigma_2^{(1_l,1_a,2_b)}) = P_{ed}^{bal}$, i.e. a $P$-symbol. We display the corresponding commutative square for $X$ and $P$ in Figure 7.

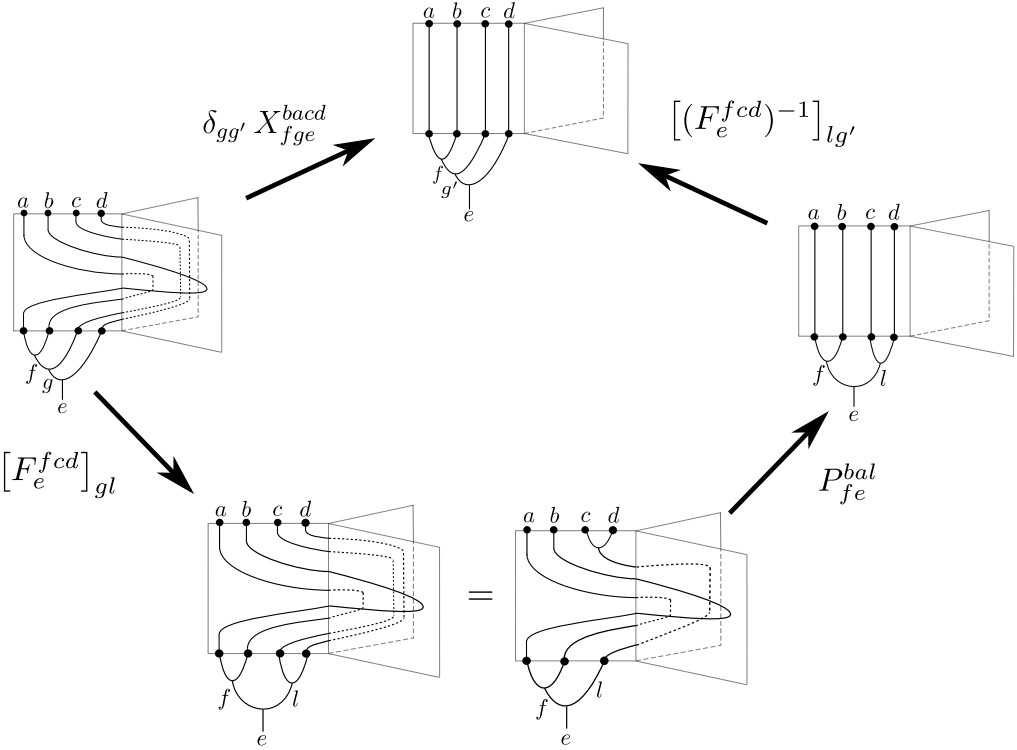

Figure 7: Here we display the polygon diagram which reduces an $X$-symbol to a $P$- symbol using fusion commuting with graph braiding. One can make an analogous figure for the $Y-$ symbol to reduce it to a $Q-$ symbol. The relevant diagram is essentially the same, except the first two particles go to edge 2 (the front plane) instead of edge 1.

This leads to the following equations,

$$X_{fge}^{bacd}\,\delta_{gg'} = \sum_l \left[F_e^{fcd}\right]_{gl} P_{fd}^{bal} \left[(F_e^{fcd})^{-1}\right]_{lg'}, \tag{18}$$

where we are explicitly imposing the diagonality of the relevant braiding exchange operator in the left fused basis. We can apply analogous reasoning the second the second generator in Equation (16) and express any $Y$-symbol as a combination of $F$-and $Q$- symbols,

$$Y_{fge}^{bacd}\,\delta_{gg'} = \sum_l^r \left[F_e^{fcd}\right]_{gl} Q_{fe}^{bal} \left[(F_e^{fcd})^{-1}\right]_{lg'}. \tag{19}$$

Hence, even though these are two new four particle generators in the graph braid group, the introduction of fusion and naturality of graph braiding allows us to express them via three particle generators. As such, in any equation utilising an $X-$ or $Y-$ symbol, we can express these symbols in terms of an equation for the $P-$ and $Q-$symbols respectively.

Consider next the two rightmost generators in Equation (16), $\sigma_3^{(1,2,1,2)}$ and $\sigma_3^{(2,1,1,2)}$. Note that the particles not being exchanged (the two closest to the junction) go to different edges. Thus, the reasoning presented in Figure 7 cannot be applied to the $A$- and $B$-symbols in order to reduce them to the $P$- or $Q$- symbols. However, one can make one further simplification. Namely, the two generators are related by the pseudocommutative relation [6], in the graph braid group,

$$\sigma_3^{(1,2,1,2)}\sigma_1^{(1,2)} = \sigma_1^{(1,2)}\sigma_3^{(2,1,1,2)}. \tag{20}$$

We can adapt this relation to our graph braiding anyon models to get the following equation which comes from an *octagon diagram*,

$$A^{badc}_{fje} \left[ F^{fdc}_e \right]_{jl} R^{dc}_l = \sum_{g,l'} \left[ F^{fdc}_e \right]_{jl'} R^{dc}_{l'} \left[ (F^{fcd}_e)^{-1} \right]_{l'g} B^{bacd}_{fge} \left[ F^{fcd}_e \right]_{gl'}. \tag{21}$$

This allows us to express the *A*-symbols via the *B*-symbols (or *vice versa*). To summarise, there are a total of 7 generators in the four-strand braid group of the trijunction, however, any anyon model can be described with only four independent sets of symbols: *R*-, *P*-, *Q*- and *B*-symbols.

Now that we have defined the action of the generators in different bases and discussed relations amongst them, we next proceed with constructing further $N = 4$ equations expressing the compatibility of graph braiding with anyon fusion. As a premise, we would like to adapt the three-particle diagram in Figure 3 where fusion commutes with graph braiding, to four particles. Recall that for $N = 3$ the relevant relations which led to the *P*- and *Q*-hexagons read

$$\sigma_1^{(1_a,2_c)} \sigma_2^{(2_c,1_a,2_b)} = \sigma_1^{(1_a,2_{b\times c})}, \quad \sigma_2^{(1_b,1_a,2_c)} \sigma_1^{(1_b,2_c)} = \sigma_1^{(1_{a\times b},2_c)}. \tag{22}$$

We can raise the above relations to $N = 4$ by conjugating both sides of the equation by a move taking anyon $d$ (closest to the junction) to edge $x$ with $x = 1, 2$. This leads to the relations

$$\sigma_2^{(x_d,1_a,2_c)} \sigma_3^{(x_d,2_c,1_a,2_b)} = \sigma_2^{(x_d,1_a,2_{b\times c})}, \quad \sigma_3^{(x_d,1_b,1_a,2_c)} \sigma_2^{(x_d,1_b,2_c)} = \sigma_2^{(x_d,1_{a\times b},2_c)}. \tag{23}$$

Note that in Equations (23) we used the convention for anyon labels given in (59). By choosing $x = 1$ we obtain two relations that allow us to express the *A*-symbols via *P*-symbols (the left relation) and the *X*-symbols via *P*-symbols (right relation). Similarly, by putting $x = 2$ we obtain two relations that allow us to express the *Y*-symbols via *Q*-symbols (the left relation) and the *B*-symbols via *Q*-symbols (right relation). One can show by a straightforward but tedious calculation that the resulting equations lead to only one independent consistency equation involving *B*- and *Q*-symbols (see also Appendix E), which comes from putting $x = 2$ in the left equation of (23) and considering the resulting octagon diagram. The resulting consistency relation reads as follows.

$$\delta_{nn'} \delta_{gg'} B^{cabd}_{nge} = \sum_{f,h,k}^r \left[ F^{cab}_g \right]_{nf} Q^{cfd}_{ge} \left[ F^{abc}_g \right]_{fh} \left[ F^{ahd}_e \right]_{gk} (Q^{cbd}_{hk})^{-1} \left[ (F^{ahd}_e)^{-1} \right]_{kg'} \left[ (F^{acb}_{g'})^{-1} \right]_{hn'}. \tag{24}$$

There is another way of realising the property of fusion commuting with braiding, namely, one can consider a $\sigma_1$-braid exchanging two composite anyons. For $N = 4$ anyons, the possible options for braiding one or two composite anyons via the simple braid $\sigma_1^{(1,2)}$ are as follows;

$$\sigma_1^{(1_{a\times b\times c},2_d)}, \qquad \sigma_1^{(1_{a\times b},2_{c\times d})}, \qquad \sigma_1^{(1_a,2_{b\times c\times d})}. \tag{25}$$

Starting from each of these braided states we can pull back the fusion vertices, similar to going from the rightmost state to the leftmost state in Figure 3. We can then resolve the resulting graph braids (i.e. expand them to obtain a concatenation of simple braids which involves the constituent factors of the composite anyons), in different ways, analogous to the planar, and graph hexagon equations. For instance, the braid in the rightmost panel from Figure 8 is the concatenation of the simple braids

$$\sigma_1^{(1_{a\times b},2_{d\times c})} = \sigma_2^{(1_b,1_a,2_d)} \sigma_1^{(1_b,2_d)} \sigma_3^{(2_d,1_b,1_a,2_c)} \sigma_2^{(2_d,1_b,2_c)}. \tag{26}$$

The relation (26) can be derived by iteratively applying the relations (23) and (22). What is more, the polygon equations obtained this way do not yield any new constraints for the relevant symbols as they readily follow from the equations obtained from the relations (23), (22) and the squares (18) and (19). We have checked that the same fact holds for all the relations stemming from braiding composite anyons using $\sigma_1$- and $\sigma_2$- graph braids. This suggests that the polygon equations (18), (19), (21) and (24) are all the consistency relations which are needed for the compatibility of fusion and graph braiding of four anyons on a trijunction. However, we do not have a rigorous proof of this fact.

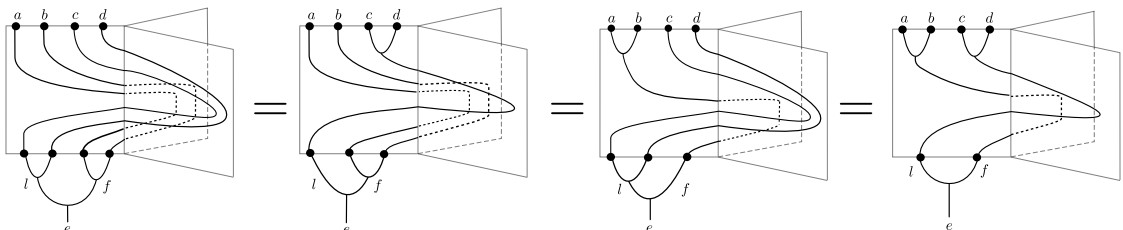

Figure 8: The fusion vertices of anyons $a, b$ and $c, d$ can be pulled through the simple braid $\sigma_1^{(1_{a \times b}, 2_{d \times c})}$ involving two composite anyons. The diagram on the furthest left expresses the composition; $\sigma_2^{(1_b, 1_a, 2_d)} \sigma_1^{(1_b, 2_d)} \sigma_3^{(2_d, 1_b, 1_a, 2_c)} \sigma_2^{(2_d, 1_b, 2_c)}$ of graph braids. This is in analogy to the relation from Figure 3 which allowed us to derive the $Q$-hexagon equations.

Another important property of the graph braided anyon models is that any symbol representing a $\sigma_j$ graph braid, with $j > 1$ can be expressed an appropriate of products of $F$-symbols and $R$-symbols. This is because one can reduce any $\sigma_3$-braid to a product of $\sigma_2$-braids (involving composite anyons) by using the relations (23). The resulting $\sigma_2$-braids can be in turn reduced to products of $\sigma_1$-braids involving composite anyons by applying relations (22). As the final result, we obtain that any $\sigma_3$-braid is a product of $\sigma_1$-braids which involve appropriate exchanges of composite anyons. Thus, translating this relation to the braiding exchange operators acting on the left-fused basis we are able to express the $A$-, $B$-, $X$- and $Y$- symbols as sums of products of $R$-symbols and $F$- symbols. This fact generalises in a straightforward way to $N > 4$, see Appendix C.

In Section 6 we have applied the $N = 4$ polygon equations (18), (19), (21) and (24) to chosen anyon models of low rank. Importantly, we have found numerous examples of Abelian and non-Abelian anyon models that satisfy all the above polygon equations and are different from planar braiding models. Examples include the Abelian $\mathbb{Z}_n$ anyons, the Ising anyons, Tambara-Yamagami anyons over $\mathbb{Z}_2 \times \mathbb{Z}_2$ and $D(\mathbb{Z}_2)$.

A natural question follows: does this procedure ever end? Namely, do we have to consider higher and higher particle numbers leading to more complicated fusion diagrams which may further constrain our anyon model? By considering the pseudocommutative relations and using the commutativity of fusion and braiding for $N > 4$ [6] one can see that any graph braid of the type $\sigma_j$ can be expressed by $F$-, $R$-, $P$-, $Q$- and $B$-symbols (see Appendix C for more explanation). Thus, no new symbols are introduced for $N > 4$. However, there still may be some new relations appearing in $N > 4$ systems. In Appendix C we take steps toward resolving this issue by conjecturing that it is enough to consider the polygon equations derived from braiding diagrams of $N = 5$ particles on a trijunction. In other words, we conjecture that the graph-braided anyon models will be coherent for $N > 5$ particles. Moreover, we conjecture that on top of the $N = 4$ polygon consistency relations introduced in this section, the only new relations appearing for $N = 5$ systems come from imposing diagonality of certain braiding exchange operators in appropriate bases (relations analogous to the square equations (18), (19)). In Appendix C we provide evidence for the existence of above generalised coherence property and sketch a possible pathway for proving it.

## 3.2 Anyon models with simplified symbols

In general, it is a computationally complex problem to determine the braiding exchange operator that corresponds to an arbitrary $\sigma_j$ graph braid. However, there exists an important simplification which resolves this issue and still leads to graph-braided anyon models that are not planar and which (conjecturally) become coherent already for $N > 4$. These are the models where the braiding exchange symbols depend only on at most four labels, namely on i) the charges of the exchanging anyons – $a$ and $b$, ii) the total charge of $a$ and $b$ – $c$, iii) the total charge of $a$, $b$ and all the anyons standing between $b$ and the junction point – $d$. In other words, if we have $N$ anyons exchanging on a trijunction and the anyon types are given by the sequence $a_N, \ldots, a_{N-j-1}, b, a, a_{j-1}, \ldots, a_1$ (where $a$ and $b$ are the anyons that exchange), then $c = a \times b$ and $d = b \times a \times a_{j-1} \times \cdots \times a_1$. We define the simplified symbols of the theory by dropping certain labels as follows

$$ R_c^{ba}, \quad P_{cd}^{ba}, \quad Q_{cd}^{ba}, \quad B_{cd}^{ba}. $$

See Appendix C for more explanation. The models with such simplified symbols have the property that all the $\sigma_j$ graph braids are described by the same symbol, regardless of the edges that are visited by the anyons

$a_1, \ldots, a_{j-1}$ and independently of the fusion tree of the anyons $a_1, \ldots, a_{j-1}$. In particular, if the anyons $a_1, \ldots, a_{j-1}$ visit edge 1, then the braid $\sigma_j^{(1,\ldots,1,1,2)}$ is always resolved by a $P$-symbol. Similarly, the braid $\sigma_j^{(2,\ldots,2,1,2)}$ is always resolved by a $Q$-symbol. If at least two of the anyons $a_1, \ldots, a_{j-1}$ visit two different edges, then the corresponding $\sigma_j$ graph braid is always resolved by a $B$-symbol. Importantly, both the Ising anyon model and the Tambara-Yamagami $\mathbb{Z}_2 \times \mathbb{Z}_2$ anyon model which for $N = 4$ have solutions different than planar, turn out to realise such a graph braided model with the simplified symbols. We further conjecture that for the simplified anyon models the coherence is attained already for $N = 5$, i.e. no new constraints appear for $N > 4$. This conjecture implies in particular that the graph-braided Ising anyon model has the free parameter $R_1^{\sigma\sigma}$ for any $N$.

### 3.3 The $H$-graph and general tree graphs

Let us next move on to consider a simple network consisting of two trijunctions joined along one edge. The resulting graph is the $H$ graph, denoted $\Gamma_H$. The features of anyon braiding models, which we describe in this section, also extend naturally to any tree graph. The $H$-graph is displayed in Figure 9. The two junction points are denoted by $v$ and $w$, with $v$ being the junction closest to anyons' initial position. The three-strand graph braid group $B_3(\Gamma_H)$ is freely generated by the following simple braids [19, 6, 26] (see also Appendix A for more explanation)

$$\sigma_1^{v;(1,2)}, \quad \sigma_2^{v;(2,1,2)}, \quad \sigma_2^{v;(1,1,2)}, \quad \sigma_1^{w;(1,2)}, \quad \sigma_2^{w;(2,1,2)}, \quad \sigma_2^{w;(1,1,2)}. \tag{27}$$

In other words, each junction point permits an exchange of particles and exchanges at different junctions are topologically inequivalent. Consequently, the exchanges at $v$ will be represented by different symbols than the exchanges at $w$. Namely,

$$\rho\left(\sigma_1^{v;(1,2)}\right) = R, \quad \rho\left(\sigma_1^{v;(1,1,2)}\right) = P, \quad \rho\left(\sigma_1^{v;(2,1,2)}\right) = Q,$$
$$\rho\left(\sigma_1^{w;(1,2)}\right) = \tilde{R}, \quad \rho\left(\sigma_1^{w;(1,1,2)}\right) = \tilde{P}, \quad \rho\left(\sigma_1^{w;(2,1,2)}\right) = \tilde{Q}.$$

Moreover, we have two different sets of hexagon equations, with each set of hexagons coming from embedding a trijunction at $v$ or $w$, respectively. There is one $P$-hexagon (see (14)) involving $P$-symbols and $R$-symbols and one $P$-hexagon involving $\tilde{P}$-symbols and $\tilde{R}$-symbols. Similarly, we have one $Q$-hexagon (see (13)) involving $Q$-symbols and $R$-symbols and one $Q$-hexagon involving $\tilde{Q}$-symbols and $\tilde{R}$-symbols.

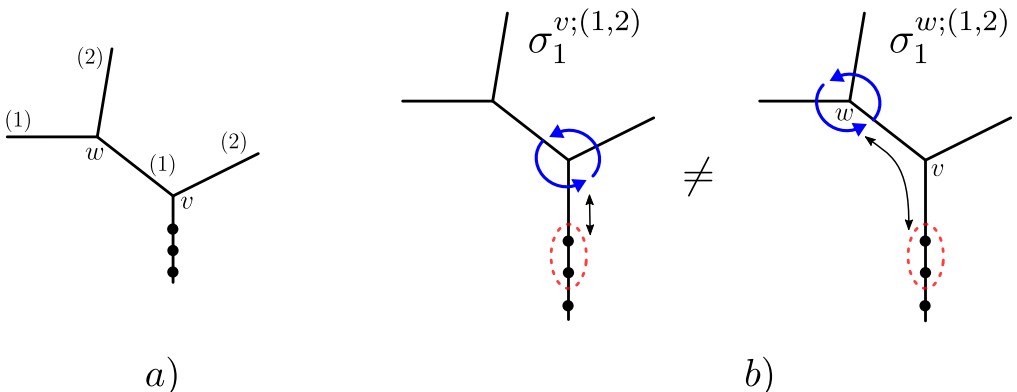

Figure 9: a) The $H$-graph. It contains two junction points denoted by $v$ and $w$. The branches of each junction are enumerated by (1) and (2) relative to the orientation of the junction with respect to the initial configuration of the anyons (black dots). b) A schematic picture showing that the exchanges at junction $v$ are topologically independent of the exchanges at junction $w$.

We can observe that the graph braid group of $\Gamma_H$ for $N = 3$ particles is essentially two copies (formally speaking, the free product) of the trijunction graph braid group, generated by exchanges at the junctions $v$ and $w$. A natural question presents itself: Could one construct a new independent consistency equation involving fusion commuting with braids at $v$ and $w$ simultaneously? The answer is no, which we will now explain.

As discussed at the beginning of this section and in [18], to introduce consistency equations from fusion commuting with graph braiding, two of the three particles must go to the same edge, and they must be joined by a fusion vertex throughout the entire exchange, so that we can "slide" the fusion vertex through the graph braid diagram. We can see an example of this in Figure 3. Consider next similar reasoning for the $H$-graph. Assume that the labels of the anyons in Figure 9 are $a$, $b$, $c$ with anyon $c$ being the closest to the junction and anyon $a$ being the furthest from the junction. In order to look for possible new relations, we need to consider all the possible exchange processes where a pair of anyons stays joined by a common fusion channel so that the fusion vertex can be pulled through the worldline diagram of the entire process. If this is the case, one obtains a new relation by comparing the effective two-particle exchange process (where the two anyons stay joined by a common fusion channel) with the original three-particle exchange process. Two possible options exist for joining the neighbouring anyons by a common fusion channel. Namely, anyons $b$ and $c$ are joined together into anyon $f$ or anyons $a$ and $b$ are joined together into anyon $e$. Suppose we slide the fusion vertex throughout the worldline diagram of a three-particle exchange process. In that case, we are left with an effective two-particle exchange process involving anyons $a$ and $f$ or $e$ and $c$, respectively. However, all two-particle exchange processes are generated by the $\sigma_1^v$ and $\sigma_1^w$ with appropriate superscripts. Thus, it is enough to consider the consistency diagrams where fusion commutes with braiding only for these types of generators. In the case when $a$ and $b$ are joined into anyon $e$, this leaves us only with the following four options for exchanges taking place at $v$ or $w$: $\sigma_1^{v;(1_c,2_e)}$, $\sigma_1^{v;(2_c,1_e)}$, $\sigma_1^{w;(1_c,2_e)}$, $\sigma_1^{w;(2_c,1_e)}$. These can only lead to separate $P$- and $Q$-hexagons for $R$ and $P/Q$ or $\tilde{R}$ and $\tilde{P}/\tilde{Q}$ respectively. Similarly, we reproduce the same set of hexagon equations when considering exchanges of $f = b \times c$ with $a$.

Consider next $N = 4$. Using the orientation of the junctions shown in Figure 9a), we have that $B_4(\Gamma_H)$ is generated by the simple braids listed in Equation (27) together with the following six $\sigma_3$-braids

$$\sigma_3^{u;(1,1,1,2)}, \quad \sigma_3^{u;(2,1,1,2)}, \quad \sigma_3^{u;(2,2,1,2)}, \quad u = v, w.$$

Consequently, the simple exchanges at $v$ are represented by one set of symbols $R, P, Q, B$ (as explained in Section 3.1) and the simple exchanges at $w$ are represented by another set of symbols $\tilde{R}, \tilde{P}, \tilde{Q}, \tilde{B}$. However, in contrast to the three-strand braid group, the four-strand braid group $B_4(\Gamma_H)$ is no longer freely generated, as we have the following commutative relation [6]

$$\sigma_3^{v;(1,1,1,2)} \sigma_1^{w;(1,2)} = \sigma_1^{w;(1,2)} \sigma_3^{v;(1,1,1,2)}. \tag{28}$$

Intuitively, relation (28) means that two disjoint pairs of anyons can be exchanged at different junctions independently. Interestingly, this relation does not impose any constraints on the corresponding symbols in the anyon model. This is due to the fact that the simple exchange $\sigma_3^{v;(1,1,1,2)}$ can be effectively represented by a $P$-symbol using the pairwise-fused basis (explained in Section 3.1) describing a spacetime process where the two anyons closest to the junction remain fused at all times. In such a pairwise-fused basis relation (28) is satisfied automatically provided that the square diagram (18) is satisfied.

To reiterate, all the possible exchanges with two out of the three anyons fused together only lead to hexagon equations which concern exchanges that are fully localised on one of the junctions. This implies that one can treat the solutions at different trijunctions of the $H$-graph as independent. For instance, if we chose an anyon model on a trijunction whose solutions to the polygon equations (18), (19), (21) and (24) have free parameters (e.g. Ising fusion rules where the $R$-symbol $R_1^{\sigma\sigma}$ is a free parameter), then these parameters remain free on the $H$-graph. Moreover, there will be two independent sets of free parameters since braids at $v$ and $w$ are topologically inequivalent. If we joined more and more trijunctions forming a tree architecture, then we could make further independent choices for the free parameters at each junction point. In Section 8 we argue that this property of graph-braided anyon models may be useful for designing more efficient topological quantum computing circuits.

## 4  Braiding and fusion on the circle

Having revisited the graph anyon models on the simplest building block of networks, i.e. the trijunction, we proceed to define an analogous construction for another simple building block which is the circle. Following this, we will study the interplay between both of these situations by moving to a lollipop graph which consists

of a single trijunction and a single loop. On a circle, we first arrange particles next to each other at a particular place on the circle (which is equivalent to fixing the basepoint for the generator of the braid group $B_N(S^1)$). We can then change the ordering by cycling particles around the loop, this is given by the move $\delta$. In other words, the braid group of the circle is a free group on one generator which we denote by $\delta$. It is uniquely defined by picking an orientation of the circle. Here, we assume the orientation to be counterclockwise. The action of $\delta$ moves one of the outermost particles around the circle according to the circle's orientation as shown in Figure 10.

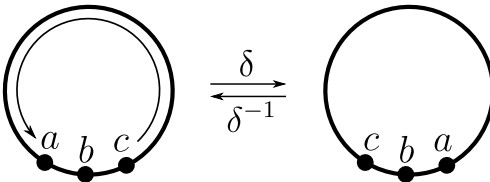

Figure 10: The $\delta$-move.

With the $\delta$- move we associate the $D$-symbols that depend on three anyon labels as shown in Figure 11.

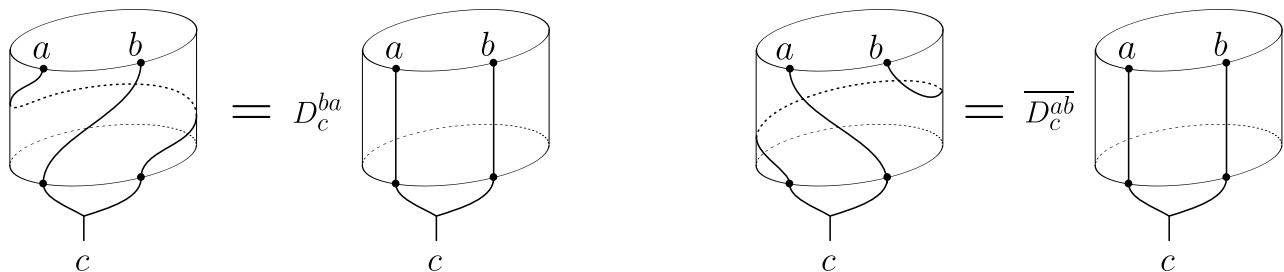

Figure 11: The braiding exchange operator associated with the $\delta$-braid is described via $D$-symbols. The definition extends in a natural way to the $\delta$-move involving $N > 2$ anyons by fusing together the $N - 1$ anyons which do not travel around the circle and by sliding their fusion vertex effectively obtaining the $\delta$-move acting on two anyons only. Note that with the above convention we necessarily have $D_b^{b1} = 1$ and $\overline{D_a^{a1}} = 1$ (the trivial anyon going around the circle), but $D_a^{1a}$ and $\overline{D_b^{1b}}$ are generally different from one.

The gauge transformations of the $D$-symbols have the same structure as the gauge transformations of the planar $R$-symbols as explained in Appendix E. Requiring the fusion to commute with the $\delta$-braid leads to two families of hexagon equations shown in Figure 12 and Figure 13.

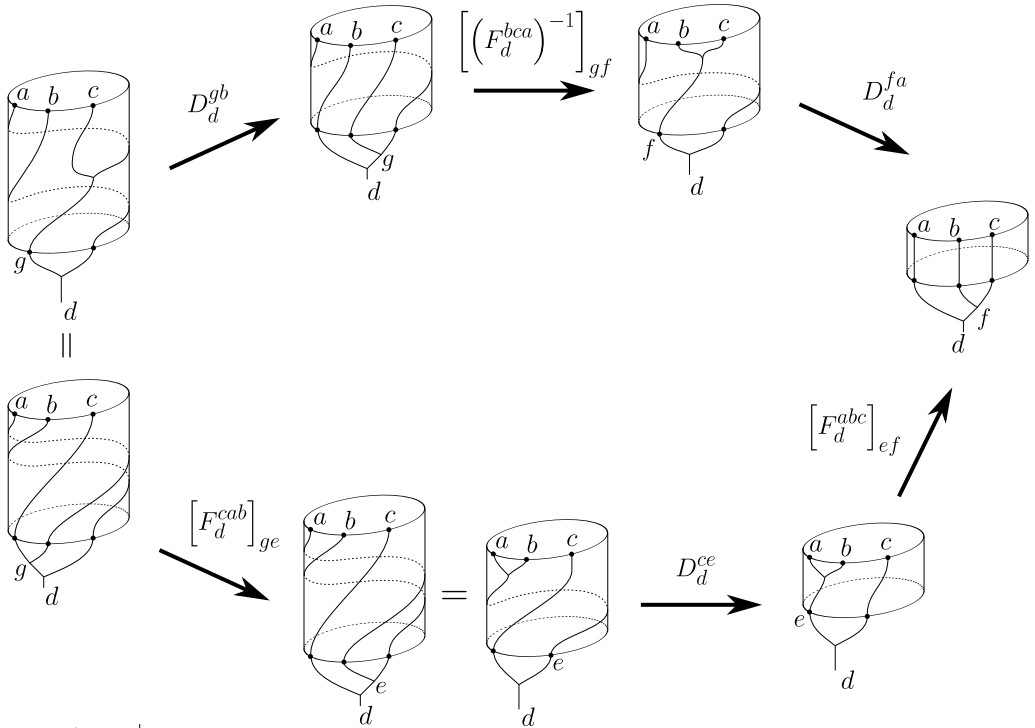

Figure 12: The naturality condition for $\delta$ showing that fusion commutes with the $\delta$-braid for $N = 3$. Equating the upper and lower path leads to Eq. (29).

The second set of hexagon equations comes from demanding the $\delta^{-1}$-move to be compatible with fusion.

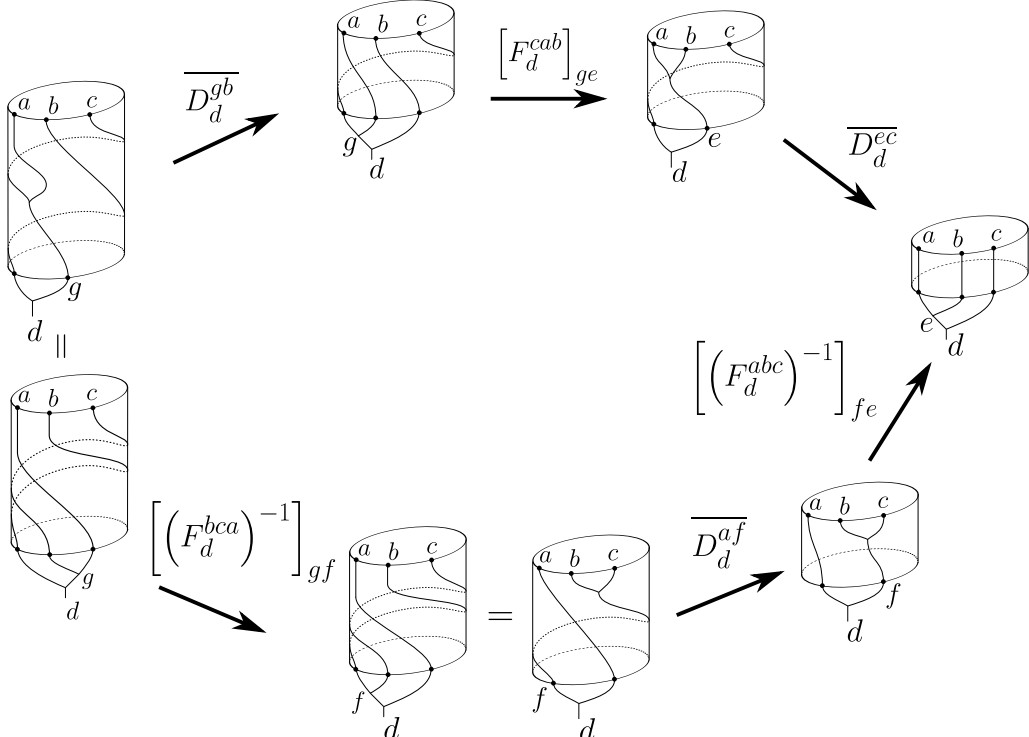

Figure 13: The naturality condition for $\delta^{-1}$ showing that fusion commutes with the $\delta^{-1}$-braid for $N = 3$. This leads to another set of hexagon equations given in Equation (30).

$$D_d^{gb} \left[ (F_d^{bca})^{-1} \right]_{gf} D_d^{fa} = \sum_e \left[ F_d^{cab} \right]_{ge} D_d^{ce} \left[ F_d^{abc} \right]_{ef} \tag{29}$$

$$\overline{D_d^{gb}} \left[ F_d^{cab} \right]_{ge} \overline{D_d^{ec}} = \sum_f \left[ \left( F_d^{bca} \right)^{-1} \right]_{gf} \overline{D_d^{af}} \left[ \left( F_d^{abc} \right)^{-1} \right]_{fe}. \tag{30}$$

In fact, hexagons (30) follow from the hexagons (29). To see that, put $c = 1$ in (29) to obtain

$$D_e^{ab} D_e^{ba} = D_e^{1e}. \tag{31}$$

Then, apply the above identity to the RHS of (30) as $\overline{D_d^{af}} = D_d^{fa} \overline{D_d^{1d}}$ and insert $1 = D_d^{gb} \overline{D_d^{gb}}$ to obtain

$$\sum_f \left[ \left( F_d^{bca} \right)^{-1} \right]_{gf} \overline{D_d^{af}} \left[ \left( F_d^{abc} \right)^{-1} \right]_{fe} = \overline{D_d^{gb}} \sum_f D_d^{gb} \left[ \left( F_d^{bca} \right)^{-1} \right]_{gf} D_d^{fa} \overline{D_d^{1d}} \left[ \left( F_d^{abc} \right)^{-1} \right]_{fe}.$$

Next, under the above sum, we recognise the LHS of (29), thus we can rewrite it as the double sum which we subsequently sum over $f$

$$\overline{D_d^{gb} D_d^{1d}} \sum_{f,e'} \left[ F_d^{cab} \right]_{ge'} D_d^{ce'} \left[ F_d^{abc} \right]_{e'f} \left[ \left( F_d^{abc} \right)^{-1} \right]_{fe} = \overline{D_d^{gb} D_d^{1d}} \sum_{e'} \delta_{ee'} \left[ F_d^{cab} \right]_{ge'} D_d^{ce'} =$$

$$= \overline{D_d^{gb}} \left[ F_d^{cab} \right]_{ge} D_d^{ce} \overline{D_d^{1d}}.$$

Finally, we use (31) again to obtain $D_d^{ce} \overline{D_d^{1d}} = \overline{D_d^{ec}}$ and the above expression becomes the LHS of (30).

As a final comment to this section, we note the connection of the $D$-symbols $D_a^{1a}$ to the twist factors. The symbol $D_a^{1a}$ is associated with the $\delta$-move taking just a single anyon $a$ around the circle. This is exactly the move which in the 2$D$ anyon theory corresponds to the topological twist. Indeed, for every anyon model, the solutions to the $D$-hexagons (29) always contain the topological twist $\theta_a$ expressed in terms of the planar $R$-symbols in Equation (6) as a special case. However, for our graph anyon models we do not have the relation (6) and thus we *define* the generalised topological twist as

$$\theta_a := D_a^{1a}. \tag{32}$$

So-defined topological twists typically can have more possible values than their counterparts known from the 2$D$ theory. For example, anyons with $\mathbb{Z}_3$ fusion have only third roots of unity as conventional twists, while the solutions to equations (29) also allow for ninth roots of unity as topological twists. Another example is the TY($\mathbb{Z}_3$) fusion category which admits no braiding at all, yet has solutions to equations (29). These solutions can be found in Appendix I.2.

Importantly, the above defined anyon theory on the circle is readily coherent, i.e. the $D$-hexagon (29) implies the compatibility of anyon fusion with the $\delta$-braid for any $N > 3$ (see Appendix B for the proof). We present solutions of the $D$-hexagons for low-rank anyon models in Section 6 and Sections G.4.2, H.3.2 and I.2. We have found that all the tested models are rigid, i.e. have a finite number of solutions with no free parameters left. We note that our anyons on a circle graph bears a striking resemblance to the tube category, see for example [31], however, establishing this connection rigorously is outside the scope of this work.

## 5 The lollipop graph

The next key step is to incorporate loops and junctions into a single graph. The simplest possible configuration is the lollipop graph, $\Gamma_L$, shown in Figure 14.

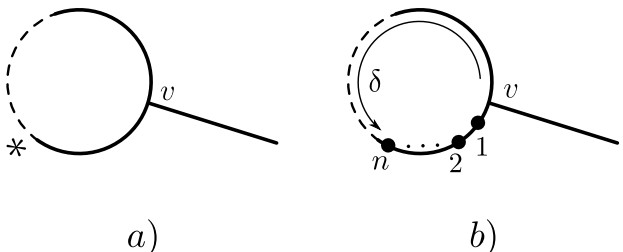

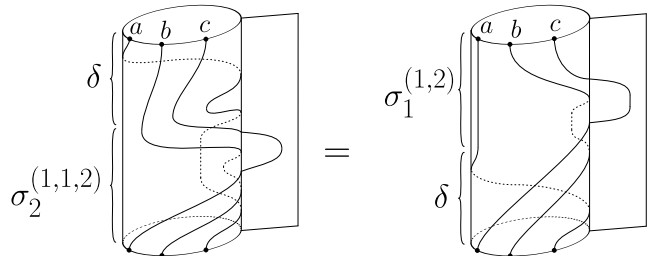

Figure 14: The lollipop graph. a) Our choice of the rooted spanning tree (solid lines) with the root $*$ which determines an embedding of the trijunction graph into the lollipop. The resulting deleted edge is marked by the dashed line. b) The base configuration of anyons corresponding to our choice of the rooted spanning tree and the $\delta$-move coming from embedding the circle-subgraph into the lollipop.

The lollipop graph contains one loop, with which we associate a $\delta$-move and one essential vertex $v$, with which we associate the simple graph braids. This is done via the embedding of the trijunction graph shown in Figure 14a (and presented in more detail in Appendix A). In other words, the graph braid group $B_3(\Gamma_L)$ is generated by

$$\delta, \quad \sigma_1^{(1,2)}, \quad \sigma_2^{(1,1,2)}, \quad \sigma_2^{(2,1,2)}.$$

The above generators are subject to one relation which connects the $\delta$-braid with the simple graph braids. Namely, we have (see also Figure 15)

$$\delta\sigma_1^{(1,2)} = \sigma_2^{(1,1,2)}\delta. \tag{33}$$

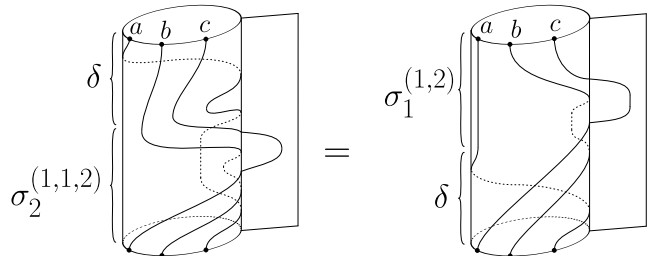

Figure 15: A pictorial proof of the lollipop relation (33).

This leads to the square diagram shown in Figure 16.

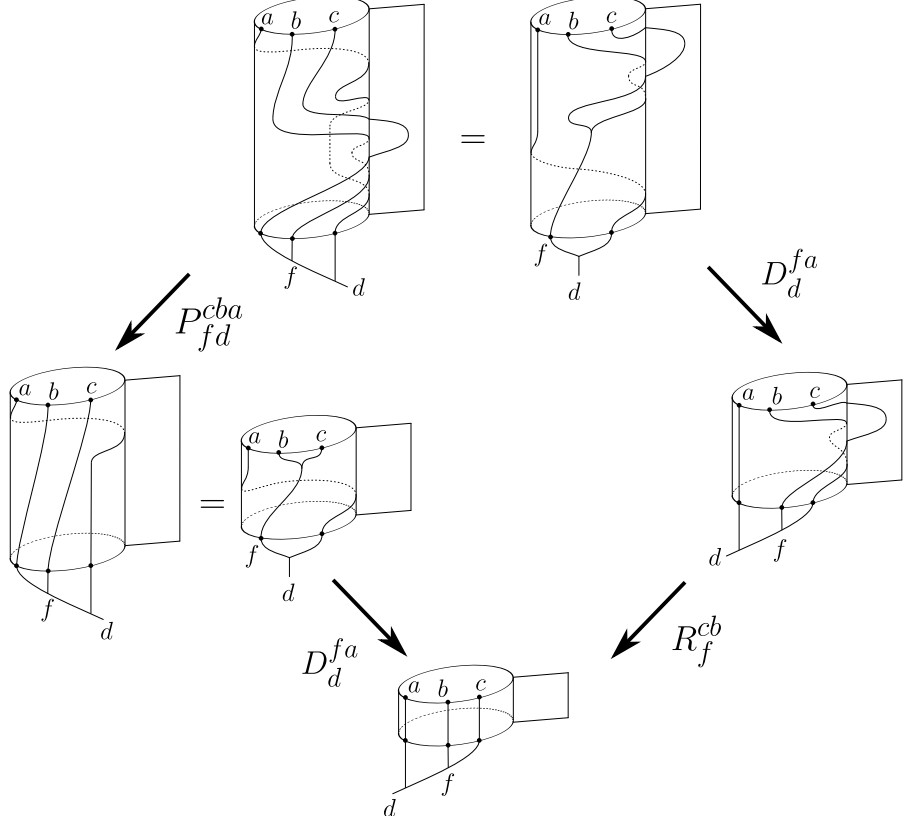

Figure 16: The square diagram corresponding to (33). The homotopy relation from Figure 15 has been used in the top panel of the diagram. The two rightmost arrows represent the braiding exchange operators corresponding to the $\delta$-move followed by the simple braid $\sigma_1^{(1,2)}$. The two leftmost arrows represent the braiding exchange operators corresponding to the simple braid $\sigma_2^{(1,1,2)}$ followed by the $\delta$-move.

The resulting equation reads

$$D_d^{fa} P_{fd}^{cba} = R_f^{cb} D_d^{fa},$$

Notably, the diagram 16 does not use any $F$-symbols. Using the fact that the $D$-symbols $D_d^{fa} \in U(1)$, the above equation boils down to

$$P_{fd}^{cba} = R_f^{cb}. \tag{34}$$

On top of the condition (34) the $P$- and $Q$- hexagons (14) and (13) are also valid equations for the lollipop as they describe the simple graph braids at the junction of the lollipop. Note that putting $P = R$ in the $P$-hexagons readily reproduces one set of the hexagon equations from the planar anyon theory (5). In other words, creating a lollipop from a trijunction by creating a single loop makes the graph braided anyon model more similar to the planar braided anyon model. As we will see in Section 7, one can continue this line of thought to make a complete transition to the planar anyon theory by considering the graph braided anyon theory on the theta-graph and more generally, on the family of triconnected graphs.

## 5.1 The $\triangle$-move

There is an auxiliary braid on the lollipop which we will extensively use in Section 7. It is the braid $\Delta$ defined in Figure 17 which takes into account the possibility of an anyon occupying the lollipop's stick while the remaining two anyons do a $\delta$-like-move. It is expressed by the standard generators as

$$\Delta = \sigma_1^{(2,1)} \delta, \tag{35}$$

where $\sigma_1^{(2,1)}$ is the inverse of the simple braid $\sigma_1^{(1,2)}$.

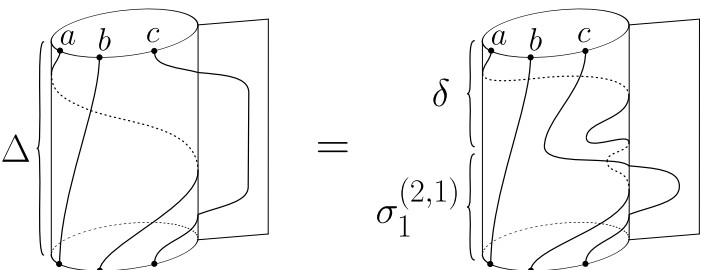

Figure 17: The braid $\Delta$. Note that if $c = 1$ (the trivial anyon), then $\Delta$ reduces to $\delta$.

The braid $\Delta$ will be represented by the $G$-symbols as shown in Figure 18.

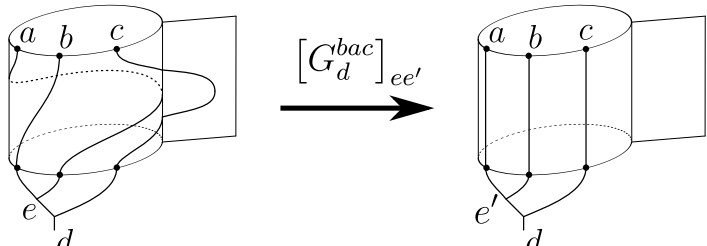

Figure 18: The definition of the $G$-symbols. The anyon $c$ moves out of the way of anyon $a$ so that $a$ can exchange with $b$ utilising the circle of the lollipop.

The relation (35) leads to the hexagon diagram from Figure 19 which connects $G$-, $D$- and $R$-symbols.

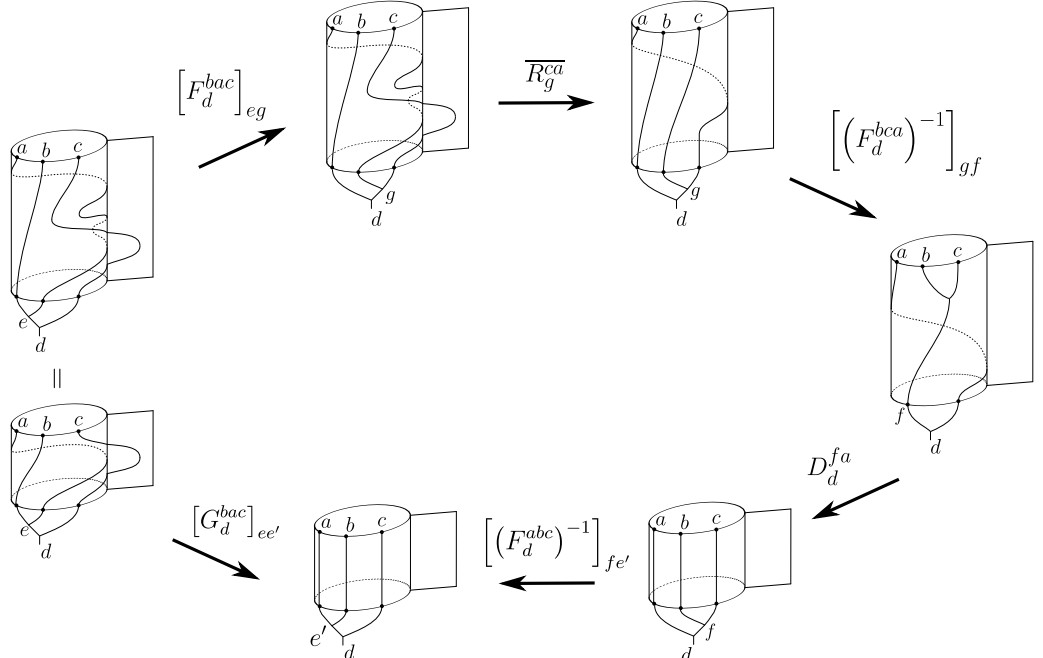

Figure 19: The hexagon following from the relation (35).

$$\left[G_d^{bac}\right]_{ee'} = \sum_{g,f}\left[F_d^{bac}\right]_{eg}\overline{R_g^{ca}}\left[\left(F_d^{bca}\right)^{-1}\right]_{gf}D_d^{fa}\left[\left(F_d^{abc}\right)^{-1}\right]_{fe'}. \tag{36}$$

In particular, if $b = 1$ we obtain the relation

$$\left[G_{ag}^{1ac}\right]_{aa'} = D_g^{ca}\overline{R_g^{ca}}\delta_{aa'}. \tag{37}$$

Furthermore,

$$\left[G_f^{b1c}\right]_{bb'} = \delta_{bb'}, \quad \left[G_e^{ba1}\right]_{ee'} = \delta_{ee'}D_e^{ba}.$$

Just as in the case of the $D$-symbols there is a completely analogous naturality for the $G$-symbols, which follows from the hexagon in Figure 19.

Note that the planar anyon theory is retrieved from the graph braided anyon theory on the lollipop by imposing that $G_d^{bac}$-symbols are independent of $c$ in which case the $G$-hexagons imply

$$\left[G_d^{bac}\right]_{ee'} = D_e^{ba}\delta_{ee'} \tag{38}$$

and become equivalent to the condition $Q = R$. This is because substituting in the hexagon equations (36) i) the $G$-symbols with the $D$-symbols according to Equation (38) and ii) $D$-symbols $D_g^{ca}$ with $\overline{R}_g^{ca}\theta_a$ (recall $\theta_a := D_a^{1a}$) according to Equation (37) makes the hexagon equations (36) equivalent to the $Q$-hexagons with $Q = R$. Thus, we have that $Q = P = R$, so under these assumptions, the simple braids on the lollipop are represented by the same $R$-symbols as the ones coming from the planar anyon theory. What is more, the symbols $G_g^{1ac}$ then acquire the interpretation as the twist factors, i.e.

$$G_g^{1ac} = \theta_a = D_a^{1a}.$$

Interestingly, the so-defined twist factors (by satisfying the extra condition (38)) can still differ from the planar twist factor (defined in equation (6)).

# 6 Solutions to the graph braiding equations

We solved the graph braiding equations for the circle, the trijunction (with three and four particles), and the lollipop graph for the following anyon models: $\mathbb{Z}_2$, Fibonacci, Ising, $\text{Rep}(D_3)$, $\text{PSU}(2)_5$, $\mathbb{Z}_3$, $\mathbb{Z}_2 \times \mathbb{Z}_2$, $\text{SU}(2)_3$, $\mathbb{Z}_4$, $\text{TY}(\mathbb{Z}_3)$, $\text{Rep}(D_4)$, and $\text{SU}(2)_4$.

Some of these anyon models have different properties when braiding is confined to a graph rather than the plane. There exist, in particular, several fusion categories that never admit planar braiding, despite having solutions for the graph-braid equations. For the anyon models we studied, we observed the following:

- The equations (29) for anyons on a circle, like the planar hexagon equations, lead to discrete sets of solutions. There are always at least as many solutions as the planar hexagons allow. Interestingly, the equations for a circle sometimes admit solutions even when the planar hexagons do not. The $\text{TY}(\mathbb{Z}_3)$ fusion model (see I.2 for the solutions) is such an example.

- The three-particle trijunction equations are the $P$- and $Q$-hexagons (equations (13) and (14)). Increasing the number of particles to $N = 4$ means adding equations (18), (19), (21) and (24). As was pointed out in [18], solutions to the trijunction equations for three particles sometimes contain free parameters. If we add the equations for four particles, then, depending on the model, this freedom either remains unaltered (e.g. for Abelian anyons), gets partially restricted (e.g. Ising anyons), or disappears completely (e.g. $\text{Rep}(D_3)$ anyons). For the models we investigated, we found that if a model has solutions for the three particle equations, it also has solutions for the four particle equations. Specific results on the number of free variables and solutions to the trijunction equations can be found in table 1.

- The equations for the lollipop graph consist of (a) the trijunction $P$- and $Q$-hexagons (14) and (13), (b) equations demanding equality between the $P$ and $R$ symbols (34), and (c) equations for anyons on a circle (29). We will call the combined set of (a) and (b) the lollipop trijunction equations. The lollipop trijunction equations are sufficient to fix all degrees of freedom in the standard trijunction solutions. Since the equations on a circle give rise to a discrete set of solutions, all investigated models have a discrete set of solutions to the full lollipop equations. Let $n_c, n_t, n_l$ denote the number of gauge-inequivalent solutions to the circle equations, lollipop trijunction equations, and full lollipop equations, respectively. Although the equations for a circle graph are independent of the lollipop trijunction equations, $n_l$ need not be equal to the product $n_c n_t$. This happens when there is still some gauge freedom left after fixing the values of the $F$-symbols. In this case, the number of solutions to each set of equations gets reduced by the same factor. This implies that the number of gauge-independent solutions to the combined set of equations will be greater than the product of the number of solutions of the individual equations. For the cases studied only the $\mathbb{Z}_2 \times \mathbb{Z}_2$ model has remaining gauge symmetry. More information on the number of solutions to the planar hexagon equations, the circle equations, lollipop trijunction equations, and full lollipop equations can be found in tables 2 and 3.

If all the anyons are Abelian (i.e. the fusion algebra is a group algebra), then:

- The trijunction equations are trivially fulfilled for 3 and 4 particles. All non-trivial $R$- symbols are thus free variables for trijunction. In particular, each set of trijunction equations admits continuous parameter families of solutions. This is not the case for the planar hexagon equations. For, e.g., $\mathbb{Z}_3$ anyons only the trivial $F$-symbols admit a braided structure and for $\mathbb{Z}_2$ and $\mathbb{Z}_2 \times \mathbb{Z}_2$ only half of the sets of $F$-symbols admit a braided structure.

- For the circle, Lollipop trijunction, and full lollipop equations, we find that, for a fixed anyon model, each set of $F$-symbols gives rise to the same number of solutions. If the $F$-symbols allow solutions to the planar hexagon equations, then some of the solutions to the lollipop equations are also planar. The number of planar solutions to the lollipop equations is always greater than the number of solutions to the hexagon equations. For more information on the number of solutions to the lollipop equations for Abelian anyons, see table 3.

If some of the anyons are not Abelian then:

- The solutions to the trijunction equations without free variables are always planar, and the solutions with free variables are planar for a discrete set of values of the free variables.

- All solutions to the lollipop equations are planar. The number of planar solutions to the lollipop equations is always greater than the number of solutions to the hexagon equations.

For more information on how we solved these equations, see Appendix E.

| Fusion Algebra | Solutions to the trijunction hexagon equations per set of unitary $F$-symbols | | | | |
| --- | --- | --- | --- | --- | --- |
| | $N = 3$ | | $N = 4$ | | |
| | # Solutions | # Free Variables | # Solutions | # Free Variables | Planar? |
| Fibonacci | 2 | None | 2 | None | Always |
| Ising | 2 | 2 | 2 | 1 | UCC |
| PSU(2)$_5$ | 2* | None | 2 | None | Always |
| SU(2)$_3$ | 2* | 2 | 2 | 1 | UCC |
| SU(2)$_4$ | 2* | 2 | 2 | 1 | UCC |
| TY($\mathbb{Z}_3$) | 0 | | | | |
| Rep($D_4$) | 4 | 10 | 4 | 1 | UCC |

Table 1: Properties of solutions to the trijunction equations for three and four particles for various non-Abelian anyon models. Here UCC means that under certain conditions on the free $R$-symbols the solutions are planar. All solutions listed are gauge-inequivalent. Note that the number of solutions corresponds to the number of gauge-inequivalent families of solutions, possibly parametrized by some free variables.
*For these models we only obtained solutions for 1 set of unitary $F$-symbols per model. See Appendix E for more info.
**For Rep($D_3$) it looks like there are more solutions to the equations for $N = 4$, but this is only due to the fact that for $N = 4$ all free parameters are fixed and thus instead of 2 continous families of solutions we find 3 discrete families of solutions.

| Fusion Algebra | Amount of solutions per type of equations (3 particles) per set of unitary $F$-symbols | | | |
|---|---|---|---|---|
| | Planar Hexagon | Circle | Lollipop Trijunction | Full Lollipop |
| Fibonacci | $2$ | $2$ | $2$ | $2^2$ |
| Ising | $2^2$ | $2^4$ | $2^2$ | $2^6$ |
| PSU$(2)_5$ | $2^*$ | $2^2$ | $2$ | $2^3$ |
| Rep$(D_3)$ | $3, 0, 0$ | $3, 3, 3$ | $3, 0, 0$ | $3^2, 0, 0$ |
| SU$(2)_3$ | $2^*$ | $2^6$ | $2$ | $2^7$ |
| TY$(\mathbb{Z}_3)$ | $0$ | $3$ | $0$ | $0$ |
| SU$(2)_4$ | $2^*$ | $2^8$ | $2$ | $2^9$ |
| Rep$(D_4)$ | $2^3$ | $2^7$ | $2^3$ | $2^{10}$ |

Table 2: Number of gauge inequivalent solutions to the consistency equations for various non-Abelian anyon models. Except for the planar hexagon equations all equations were constructed for systems with only three anyons. All of the solutions to the lollipop trijunction equations in this table are planar, i.e. $P = Q = R$. For Rep$(D_3)$ a different amount of solutions was found for the different solutions to the pentagon equations and so we used a notation where the $i^{th}$ number in each column corresponds to data regarding the $i^{th}$ solution to the pentagon equations. *For these models we only obtained solutions for 1 set of unitary $F$-symbols per case. See appendix E for more info.

| Fusion Algebra | Number of solutions per type of equations (3 particles) per set of equivalent $F$-symbols | | | | |
|---|---|---|---|---|---|
| | Planar Hexagon | Circle | Lollipop Trijunction | Full Lollipop | Lollipop but non-planar |
| $\mathbb{Z}_2$ | $2$ | $2^2$ | $2$ | $2^3$ | $0$ |
| $\mathbb{Z}_3$ | $3$ | $3^3$ | $3^2$ | $3^5$ | $\left(\frac{2}{3}\right)3^5$ |
| | $0$ | $3^3$ | $3^2$ | $3^5$ | $3^5$ |
| $\mathbb{Z}_2 \times \mathbb{Z}_2$ | $2^3$ | $2^7$ | $2^5$ | $2^{13}$ | $\left(\frac{3}{4}\right)2^{13}$ |
| | $0$ | $2^7$ | $2^5$ | $2^{13}$ | $2^{13}$ |
| $\mathbb{Z}_4$ | $2^2$ | $2^8$ | $2^6$ | $2^{14}$ | $\left(\frac{15}{4}\right)2^{14}$ |
| | $0$ | $2^8$ | $2^6$ | $2^{14}$ | $2^{14}$ |

Table 3: Number of gauge inequivalent solutions to the consistency equations for various Abelian anyon models. Here we say two sets of $F$-symbols are equivalent if they both have solvable planar hexagon equations or not. We chose to do this because, within each equivalence class, all members give rise to identical rows.

# 7 Θ-graph yields effective planar anyon models

The Θ-graph shown in Figure 20a) has two independent loops and two essential vertices of degree three.

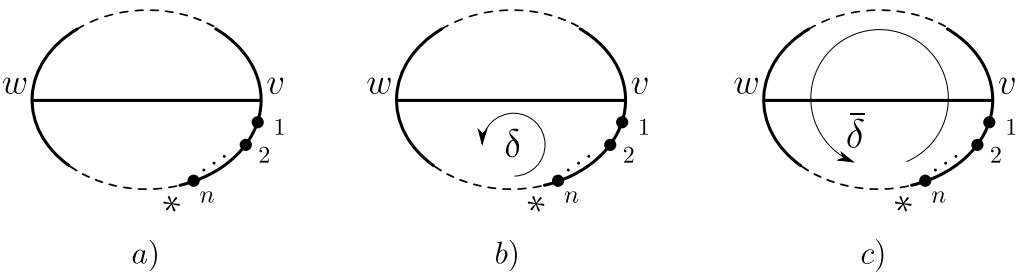

Figure 20: a) The Θ-graph $\Gamma_\Theta$ together with the underlying choice of the rooted spanning tree (solid lines) with the root $*$ and the initial position of anyons. b) and c) The choice of the spanning tree uniquely defines the circular moves $\delta$ and $\bar\delta$ of $B_n(\Gamma_\Theta)$.

Using the universal generators of graph braid groups from [6] (which are also described in Appendix A), we have that $B_3(\Gamma_\Theta)$ is generated by the respective simple braids at vertices $v$ and $w$

$$\sigma_1^{v;(1,2)}, \quad \sigma_2^{v;(1,1,2)}, \quad \sigma_2^{v;(2,1,2)}, \quad \sigma_1^{w;(1,2)}, \quad \sigma_2^{w;(1,1,2)}, \quad \sigma_2^{w;(2,1,2)}$$

and the two circular moves $\delta$ and $\bar\delta$. As explained in Appendix A, all the above generators are defined relative to a choice of the spanning tree of the graph Θ which is shown in Figure 20. However, there are many relations between these generators which allow one to present the group $B_3(\Gamma_\Theta)$ using only three independent generators $\sigma_1^{v;(1,2)}$, $\delta$ and $\bar\delta$ (in fact, the same holds for $B_n(\Gamma_\Theta)$ with any $n \geq 2$ [6]). What is more, by taking the quotient of $B_n(\Gamma_\Theta)$ which identifies all the circular moves with each other, the graph braid group $B_n(\Gamma_\Theta)$ becomes the standard Artin braid group describing anyons in the plane. In the following, we will look into these relations in detail and study their consequences for the graph anyon model on the Θ-graph. In particular, we will show that by assuming that the circular moves $\delta$ and $\bar\delta$ on the Θ-graph are represented by the same $D$-symbols, the relations between the generators of $B_3(\Gamma_\Theta)$ imply

$$P_{ed}^{bac} = Q_{ed}^{bac} = R_e^{ba}, \tag{39}$$

$$\tilde{P}_{ed}^{bac} = \tilde{Q}_{ed}^{bac} = \tilde{R}_e^{ba}, \tag{40}$$

and

$$R_e^{ba} = \tilde{R}_e^{ba}, \tag{41}$$

where the symbols in (39) refer to the simple exchanges at the vertex $v$ and the symbols in (40) refer to the simple exchanges at the vertex $w$. By Theorem 1 in [6] (and Proposition 5 therein), our results apply not only to the Θ- graph, but also to the more general family of triconnected graphs.

Let us start with Equalities (39). These equalities follow immediately from the lollipop relations for the lollipop subgraphs $\Gamma_{L,v}$ and $\Gamma_{\bar{L},v}$ from Figure 21a) and c).

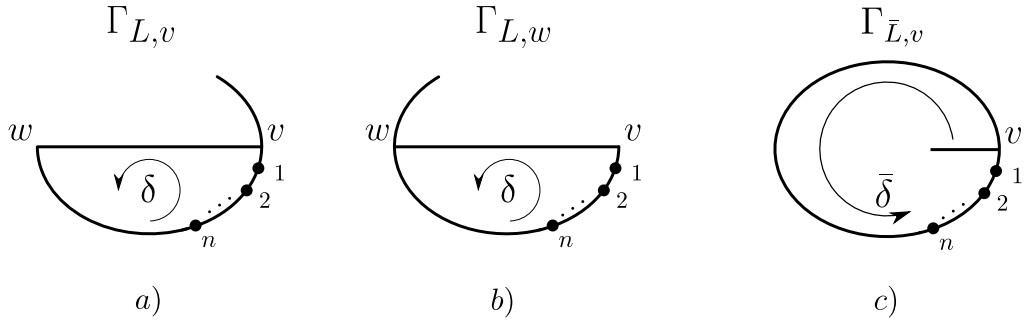

Figure 21: The relevant three different embeddings of the lollipop graph into the Θ-graph.

To see this, apply the diagram from Figure 16 to the respective lollipop relations

$$\sigma_2^{v;(1,1,2)} \delta = \delta \sigma_1^{v;(1,2)}, \quad \sigma_2^{v;(2,1,2)} \bar\delta = \bar\delta \sigma_1^{v;(1,2)}.$$

The first diagram yields $P_{ed}^{bac} = R_e^{ba}$ and the second diagram yields $Q_{ed}^{bac} = R_e^{ba}$, exactly as we derived Equation (34). Similarly, the lollipop relation for the subgraph $\Gamma_{L,w}$ from Figure 21b) gives

$$\sigma_2^{w;(1,1,2)} \delta = \delta \, \sigma_1^{w;(1,2)},$$

thus $\tilde{P}_{ed}^{bac} = \tilde{R}_e^{ba}$.

The derivation of the remaining equalities $R_e^{ba} = \tilde{R}_e^{ba}$ and $\tilde{Q}_{ed}^{bac} = \tilde{R}_e^{ba}$ is considerably more complicated and technical. Importantly, it requires considering the anyon worldlines as world-ribbons and introducing ribbon half-twists. Due to the technical and complicated nature of the proof, we postpone it to Appendix D where we also describe the world-ribbon half-twists on graphs in more detail.

To summarise, we have shown that on the $\Theta$-graph, any graph-braided anyon model is equivalent to the planar anyon model if all the circular moves $\delta$ are represented by the same $D$-symbol. This can be viewed as a mathematical justification for translating results known from the anyon theory in 2$D$ to the network-based setting. For instance, it is known that the Majorana zero modes which were initially proposed in two-dimensional FQHE systems [50, 33], and later proposed in one-dimensional networks [57, 47], can host the same exchange statistics in both settings (see [5, 30, 17] for explicit models for the Majorana zero mode exchange on the tri-junction). However, our approach here is different from the previous work, because it is independent of the microscopic model.

# 8    Consequences for the quantum circuit depth using topological quantum gates

In the standard paradigm of topological quantum computing schemes, the quantum gates acting on a finite set of qudits come from the unitary matrices $\rho(\sigma_i) \in U(d)$. The representation $\rho$ depends on the anyon model at hand and on the chosen topological Hilbert space $\mathcal{H}_{top}$ which is also associated with the particular way of encoding qudits in $\mathcal{H}_{top}$. It is well-known that a minimal requirement to realise a universal quantum computer is to have i) a set of universal single-qudit gates and ii) at least one entangling two-qudit gate. More formally, for a finite set of single qudit gates $\mathcal{S} \subset U(d)$ we denote the group generated by the matrices from $\mathcal{S}$ by $\langle \mathcal{S} \rangle$. The elements of the group $\langle \mathcal{S} \rangle$ are all the possible unitary matrices obtained by sequentially composing gates from $\mathcal{S}$. The set of gates $\mathcal{S}$ is universal if and only if all the unitary matrices from $\langle \mathcal{S} \rangle$ fill in the group $U(d)$ densely. In other words, any matrix $U \in U(d)$ can be approximated by a sequence of gates from a universal set $\mathcal{S}$ with arbitrary precision $\epsilon$. However, the circuit depth, i.e. the length of the sequence of gates necessary to approximate (compile) a given $U$ increases when the required precision grows, see the celebrated Solovay-Kitaev algorithm [21, 1]. In this section, we argue that topological quantum gates coming from the graph braided anyon models can reduce the circuit depth when compared to quantum gates coming from the 2D braided anyon models.

In short, the reason why graph braided anyon models can lead to lower-depth quantum circuits is that the simple braids realised at different junctions of the graph can be topologically inequivalent, i.e. cannot be transformed one into another via isotopies of their corresponding world-lines. This allows us to associate different sets of the $R$-, $P$- $Q$-symbols (and their higher-particle number counterparts) with the junctions which yield topologically inequivalent braids. Such a phenomenon occurs, for instance, in the $H$-graph as discussed in Section 3.3. Another example of a network architecture where this phenomenon occurs is the stadium graph or, more generally, a biconnected modular network that consists of a chain of triconnected modules that are connected by bridges consisting of two edges [6, 48], see Figure 22. This has also been pointed out in the case of Abelian quantum statistics on graphs in [32].

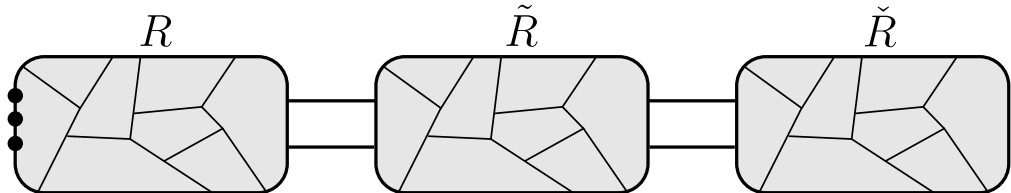

Figure 22: A schematic representation of a modular biconnected network composed of three triconnected modules (represented by grey boxes). According to the general prescription presented in Appendix A, the base configuration of anyons is on an external edge of the leftmost module. The simple graph braids realised in different modules are topologically independent. By the results of Section 7, the braiding within each module is effectively governed by an independent set of $R$-symbols which constitute a solution to the hexagon equations from the corresponding 2D anyon theory. Thus, such a network architecture allows for simultaneous coexistence and mixing of different sets of $R$-symbols.

For concreteness, let us focus on the stadium graph The stadium graph and its generalisations (involving more vertical ribs) have been proposed as a typical setup for network-based topological quantum computer [5]. As shown in Figure 23, there are two ways of embedding a $\Theta$-graph into the stadium graph where the embedded $\Theta$-graph contains either the opposite pairs of essential vertices $v$ and $v'$ or $w$ and $w'$.

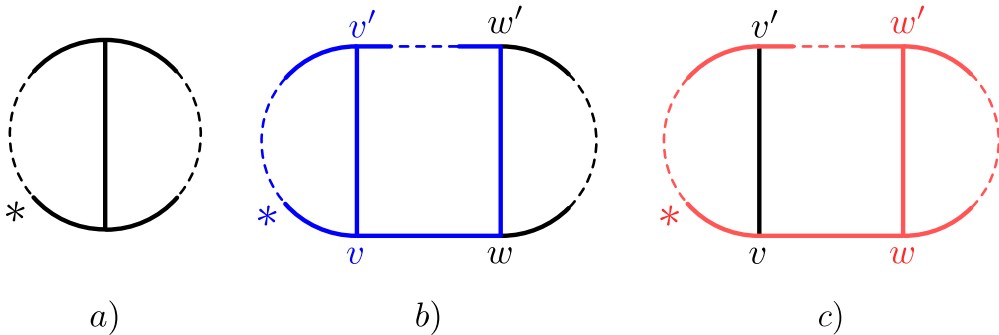

Figure 23: a) The $\Theta$-graph together with the choice of the rooted spanning tree. b) and c) Two different embeddings of the $\Theta$-graph into the stadium graph (marked by red and blue) which by the results of Section 7 imply that the simple braids at $v$ and $v'$ are equivalent to the braiding in 2D and are represented by the same set of $R$-symbols. Similarly, the simple braids at $w$ and $w'$ are represented by another set of $R$-symbols coming from the 2D anyon theory.

Thus, by the results of Section 7 any graph braided anyon model on the stadium graph will admit two independent sets of the planar $R$-symbols. Namely, the simple braids at $v$ or $v'$ will be represented by one set of $R$-symbols coming from the 2D braided anyon model and the simple braids at $w$ or $w'$ will be represented by another, *a priori* different, set of $R$-symbols coming from the 2D braided anyon model. Let us reiterate the crucial fact that, the simple braids at $w$ and $w'$ are topologically independent from the simple braids at $v$ or $v'$, thus it is *a priori* possible to represent them by different sets of $R$-symbols. This in turn can increase the number of the available topological single-qudit quantum gates which constitute the set $\mathcal{S}$. Having access to a larger set of topological gates $\mathcal{S}$ gives one more flexibility when compiling the target quantum algorithm and thus increases the efficiency of the given quantum circuit by lowering the circuit depth.

The potential advantage of using the stadium graph architecture and its generalisations is also evident when considering certain non-universal anyon models. For instance, consider the Tambara-Yamagami model with $G = \mathbb{Z}_2 \times \mathbb{Z}_2$ [70]. Denote by $\sigma$ the anyon with the property

$$\sigma \times \sigma = \bigoplus_{g \in G} g, \quad G = \mathbb{Z}_2 \times \mathbb{Z}_2.$$

The topological Hilbert space of the three $\sigma$-anyons of the total charge $\sigma$ is given by

$$\mathcal{H}_{top} = \mathrm{Span}\{|\sigma, \sigma \to g\rangle |g, \sigma \to \sigma\rangle : \quad g \in G\} \cong \mathbb{C}^4.$$

In such a setting, the braiding operators are single-ququart topological quantum gates. In the stadium-graph geometry, the simple braids $\sigma_1^{v;(2,1)}$ and $\sigma_1^{v';(2,1)}$ are represented by the same braiding exchange operator $R$ which is a diagonal $4 \times 4$ matrix whose diagonal entries are the $R$-symbols $R_g^{\sigma\sigma}$, $g \in G$, which are solutions to the hexagon equations for the anyon model in $2D$. The simple braids $\sigma_1^{w;(2,1)}$ and $\sigma_1^{w';(2,1)}$ are represented by the matrix $\tilde{R}$ constructed from another set of solutions to the hexagon equations for the anyon model in $2D$. For concreteness, let us choose the following solutions to the planar hexagon equations

$$
R = \begin{pmatrix} e^{i\pi/4} & 0 & 0 & 0 \\ 0 & e^{i3\pi/4} & 0 & 0 \\ 0 & 0 & e^{i3\pi/4} & 0 \\ 0 & 0 & 0 & e^{-i3\pi/4} \end{pmatrix}, \quad \tilde{R} = \begin{pmatrix} -i & 0 & 0 & 0 \\ 0 & 1 & 0 & 0 \\ 0 & 0 & -1 & 0 \\ 0 & 0 & 0 & -i \end{pmatrix}.
$$

The relevant $\sigma_2$-braids $\sigma_2^{v;(112)}, \sigma_2^{v;(212)}, \sigma_2^{v';(212)}$ and $\sigma_2^{v';(212)}$ are represented by the matrix $B = \left(F_\sigma^{\sigma\sigma}\right)^\dagger R F_\sigma^{\sigma\sigma}$ while $\sigma_2^{w;(112)}, \sigma_2^{w;(212)}, \sigma_2^{w';(212)}$ and $\sigma_2^{w';(212)}$ are represented by the matrix $\tilde{B} = \left(F_\sigma^{\sigma\sigma}\right)^\dagger \tilde{R} F_\sigma^{\sigma\sigma}$. Here, the relevant $F$-matrix reads

$$
F_\sigma^{\sigma\sigma} = \frac{1}{2} \begin{pmatrix} -1 & -1 & -1 & -1 \\ -1 & 1 & -1 & 1 \\ -1 & -1 & 1 & 1 \\ -1 & 1 & 1 & -1 \end{pmatrix}.
$$

Let us next consider the (finite) groups generated by the sets $\mathcal{S} := \{R, B\}$ and $\tilde{\mathcal{S}} := \{\tilde{R}, \tilde{B}\}$. We will focus on how the resulting quantum gates act on a single ququart which means that we neglect the global phase factors. In other words, we look at the resulting groups projectively by projecting every element to the group $PSU(4)$. It can be verified in a straightforward way that the groups $\langle \mathcal{S} \rangle \subset PSU(4)$ and $\langle \tilde{\mathcal{S}} \rangle \subset PSU(4)$ are different and both are isomorphic to $S_4$, the permutation group of four elements

$$
\langle \mathcal{S} \rangle \neq \langle \tilde{\mathcal{S}} \rangle, \quad \langle \mathcal{S} \rangle \cong \langle \tilde{\mathcal{S}} \rangle \cong S_4 \subset PSU(4).
$$

Furthermore, by considering combinations of exchanges on the two $\Theta$-subgraphs of the stadium graph we can generate the group $\langle \mathcal{S} \cup \tilde{\mathcal{S}} \rangle \subset PSU(4)$ which is a finite group of rank 96 and strictly contains the groups $\langle \mathcal{S} \rangle$ and $\langle \tilde{\mathcal{S}} \rangle$. Thus, by combining braids at different junctions of the stadium graph we are able to generate a bigger (although still finite) subgroup of $PSU(4)$ which means that we have increased the computational power when compared to the standard $2D$ setting.

The crucial feature of the above calculation was that the subgroups of $PSU(d)$ generated by the braiding exchange operators $R$, $B$ and $\tilde{R}$, $\tilde{B}$ were different. A necessary condition for this to happen is that the (unitary) braiding exchange operator $R$ is different than $e^{i\phi}\tilde{R}$ for every $\phi \in [0, 2\pi]$. Finding such operators $R$ and $\tilde{R}$ is not possible for every model. For instance, in the Ising model (Tambara-Yamagami with $G = \mathbb{Z}_2$) all the braiding exchange operators corresponding to different hexagon solutions are related via multiplication by such a global phase factor. The Tambara-Yamagami model with $G = \mathbb{Z}_2 \times \mathbb{Z}_2$ is the lowest rank fusion category we could find where the different braiding exchange operators are not related by a global phase factor.

# 9 Conclusions

In this work, we have developed a universal framework for studying topological quantum systems hosting anyonic excitations on quantum wire networks. Using the results of this work, any 2D anyon theory (understood as a fixed set of fusion rules and $F$-symbols) can be readily translated to a network setting. Our framework assumes the same basis of fusion states as on the plane (described in Section 2). It is not obvious that this is a full description of the states on a graph. For instance, even on a 2D torus, a description of the topological Hilbert space requires labels associated with the nontrivial loops around the torus. One may expect such extra labels to appear also for graphs with loops or perhaps even for graphs without loops. However, we have decided to work

with our choice of the fusion basis as a starting point. This has already led to interesting classes of solutions – we have found nontrivial solutions for models that do not admit braiding in 2D as well as new classes of solutions for other models. We have also shown that the character of Abelian and non-Abelian exchange depends strongly on the structure of the given network. In particular, the possible braiding exchange operators that arise from our framework applied to simple junctions or tree graphs are less constrained than the ones that arise in more complex networks. At the far end of this spectrum of possibilities are triconnected networks, for which the resulting exchange operators are equivalent to the 2D anyon theory. Hence, for triconnected networks we recover coherence as well as rigidity (number of solutions modulo gauge is finite). At the other end, we have the trijunction, where we find the most freedom in the braiding exchange operators as there exist continuous families of solutions to our polygon equations. Coherence remains an open question. However, we conjecture that on a trijunction the theory is coherent for $N > 4$ particles and we discuss evidence for this. For biconnected and one-connected networks we have found numerous examples of new Abelian and non-Abelian exchange statistics that do not exist in 2D. We have argued that physically realising some of these possibilities could lead to proposals for topological quantum computers where quantum algorithms would be compiled more efficiently. A natural next step would be to look for physical models which could host the Abelian or non-Abelian quantum statistics that do not exist in 2D anyon models. One possible way of finding systems that host the new exchange statistics on networks would be through certain generalisations of discrete gauge theories. This approach has been employed in [59].

Another direction posed by our work is an examination of the space of states on a graph. In particular, we chose to consider each edge of the graph as hosting particles with the same fusion rules & $F$-symbols as anyons in the plane. In doing so, we found several similarities. One example of this is the absence of a solution to the graph hexagon equations for parafermion zero modes, which in this context are associated with the non-Abelian anyon in $TY(\mathbb{Z}_N)$. For example the fusion rules for parafermions in this context can be found in the Supplementary Material of [15]. In this sense, parafermionic zero modes can be seen as a generalisation of Majorana zero modes, which occur for the particular choice $N = 2$. Crucially, when $N \neq 2^p$ there is no solution to both the planar hexagon equations [67] and our graph hexagon equations on a star graph, see Section F. However, parafermionic zero modes have been proposed to exist on one-dimensional wires [27, 41, 42]. In addition to this, it has also been proposed that parafermion zero modes can exist on interfaces between FQHE states [16] which, when arranged into certain geometries, may allow the zero modes to be braided [15]. Therefore, it seems surprising that we find no solution to the graph hexagon equations on a star graph for parafermions (although we do find solutions on a circle graph). It seems possible that considering each edge of the graph as a non-trivial boundary of a 2D topological order may be the most comprehensive description.

## Acknowledgements

The authors would like to thank Alex Bullivant for bringing reference [31], to our attention and for useful discussions about the connection between anyons on the circle graph and the tube category construction.

**Funding information**  J.K.S. and G.V. acknowledge financial support from Science Foundation Ireland through Principal Investigator Awards 12/IA/1697 and 16/IA/4524. A.C. was supported through IRC Government of Ireland Postgraduate Scholarship GOIPG/2016/722. The authors acknowledge financial support from the Heilbronn Institute for Mathematical Research within the Focused Research Grants programme, which facilitated a workshop on this topic.

## A  Generators of graph braid groups for general graphs

In this section we will discuss some necessary facts about the generators of graph braid groups. In particular, we recap the systematic construction of the generating set of graphs braids which works for any planar graph [6]. We subsequently use this general procedure to study some small canonical graphs from the main text of the paper. The simple braids are specific generators of the graph braid group of a rooted star graph. In general, graph braids are created via sequences of moves transporting anyons to certain edges of the graph and returning them to their original configuration. It will be convenient to introduce a separate notation for a move where

a single anyon is transported from one location to another. The relevant moves are called the $\beta$-moves – they transport anyons from the base configuration on the edge containing the root to another leaf of the star graph. A $\beta$-move, $\beta_x$, is decorated by a subscript $1 \leq x \leq E$ ($E+1$ is the number of legs of the star graph) which denotes the index of the leaf of the star graph where the anyon which is the closest to the junction is transported from the base configuration. Consequently, $\beta_x^{-1}$ transports an anyon from leaf with label $x$ back to the base configuration (see Figure 24).

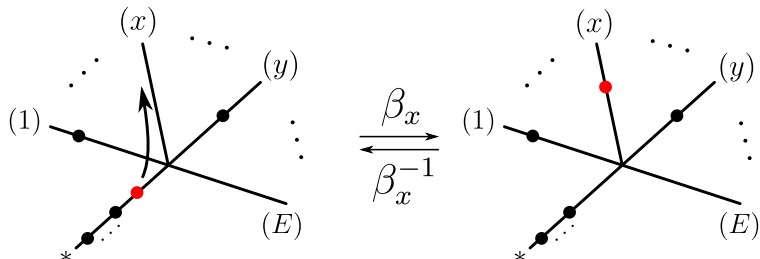

Figure 24: The auxiliary move $\beta_x$ and it's inverse. The root is decorated by $*$.

Thus, an exchange of two anyons which involves leaves $x$ and $y$ with $1 \leq x < y \leq E$ is given by the commutator of the corresponding $\beta$-moves

$$\left[\beta_x, \beta_y\right] := \beta_x \beta_y \beta_x^{-1} \beta_y^{-1}.$$

We will call this a simple exchange and denote by $\sigma_1^{(x,y)}$, see Figure 25.

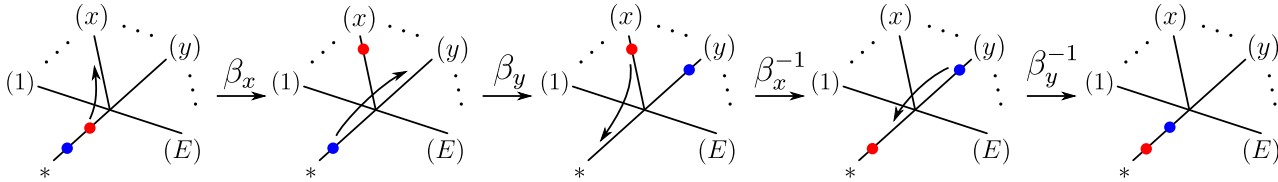

Figure 25: The simple exchange $\sigma_1^{(x,y)} = \left[\beta_x, \beta_y\right]$. The root is decorated by $*$.

In order to realise a simple exchange of anyons $i$ and $i+1$ one needs to first distribute the anyons 1 through $i-1$ on the leaves of the star graph and move them back in the same order after the exchange. This is realised by the following sequence of $\beta$-moves

$$\sigma_i^{(x(1),...,x(i+1))} = \beta_{x(1)} \ldots \beta_{x(i-1)} \left[\beta_{x(i)}, \beta_{x(i+1)}\right] \beta_{x(i-1)}^{-1} \ldots \beta_{x(1)}^{-1}, \tag{42}$$

where $x(k)$ is the leaf visited by $k$th anyon (with anyon 1 being the closest to the junction). Analogous $\beta$-moves can be realised on a rooted tree graph $(T,*)$. Namely, let $v \in T$ be an essential vertex (i.e. a vertex of degree $d(v) \geq 2$). One can embed a rooted star graph of the order $d(v)$, $(S, *_S)$, into a neighbourhood of $v$ in $T$

$$\iota_v : (S, *_S) \to (T, *),$$

so that the essential vertices are mapped onto each other and $\iota_v(*_S)$ lies on the unique path connecting $v$ with $* \in T$, as shown in Figure 26a. Then, the move $\beta_{v,x}$ is defined as the composition $\beta_{v,x} := \beta_0 \beta_x$, where $\beta_0$ transports an anyon from the base configuration on the edge containing the root $* \in T$ to $\iota_v(*_S)$ and $\beta_x$ transports the same anyon from $\iota_v(*_S)$ to the leaf $a$ of the embedded star graph, see Figure 26b.

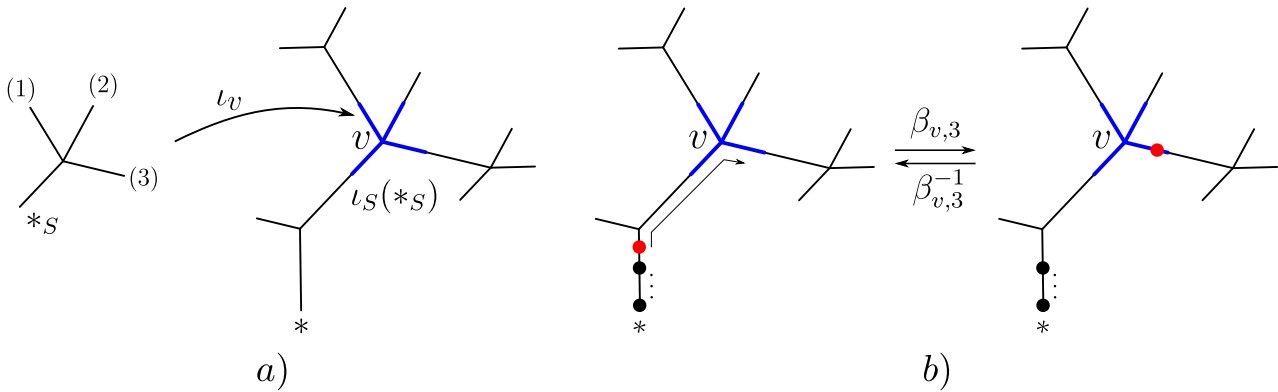

Figure 26: a) Embedding a star graph into a neighbourhood of the essential vertex $v$ of a tree graph. b) The resulting $\beta_{v,3}$-move.

Thus, we analogously define the simple braids associated with the vertex $v$ as

$$\sigma_i^{v;(x(1),\ldots,x(i),x(i+1))} = \beta_{v,x(1)}\cdots\beta_{v,x(i-1)}\left[\beta_{v,x(i)},\beta_{v,x(i+1)}\right]\beta_{v,x(i-1)}^{-1}\cdots\beta_{v,x(1)}^{-1}.$$

Note that any simple graph braid in $B_n(\Gamma)$ can be embedded into $B_{n+1}(\Gamma)$ via conjugation by a $\beta_{u,y}$-move where $u$ is an essential vertex of $\Gamma$ and $1 \le y \le d(v)-1$.

$$B_n(\Gamma) \ni \sigma_i^{v;(x(1),\ldots,x(i+1))} \mapsto \beta_{u,y}\,\sigma_i^{v;(x(1),\ldots,x(i+1))}\,\beta_{u,y}^{-1} \in B_{n+1}(\Gamma).$$

In particular, if $u = v$, then we have

$$\beta_{v,y}\,\sigma_i^{v;(x(1),\ldots,x(i),x(i+1))}\,\beta_{v,y}^{-1} = \sigma_{i+1}^{v;(y,x(1),\ldots,x(i),x(i+1))}.$$

We use this fact several times throughout the paper, see for instance Equation (23).

For a planar graph $\Gamma$ which is not a tree graph, i.e. $\Gamma$ which contains loops, some new generators appear. The new generators correspond to different ways of embedding the circle graph and its corresponding $\delta$-move which has been introduced in Section 4. Let us next review a systematic way of counting the relevant embeddings of the circle graph into any planar graph $\Gamma$ which is not a tree. We first fix the planar embedding of $\Gamma$, $\iota_\Gamma : \Gamma \to \mathbb{R}^2$. Next, we choose a spanning tree $T \subset \Gamma$ (a tree which contains all the vertices of $\Gamma$) such that every essential vertex of $\Gamma$ is contained in $T$ together with its star-shaped neighbourhood in $\Gamma$ (formally, this may require adding some dummy vertices of order two in the interiors of certain edges of $\Gamma$, a procedure called edge subdivision, for details see [6]). Using such a choice of the spanning tree we build a basis of loops of $\Gamma$ in the following way. The number of loops of $\Gamma$ is equal to the first Betti number of $\Gamma$, $B_1(\Gamma) = |E(\Gamma)| - |V(\Gamma)| + 1$, where $E(\Gamma)$ and $V(\Gamma)$ are the sets of edges and vertices of $\Gamma$ respectively. Edges from $E(\Gamma)$ which do not belong to $T$ are called the deleted edges. The number of deleted edges is equal to $B_1(\Gamma)$ and each $e \in E(\Gamma) - E(T)$ defines a loop in $\Gamma$ in the following way. Let $v$ and $w$ be the end-vertices of $e$. We necessarily have that $v, w \in T$, thus there is a unique path $P(v,w) \subset T$ that connects $v$ with $w$ in $T$. Thus, the union $l_e := e \cup P(v,w)$ is a loop in $\Gamma$ (see Figure 27a). The set $\mathcal{L}_\Gamma := \{l_e | e \in E(\Gamma) - E(T)\}$ forms a generating set of simple loops of $\Gamma$.

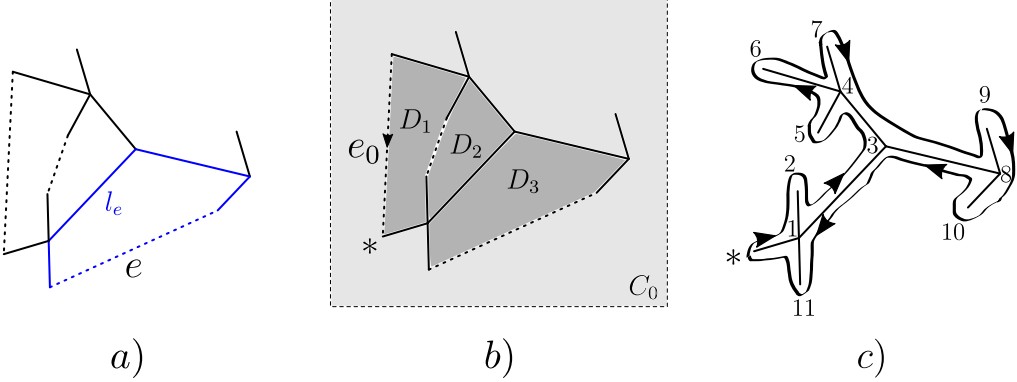

Figure 27: a) A graph with a spanning tree marked by solid lines and the deleted edges marked by dashed lines. The loop $l_e$ is a simple loop corresponding to the deleted edge $e$. b) The planar embedding of $\Gamma$ and its resulting decomposition of the plane into four connected components: the bounded discs $D_1, D_2, D_3$ and the unbounded component $C_0$. A possible choice for the external deleted edge $e_0$ and the root $*$ is shown. c) The linear ordering of the graph's vertices implied by the planar embedding and the choice of the root.

The set $\mathbb{R}^2 - \iota(\Gamma)$ is topologically a disjoint union of a number of connected components. In particular, among the connected components there are disks $D_1, \ldots, D_{B_1(\Gamma)}$ which are enclosed by the loops from $\mathcal{L}_\Gamma$ and one unbounded component which we denote by $C_0$ (see Figure 27b). Let us next choose a deleted edge $e_0$ which is external relative to the embedding $\iota_\Gamma$, i.e. it belongs to the boundary of $C_0$. We define the root $* \in T$ as one of the endpoints of $e_0$. We give the edge $e_0$ an orientation directed from its other endpoint to the root as shown in Figure 27b. Then, $l_{e_0}$ is the oriented loop which supports the circular move $\delta$ from Section 4 whose orientation is induced by the orientation of $e_0$. To every other deleted edge $e$ we associate an independent circular move $\delta_e$ in the following way. We order the vertices of $\Gamma$ by drawing a ribbon around $T$ in a clockwise direction, starting at the root and labelling all the visited edges by consecutive integers as shown in Figure 27c. This way, each edge $e \in \Gamma$ acquires an orientation which points from the vertex labelled by the higher number (called the initial vertex $\iota(e)$) to the vertex labelled by the lower number (called the terminal vertex $\tau(e)$, $\iota(e) > \tau(e)$). In particular, this holds for every deleted edge $e$ and induces an orientation of the associated loop $l_e$. We are now ready to define the circular move $\delta_e$ associated with a deleted edge $e$. There are two cases (see Figure 28 for examples).

1. If $\iota(e) < \iota(e_0)$, then $\delta_e$ i) takes an anyon from the base configuration at the edge of $T$ containing the root $*$ and moves it to the vertex $\tau(e)$ along the unique path $P(*, \iota(e)) \subset T$, ii) moves the anyon from $\tau(e)$ to $\iota(e)$ along $e$, iii) moves the anyon from $\iota(e)$ to $\iota(e_0)$ along the unique path $P(\iota(e), \iota(e_0)) \subset T$, iv) moves the anyon from $\iota(e_0)$ to $\tau(e_0) = *$ along $e_0$.

2. $\iota(e) > \iota(e_0)$, then $\delta_e$ i) takes an anyon from the base configuration at the edge of $T$ containing the root $*$ and moves it to the vertex $\iota(e)$ along the unique path $P(*, \iota(e)) \subset T$, ii) moves the anyon from $\iota(e)$ to $\tau(e)$ along $e$, iii) moves the anyon from $\tau(e)$ to $\iota(e_0)$ along the unique path $P(\tau(e), \iota(e_0)) \subset T$, iv) moves the anyon from $\iota(e_0)$ to $\tau(e_0) = *$ along $e_0$.

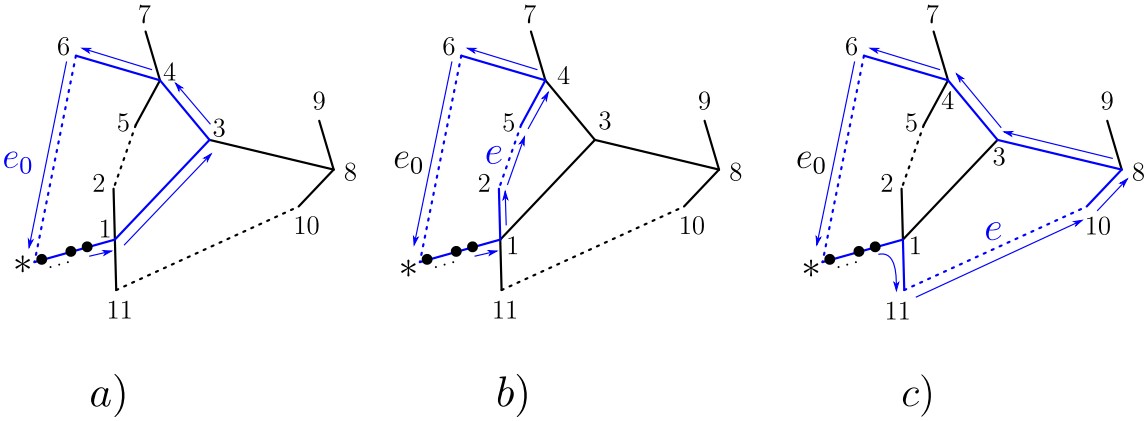

a)          b)          c)

Figure 28: Examples of $\delta$-moves for a graph with multiple loops. a) The $\delta$-move associated with the edge $e_0$ which is incident to the root $*$. b) The $\delta_e$-move when $\iota(e) < \iota(e_0)$. c) The $\delta_e$-move when $\iota(e) > \iota(e_0)$.

Summing up, the graph braid group of $\Gamma$ is generated by the simple braids

$$\sigma_i^{v;(x(1),\dots,x(i),x(i+1))}, \quad 1 \leq x(j) \leq d(v), \quad x(i) < x(i+1), \quad d(v) > 2,$$

and the circular moves

$$\delta, \quad \delta_e, \quad e \in E(\Gamma) - E(T) - \{e_0\}.$$

# B   Coherence for graph anyon models on the circle

In this section we will discuss the graph anyon model on a circle for $N > 3$ particles. In particular, we show that such a model has the coherence property mentioned in Section 2. Our aim is to show that the consistency equations for four particles are already guaranteed by the solution of the circle hexagon equation for three particles given in Equation (29). In other words, no new constraints for the $D$ symbols appear for $N > 3$. This is in contrast to the simple braids at junctions, in which, the addition of new particles introduces new, topologically inequivalent generators (up to when the number of particles is at least one greater than the valence of the junction), as discussed in Section 3.1 and Appendix C.

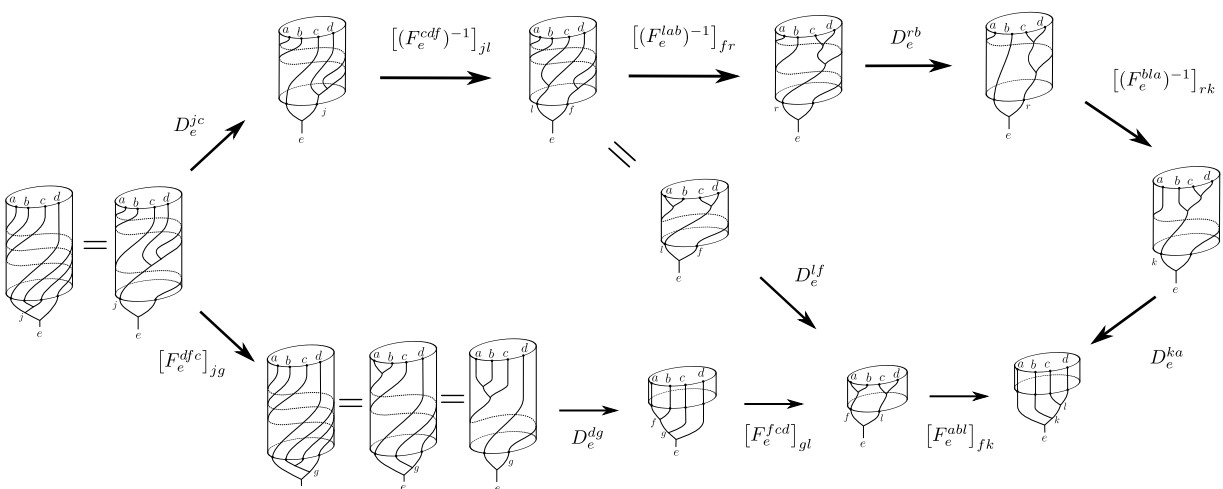

Figure 29: A four-particle consistency diagram for particles cycling around the circle graph. In the bottom left we display the fusion commuting with graph braiding states, which are related by sliding a fusion vertex through a $\delta$-braid. The middle diagonal path is what informs us this is will not lead to new constraints on the $D$- symbols, as it can already be tessellated by the hexagon diagrams.

We start by considering the action of the $D$- symbols given in Equation (11). The $D$- symbol depends on the topological charge of the particle cycling around the circle and the total topological charge of the remaining

particles. See, for instance the action of $D_d^{fa}$ in the upper path of Figure 12. Similarly, if there are $N$ particles on the circle, then anyon $f$ is the total charge of a set of $N-1$ anyons and the action of the $D_d^{fa}$- symbol only depends on the total topological charge of the $N-1$ particle group and $a$.

To construct a consistency equation, we consider a diagram where fusion commutes with braiding of four particles and then look at all the possible ways to resolve the braids. This strategy has been employed to derive the $D$-hexagons in Figure 12 (see Figure 3 for a similar treatment of the trijunction). Equivalently, we can view this methodology as expressing a braid involving a composite anyon in terms of the composition of simple braids of its constituents. So, explicitly for four particles, we would like to impose the following relations on our anyon model (we use the labelling convention from Equation (59));

$$\delta(k,a)\,\delta(r,b)\,\delta(j,c) = \delta(l,f)\,\delta(j,c) = \delta(d,g),\tag{43}$$

where the right entry of $\delta$ labels the topological charge of the particle cycling around the graph and the left entry labels the total topological charge of the remaining particles. This is analogous to Equation (22) for a junction. The resulting diagram is shown in Figure 29. Note that in the bottom left of Figure 29 we have equated three states using the fact that fusion commutes with braiding several times. From these states, we can construct consistency equations, stemming from applying appropriate $D$-moves and $F$-moves to make the diagram commutative. In other words, every loop in the diagram in Figure 29 represents a consistency relation. However, the crucial observation is that the large outer loop (decagon diagram) is a composition of two smaller loops, each containing six states (hexagon diagrams). Thus, satisfying the consistency equations corresponding to the two inner hexagonal diagrams will imply that the consistency equations corresponding to the outer decagonal diagram will be satisfied as well. Let us next take a closer look at the leftmost hexagon diagram. Note first that in the resulting equations the constituents of the composite anyon $f = a \times b$ will not appear, as the particles $a$ and $b$ are always connected by a common fusion channel. Thus, this diagram is effectively a three-particle diagram involving particles $f$, $c$ and $d$. Comparing the two paths starting from the leftmost state and ending at the pairwise associated state at the bottom of the diagonal path we obtain the following consistency equation

$$\sum_g \left[(F_e^{dfc})\right]_{jg} D_e^{dg} \left[F_e^{fcd}\right]_{gl} = D_e^{jc} \left[(F_e^{cdf})^{-1}\right] D_e^{lf}.\tag{44}$$

Importantly, Equation (44) becomes identical to the $D$-hexagon from Equation (29) after appropriate relabelling of anyons. Similarly, the rightmost sub-hexagon diagram leads to effective three-particle equations involving anyons $a$, $b$ and $l = c \times d$

$$\sum_f \left[F_e^{lab}\right]_{rf} D_e^{lf} \left[F^{abl}\right]_{fk} = D_e^{rb} \left[(F_e^{bla})^{-1}\right]_{rk} D_e^{ka},\tag{45}$$

which can also be identified as $D$-hexagon equations after relabelling.

To summarise, we started with the four particle fusion commuting with graph braiding states which could have led to new constraints on the $D$- symbols, however, we were able to recognise that this diagram was readily satisfied just by the three particle hexagon diagrams. Therefore, when solving these equations for a given fusion model this will add no new constraints (see Section E for details on solving these equations). Inducting over the number of particles, one can see that similar reasoning shows that we can always do this on a circle, and as we add more and more particles this will lead to no new constraints.

## C  Towards coherence for anyon models on general graphs

In this section, we will discuss how our graph anyon models change when increasing particle number. In particular, we will look into the coherence property of these anyon models – is there a particle number $N_0$ above which no new consistency relations appear?

Firstly, let us discuss what coherence is and why it is *a priori* not clear that our anyon models have this property. The coherence of anyon theory in the plane has been discussed e.g. in [39, 51]. Let us start with fusion (monoidal) coherence. In order to formulate fusion coherence for $N$ anyons, one considers a diagram whose nodes are all the possible $N$-anyon fusion trees ($N$ anyons at the top of the tree fusing to one total charge) and the edges are the $F$-moves between the fusion trees. The coherence theorem for $F$-moves (fusion coherence)

states that any sequence of $F$-moves between a fixed pair of nodes of such a diagram results in the same morphism of the corresponding topological Hilbert spaces, provided that the pentagon equations are satisfied (together with some trivially satisfied square diagrams). In other words, solving the consistency equations for the $F$-moves in the case of $N = 4$ anyons implies that the entire theory is consistent for any $N > 4$. Similarly, the braided coherence theorem states that any sequence of morphisms which involves $F$- and $R$-moves between two fixed states results with the same morphism of the corresponding topological Hilbert spaces provided that the pentagon and hexagon equations are satisfied. The proof of this theorem relies on more abstract results in category theory, known as the Mac Lane monoidal (fusion) coherence theorem [49], and the braided coherence theorem [60].

One of the first things to observe is that we *do* have the fusion coherence, since, as discussed in [18] and in Section 3, we do not modify the fusion rules and we use the same $F$-symbols as the planar anyon models. However, the braiding structure on networks is different. When considering higher numbers of anyons, more and more topologically inequivalent generators of the graph braid group are introduced (see Section 3.1, further details of which can be found in [6]). In order to faithfully represent the new generators of the graph braid group, we introduce new symbols. For example, increasing the number of particles from three to four on a trijunction introduces the new generators in Equation (16). Therefore, new consistency equations (expressing the compatibility of fusion and braiding) are introduced; (18), (19), (21) and (24). As we explain in Section 7, if the graph is sufficiently highly connected, we recover planar braiding, and therefore all the aforementioned coherence theorems known from the planar anyon theory. However, all the biconnected and one-connected graphs require separate treatment. For concreteness, we will next focus on a trijunction. The entire following discussion extends *mutatis mutandis* to arbitrary graphs.

As we explained in Section 3, our aim is to build an anyon theory which faithfully represents topologically inequivalent graph braid group generators. This is done by assigning different symbols to topologically inequivalent generators. If we solve all of the $N$-particle consistency equations, increasing the number of particles to $N + 1$ introduces new generators and, in principle, new relations, which may not be satisfied by solutions to the consistency equations for $N$ particles. As we conjecture below, for every graph $\Gamma$ there exists a certain number $N_0(\Gamma)$ such all the consistency relations for any $N > N_0(\Gamma)$ are readily satisfied by the solutions to the $N = N_0(\Gamma)$ consistency relations. This is what we call the *graph braided coherence conjecture*. For the trijunction, $\Gamma = \Gamma_T$, we conjecture that $N_0(\Gamma_T) = 5$. For the simplified anyon models defined in Section 3.2 (all the symbols having at most four labels), we conjecture that $N_0(\Gamma_T) = 4$.

What is more, we conjecture what the complete set of consistency relations looks like for any $N$. We distinguish three types of consistency relations for anyon models on the trijunction that form the complete set of relations: the pseudocommutative relations (Equation (47)), the analogues of the $N = 3$ hexagon relations (Equation (50)) and the topological charge conservation relations.

1. **The pseudocommutative relations**. As we explain in Appendix A, a simple graph braid on the trijunction has the form

$$\sigma_i^{(x(1),\ldots,x(i-1),x(i),x(i+1))}, \tag{46}$$

where $x(k) \in \{1, 2\}$ and $x(i) < x(i+1)$. However, such simple braids form an over-complete generating set of the corresponding graph braid group as they are subject to the following pseudocommutative relations [6, 48], for $j - i \geq 2$ and $j > i$

$$\sigma_j^{(x(1),\ldots,x(j+1))}\sigma_i^{(x(1),\ldots,x(i+1))} = \sigma_i^{(x(1),\ldots,x(i+1))}\sigma_j^{(x(1),\ldots,x(i-1),x(i+1),x(i),x(i+2),\ldots,x(j+1))}. \tag{47}$$

An example of such a relation has been considered in Section 3.1 in Equation (20). The pseudocommutative relations allow us to find a minimal generating set of the braid group for a junction of any valence (for junctions other than just trijunctions, one needs to also consider certain pseudobraiding relations) [6, 18], ultimately showing that graph braid groups for particles moving on a single junction are free groups [6]. In the case of the trijunction, the minimal generating set consists of generators of the form

$$\sigma_j^{(2,\ldots,2,1,\ldots,1,1,2)}, \tag{48}$$

where the first string of twos has length $K$ and the second string of ones has length $L$, so that $K + L + 1 = j$. In other words, any simple braid $\sigma_i^{(x(1),\ldots,x(i-1),x(i),x(i+1))}$ can be expressed as a product of the minimal generators

of the form (48) with $j \leq i$. Thus, the braiding exchange operator corresponding to any simple braid can be determined from the braiding exchange operators representing the minimal generators (48) via the corresponding polygon diagrams. Therefore, it is sufficient to assign different symbols to different minimal generators. For $N = 4$, the relevant symbols are $R$, $P$, $Q$ and $B$ (see Section 3.1).

Importantly, any generator of the form (48) is effectively a four-particle exchange process (at most) and as such can be expressed in terms of the $R$-, $P$-, $Q$- or $B$-symbols. To see this, consider the corresponding exchange operator representing the simple braid

$$\sigma_j^{\left(2_{d_1},\ldots,2_{d_K},1_{c_1},\ldots,1_{c_L},1_a,2_b\right)},$$

where the anyons $d_1 \times \cdots \times d_K = f$ and the anyons $c_1 \times \cdots \times c_K = e$ are joined by a common fusion channel and the total charge of the entire system is $i$. In such a basis, the fusion vertices $e$ and $f$ can be pulled through the entire spacetime diagram of the exchange process, so that such a process effectively becomes a four-particle process. Thus, the above simple braid can be represented by $B_{ghi}^{baef}$, where $g = a \times b$ and $h = g \times e$. If $K = 0$ or $L = 0$ this reduces to $P_{gi}^{bae}$ or $Q_{gi}^{baf}$ respectively. Therefore, by the preceding discussion, we conclude that any simple braid (46) can be expressed by the $R$-, $P$-, $Q$- and $B$-symbols. However, the resulting expressions can be quite involved as they require using pseudocommutative relations repeatedly.

2. **Analogues of the hexagons for higher particle number.** Recall that the $P$- and $Q$-hexagons were derived from the following relations expressing the fact that fusion commutes with graph braiding for $N = 3$ anyons on a trijunction [18],

$$\sigma_1^{(1_a,2_c)}\sigma_2^{(2_c,1_a,2_b)} = \sigma_1^{(1_a,2_{b\times c})}, \quad \sigma_2^{(1_b,1_a,2_c)}\sigma_1^{(1_b,2_c)} = \sigma_1^{(1_{a\times b},2_c)}. \tag{49}$$

One can lift the above relation in order to impose the commutativity of fusion and braiding for any $N > 3$ by adding a string of particles from the side of the junction. This has been done for $N = 4$ in Equation (23). In general, if the added particles have the charges $d_1,\ldots,d_M$, we obtain the analogous relations for $N = M + 3$ by appending a sequence $\overline{x} = \left(x(1)_{d_1},\ldots,x(M)_{d_M}\right)$ to each superscript in Equations (49). Here, $x(k) \in \{1, 2\}$ denotes the branch of the trijunction visited by the $k$th anyon (of the charge $d_k$ with anyon $d_1$ being the closest one to the junction). The resulting lifted relations read

$$\sigma_{M+1}^{(\overline{x},1_a,2_c)}\sigma_{M+2}^{(\overline{x},2_c,1_a,2_b)} = \sigma_{M+1}^{(\overline{x},1_a,2_{b\times c})}, \quad \sigma_{M+2}^{(\overline{x},1_b,1_a,2_c)}\sigma_{M+1}^{(\overline{x},1_b,2_c)} = \sigma_{M+1}^{(\overline{x},1_{a\times b},2_c)}. \tag{50}$$

As explained in Section 3.1 for $N = 4$, the relations (50) show a key property of the graph-braided anyon models. Namely, the graph braiding exchange operator representing a simple braid $\sigma_i^{(\overline{x},y,1,2)}$ can be expressed in terms of the graph braiding exchange operators representing the simple braid $\sigma_{i-1}^{(\overline{x},1,2)}$. By repeating this argument $(i-1)$ times, we obtain that the graph braiding exchange operator representing a simple braid $\sigma_i^{(\overline{x},y,1,2)}$ can be expressed in terms of $F$- and $R$-symbols only.

Crucially, we have observed that the consistency relations (50) for $M > 1$ are readily implied by the (four-particle) consistency relations for $M = 1$ (a statement which we will prove elsewhere). The four-particle consistency relations are also presented in Section 3.1 in Equation (23) where we also explain that they lead to the octagon Equations (24).

3. **Charge conservation relations**. As we explained in Section 3.1, the total charge of a given subset of particles is conserved throughout an exchange process if this subset of particles can be enclosed by a disk such that no particle enters or leaves the disk during the exchange, see Figure 5. This implies certain diagonality conditions for the braiding exchange operators in appropriate bases. Namely, whenever the total charge of a subset of particles is conserved during an exchange, the corresponding braiding exchange operator must be diagonal in the basis where this subset of particles is joined by a common fusion channel. In particular, the exchange operator representing a simple graph braid (46) has to be diagonal in the left-fused basis (see the top panel of Figure 5). What is more, whenever $x(k) = x(k + 1)$ in (46) for some $k$, the corresponding exchange operator is diagonal in another basis where particles $k$ and $k + 1$ are joined by a common fusion channel. This observation has been used to derive the $N = 4$ square diagrams for the $X$- and $Y$-symbols in Figure 7 and Equations (18) and (19). For $N > 4$ this leads to fully analogous, but more complicated diagonality conditions.

On the other hand, anyon models with simplified symbols introduced in Section 3.2 avoid these complications as the charge conservation relations are automatically satisfied for these models. However, for the general (non-simplified) models on a trijunction we conjecture that it is enough to satisfy the charge conservation relations only for $N \leq 5$.

# D   Half-twist of the world-ribbons on junctions

In this section, we view the anyon "world-ribbons" as ribbon braiding diagrams embedded in $\mathbb{R}^3$, see the related work by Turaev [62]. The concept of half twist has been discussed in the categorical context in Refs. [68, 35, 66].

By considering anyons' world-lines as world-ribbons, we need to introduce some extra moves which induce a half-twist (sometimes called a $\pi$-twist) of the world-ribbon at hand. Such a half-twist can be realised in the planar theory in the way shown in Figure 30 and will be denoted by $\tau$.

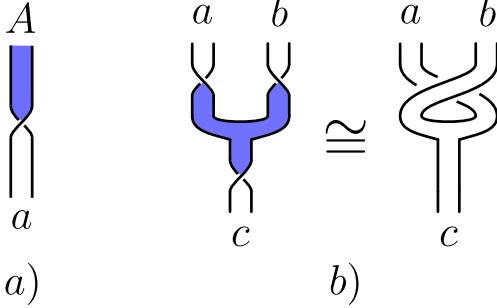

Figure 30: Half-twists of world-ribbons in the plane. For clarity, we colour the two sides of the ribbon by white and blue. a) A world-ribbon half-twist in the plane, $\tau$. b) Three consecutive half-twists are equivalent to the simple exchange $\sigma_1$. Note the orientations of the twists.

In order to incorporate the half-twists as morphisms of the topological Hilbert spaces, we denote the two sides of the world-ribbon of anyon $a$ by $a$ (white ribbon) and $A$ (blue ribbon) respectively. Let us start with the simplest situation of a single world-ribbon. The resulting quantum states form the following Hilbert spaces

- two one-dimensional spaces $V_a$ and $V_A$ for the non-twisted ribbon,

- two one-dimensional spaces for the anti-clockwise twisted ribbon denoted by $V_A^a$ and $V_a^A$,

- two one-dimensional spaces for the clockwise twisted ribbon denoted by $\tilde{V}_A^a$ and $\tilde{V}_a^A$.

Note that the ribbon half-twists are not local operations, as they change the boundary conditions at the endpoints of the world-ribbon. As such, they do not have corresponding gauge-invariant symbols. However, because a twist is a morphism between two one-dimensional Hilbert spaces, we can represent it as a complex number. Consequently, an anti-clockwise half-twist of a world-ribbon induces a morphism $\hat{\tau}$ between the one-dimensional Hilbert spaces $V_A$ and $V_a^A$ or $V_a$ and $V_A^a$. By picking bases of the relevant one-dimensional spaces we can represent the morphism $\hat{\tau}$ by ( gauge dependent ) complex numbers $T_a^A$ and $T_A^a$. Similarly, by $\bar{\tau}$ we will denote the morphism between the vector spaces $V_A$ and $\tilde{V}_a^A$ or $V_a$ and $\tilde{V}_A^a$ with the clockwise twist. The morphism $\hat{\bar{\tau}}$ will be represented by the complex number $\tilde{T}_a^A$ and $\tilde{T}_A^a$ – see Figure 31.

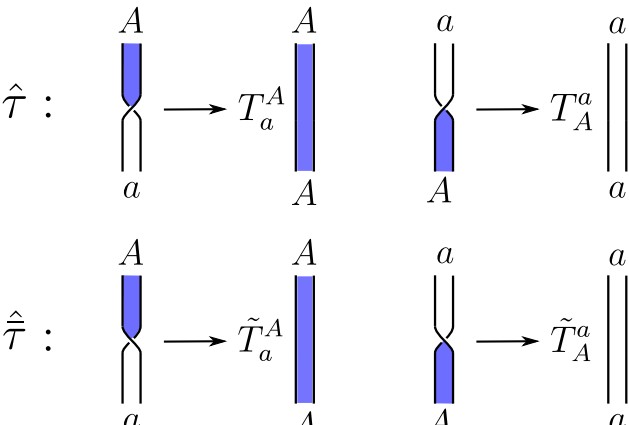

Figure 31: The morphisms $\hat{\tau}$ and $\hat{\tilde{\tau}}$ representing the anti-clockwise and clockwise half-twists respectively.

Clearly, composing an anti-clockwise half-twist with a clockwise half-twist results in a trivial move, thus we have the relations

$$T_a^A \tilde{T}_A^a = 1, \quad T_A^a \tilde{T}_a^A = 1.$$

What is more, two half-twists amount to a full twist represented by the twist factors. This gives rise to the relation

$$T_a^A T_A^a = \theta_A = \theta_a.$$

In other words, there is a canonical isomorphism between the spaces $V_A^a$ and $\tilde{V}_A^a$ or $V_a^A$ and $\tilde{V}_a^A$ induced the full twist and represented by the (gauge-invariant) twist factors $\theta_A = \theta_a$. Moreover, by the homotopy relation from Figure 30b we can connect the $T$-symbols with the $R$-symbols via

$$\frac{T_c^C}{T_A^a T_B^b} = R_c^{ba}.$$

A half-twist can be realised on a trijunction in the way where the ribbon visits edge (1) of the junction, moves to edge (2) and goes back to its original position. Such a move will be denoted by $\tau^{(1,2)}$ – see Figure 32a. Similarly, for $N = 2$ world-ribbons we can define half-twists of the world-ribbon of the anyon which is further from the junction. Then, the first anyon needs to make space for the half-twist by first moving either to edge (1) or edge (2) of the junction. This leads to two independent ways of twisting the second anyon's world-ribbon which we denote by $\tau^{(112)}$ and $\tau^{(212)}$ respectively – see Figure 32b and Figure 32c.

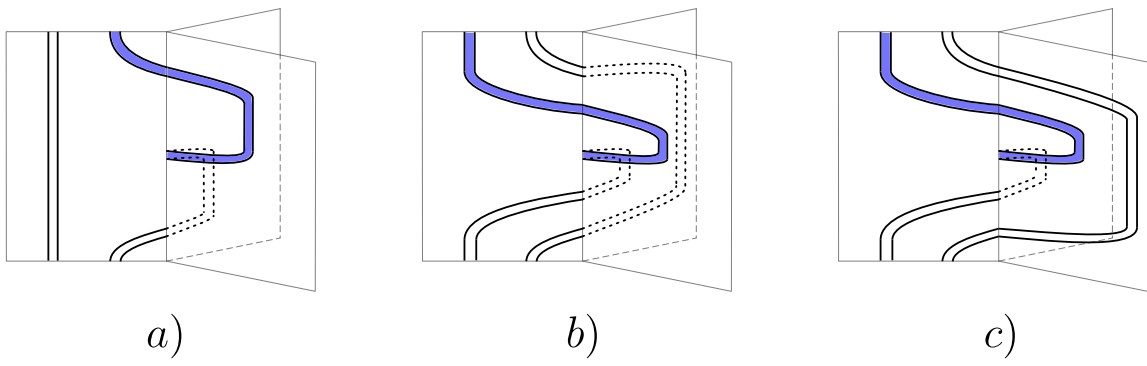

Figure 32: Three independent world-ribbon half-twists for $N = 2$ anyons on a trijunction. For clarity, we colour the two sides of the ribbon by white and blue. a) The move $\tau^{(1,2)}$ where the world-ribbon of the anyon located closest to the junction gets the half-twist. b) and c) The moves $\tau^{(112)}$ and $\tau^{(212)}$ where the world-ribbon of the anyon located furthest from the junction gets the half-twist.

The trijunction half-twists are represented by analogous morphisms of the topological Hilbert spaces as it was in the case of the half-twists in the plane. There also exists a trijunction counterpart of the relation from

Figure 30b which reads (see also Figure 33)

$$\tau^{(1_c,2_C)}\tau^{(2_B,2_A,1_a)}\tau^{(2_B,1_b)} = \tau^{(1_a,1_b,2_B)}\sigma_1^{(1_a,2_B)}\tau^{(2_B,1_b)}.$$

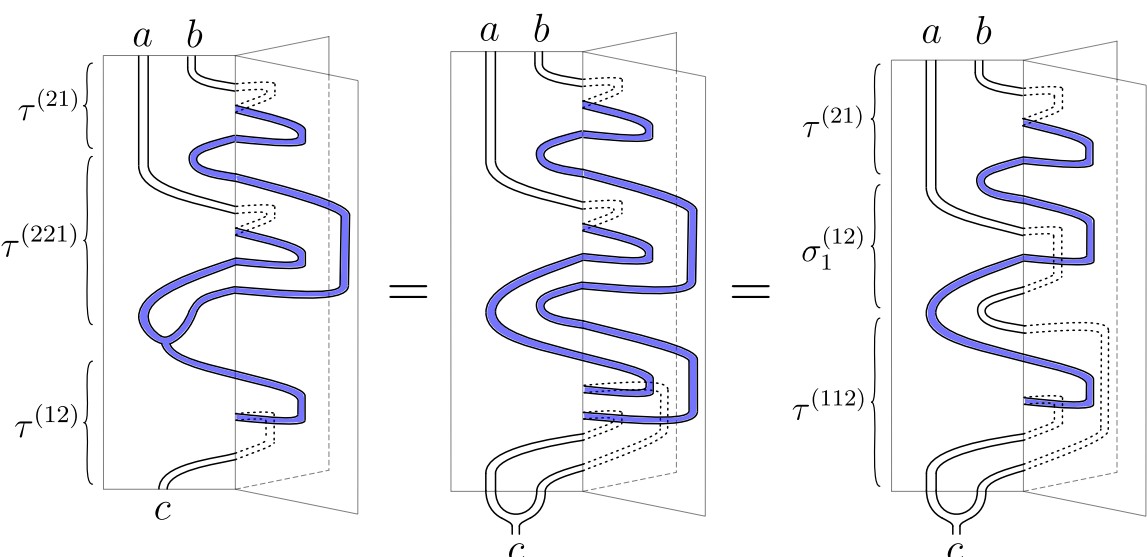

Figure 33: A relation between half-twists on the trijunction which is a counterpart of the relation between the corresponding moves in the plane from Figure 30b.

The moves $\tau^{(1,2)}$, $\tau^{(1,1,2)}$ and $\tau^{(2,1,2)}$ generalise to respective half-twists $\tau_v^{(1,2)}$, $\tau_v^{(1,1,2)}$ and $\tau_v^{(2,1,2)}$ on any tree graph in a natural way by embedding them on a local trijunction at an essential vertex $v$.

There is an important difference between the half-twists in the planar anyon theory and the above introduced half-twists in the anyon theory on networks. Namely, in $2D$ any spacetime diagram which involves fusion, braiding and half-twists can be resolved using local $R$-moves and full twists provided that all the world-ribbons that enter the diagram and leave the diagram are of the same colour (e.g. all white) [62]. An example of that is shown in Figure 30b) where three half-twists are resolved by a single $R$-move. However, this is no longer true for spacetime diagrams on a network. For instance, consider an analogous diagram involving three half-twists of the world-ribbons which is shown on the leftmost panel in Figure 33. It is not possible to continuously pull the bottom half-twist in the rightmost panel in Figure 33 through the $\sigma_1^{(1,2)}$ graph braid to cancel the top half-twist. Thus, it is not possible to resolve the spacetime diagram from the rightmost panel in Figure 33 using an $R$-move.

## D.1 Anyon models on the $\Theta$-graph: proving $R_e^{ba} = \tilde{R}_e^{ba}$ and $\tilde{Q}_{ed}^{bac} = \tilde{R}_e^{ba}$

In this section, we continue the proof from Section 7. Recall that the aim is to prove that any anyon model on the $\Theta$-graph yields a planar anyon model provided that the circular moves $\delta$ and $\bar{\delta}$ are represented by the same $D$-symbols. Let us start with deriving the equality $R_e^{ba} = \tilde{R}_e^{ba}$, which means that the braiding exchange operators at $v$ and $w$ are equal (in contrast to the $H$ graph in Section 3.3 where these braiding exchange operators were independent of each other).

Consider the lollipop embeddings $\Gamma_{L,v}$ and $\Gamma_{L,w}$ from Figure 21 and their corresponding three-anyon $\Delta_v$- and $\Delta_w$-moves (introduced in Section 5.1). We have the relations connecting the respective $\Delta_v$- and $\Delta_w$-moves with the simple braids $\sigma_1^{v;(1,2)}$ and $\sigma_1^{w;(1,2)}$ via the $\delta$-move (shown in Figure 34)

$$\delta = \sigma_1^{v;(1,2)}\Delta_v = \sigma_1^{w;(1,2)}\Delta_w$$

which imply

$$\sigma_1^{v;(1,2)}\Delta_v = \sigma_1^{w;(1,2)}\Delta_w. \tag{51}$$

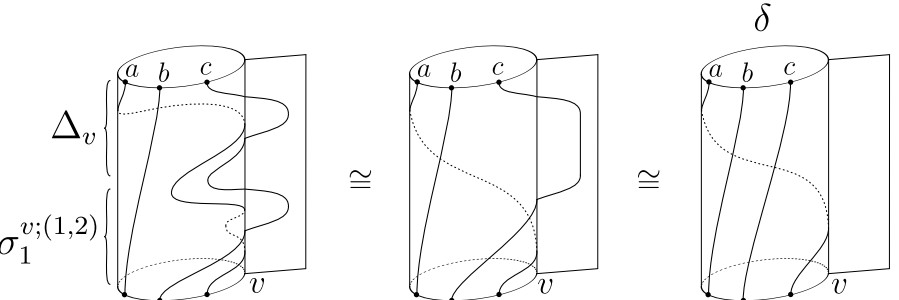

Figure 34: The homotopy equivalence $\sigma_1^{v;(1,2)}\Delta_v = \delta$. The same reasoning holds for the relation $\sigma_1^{w;(1,2)}\Delta_w = \delta$ associated with the lollipop subgraph $\Gamma_{L,w}$.

Relation (51) translates to the following hexagon, where $\Delta_v$ and $\Delta_w$ are represented by the symbols $G$ and $\tilde{G}$ respectively.

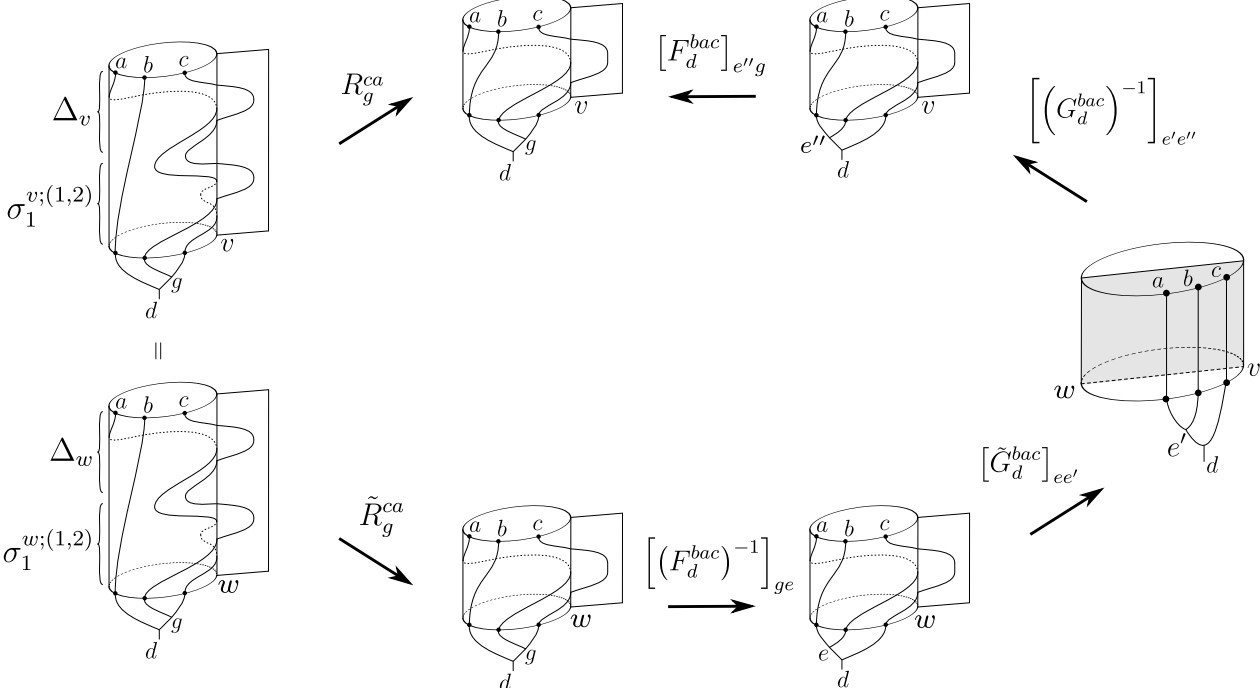

Figure 35: The hexagon following from the relation (51). Note that all the states should be treated as states on the $\Theta$-graph. For the sake of the clarity of the presentation, the upper and lower paths of the diagram only show the relevant embedded lollipops $\Gamma_{L,v}$ and $\Gamma_{L,w}$ from Figure 21.

$$R_g^{ca} = \tilde{R}_g^{ca} \sum_{e,e',e''} \left[\left(F_d^{bac}\right)^{-1}\right]_{ge} \left[\tilde{G}_d^{bac}\right]_{ee'} \left[\left(G_d^{bac}\right)^{-1}\right]_{e'e''} \left[F_d^{bac}\right]_{e''g}. \tag{52}$$

Next, we argue that the moves $\Delta_v$ and $\Delta_w$ are in fact represented by the same $G$-symbols. If this is the case, then Equation (52) simplifies to the desired relation $R_g^{ca} = \tilde{R}_g^{ca}$. To prove the equality of the $G$-symbols, consider an auxiliary move $\gamma$ which takes an anyon around the top loop of the $\Theta$-graph in an anti-clockwise fashion (see the right panel in Figure 36). The move $\gamma$ can be expressed via the $\delta$-moves and a half-twist as

$$\gamma = \bar{\delta}\delta^{-1}\tau_w^{(1,2)}, \tag{53}$$

where $\tau_w^{(1,2)}$ is the world-ribbon half-twist at vertex $w$ as defined at the beginning of Appendix D (Figure 32a). This relation is proved in Figure 36.

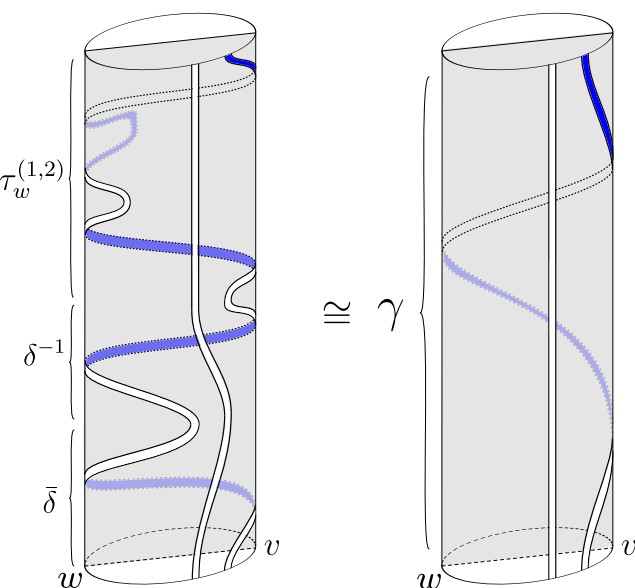

Figure 36: A pictorial proof of the homotopy equivalence of world-ribbons $\gamma = \bar{\delta}\delta^{-1}\tau_w^{(12)}$.

The assumption that both $\delta$ and $\bar{\delta}$ moves are represented by the same set of $D$-symbols implies that $\gamma$ induces the same morphisms of the topological Hilbert spaces as the half-twist $\tau_w^{(1,2)}$. In the remaining part of this section, we assume that the half-twist $\tau_w^{(1,2)}$ of the world-ribbon of anyon $a$ is a morphism of one-dimensional Hilbert spaces only (i.e. it does not depend on the spacetime histories of the remaining anyons in the fusion tree). Under such an assumption, the morphism representing the half-twist $\tau_w^{(1,2)}$ can be represented as a complex number $T_a$. We have the relation

$$\gamma^{-1}\Delta_v\gamma = \Delta_w, \tag{54}$$

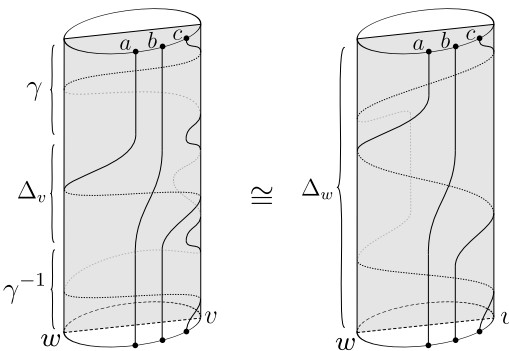

Figure 37: The homotopy equivalence $\gamma^{-1}\Delta_v\gamma = \Delta_w$. The left diagram includes two nontrivial half-twists of the world ribbons implicitly included in the $\gamma$-moves via the relation (53). For the sake of simplicity, the ribbon structure is not shown and only particles' world lines are presented.

The relation (54) implies that $\Delta_v = \Delta_w$ ($G = \tilde{G}$ in terms of the $G$-symbols) – see Figure 38.

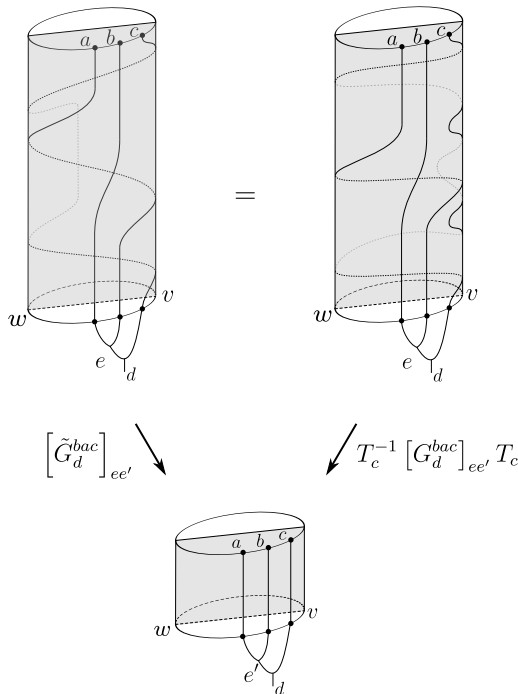

Figure 38: The diagram following from the relation (54). Assuming that both circular moves $\delta$ and $\bar{\delta}$ are represented by the same $D$-symbols, the polygon implies $T_c^{-1}\left[G_d^{bac}\right]_{ee'} T_c = \left[\tilde{G}_d^{bac}\right]_{ee'}$, i.e. $\left[\tilde{G}_d^{bac}\right]_{ee'} = \left[G_d^{bac}\right]_{ee'}$. For the sake of simplicity, the ribbon structure is not shown and only particles' world lines are presented.

Hence, using the fact that $G_d^{bac} = \tilde{G}_d^{bac}$ in (52), we obtain

$$R_g^{ca} = \tilde{R}_g^{ca} \sum_{e,e''}\left[\left(F_d^{bac}\right)^{-1}\right]_{ge} \delta_{ee''}\left[F_d^{bac}\right]_{e''g} = \tilde{R}_g^{ca}.$$

Finally, let us show that $\tilde{Q}_{ed}^{bac} = \tilde{R}_e^{ba}$. To this end, we use another relation involving the $\gamma$-move which reads

$$\gamma\sigma_2^{w;(2,1,2)}\gamma^{-1} = \left[\beta_{v,2},\left(\beta_{v,1}\right)^2\right]\sigma_1^{w;(1,2)}\left[\left(\beta_{v,1}\right)^2,\beta_{v,2}\right], \tag{55}$$

where we use the notation involving the $\beta$-moves described in Appendix A. In order to prove relation (55), we first note that the LHS is homotopy equivalent to

$$\gamma\sigma_2^{w;(2,1,2)}\gamma^{-1} = \beta_{v,2}\sigma_1^{w;(1,2)}\beta_{v,2}^{-1}.$$

Next, we expand the RHS and LHS completely in terms of the corresponding $\beta$-moves as follows.

$$\beta_{v,2}\sigma_1^{w;(1,2)}\beta_{v,2}^{-1} = \left(\beta_{v,2}\beta_{w,1}\beta_{w,2}\right)\left(\beta_{w,1}^{-1}\beta_{w,2}^{-1}\beta_{v,2}^{-1}\right),$$
$$\left[\beta_{v,2},\left(\beta_{v,1}\right)^2\right]\sigma_1^{w;(1,2)}\left[\left(\beta_{v,1}\right)^2,\beta_{v,2}\right] = \left(\left[\beta_{v,2},\left(\beta_{v,1}\right)^2\right]\beta_{w,1}\beta_{w,2}\beta_{v,2}\right) \times$$
$$\times \left(\beta_{v,2}^{-1}\beta_{w,1}^{-1}\beta_{w,2}^{-1}\left[\left(\beta_{v,1}\right)^2,\beta_{v,2}\right]\right).$$

Note that in the last equality we have not only expanded the braids in terms of $\beta$-moves, but also inserted an extra expression $\beta_{v,2}\beta_{v,2}^{-1}$ which is homotopy equivalent to the trivial move. In Figure 39 we prove the first "half" of the relation (55), i.e.

$$\beta_{v,2}\beta_{w,1}\beta_{w,2} \cong \left[\beta_{v,2},\left(\beta_{v,1}\right)^2\right]\beta_{w,1}\beta_{w,2}\beta_{v,2}. \tag{56}$$

The homotopy equivalence of the other pair of the relevant terms follows in an analogous way.

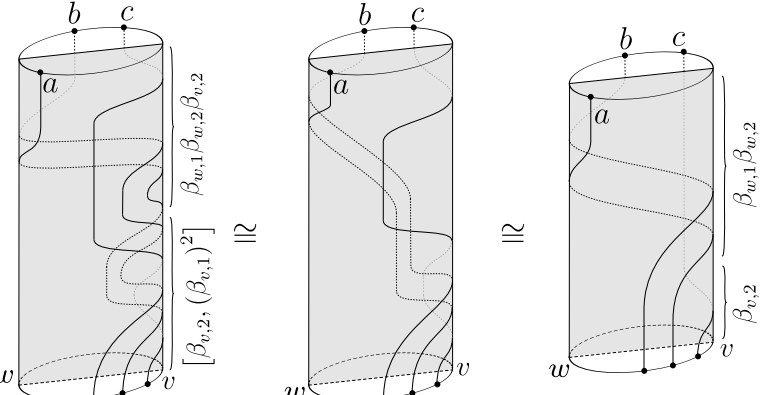

Figure 39: A pictorial proof of the homotopy equivalence (56).

Let us next consider the polygon which corresponds to the relation (55). In the leftmost and rightmost pictures of the Figure 39 we can see that most of the relevant moves do not split the worldlines of anyons $a$ and $b$ which start as the furthest ones from the vertex $v$. Thus, we can fuse the two anyons in the common channel $e = a \times b$ to simplify the corresponding polygon equation so that no $F$-moves are used (see Figure 40).

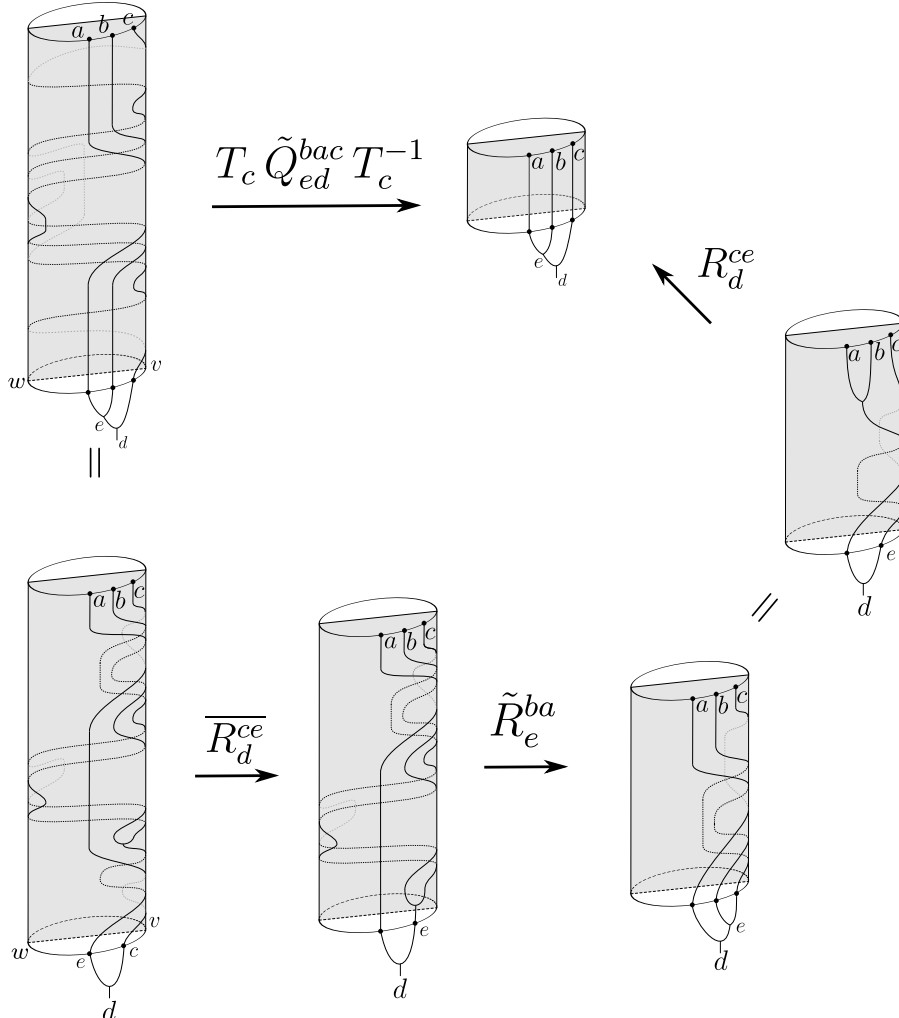

Figure 40: The polygon following from the relation (55).

The polygon from Figure 40 yields

$$T_c \tilde{Q}^{bac}_{ed} T_c^{-1} = \overline{R^{ce}_d} \tilde{R}^{ba}_e R^{ce}_d \,,$$

which implies $\tilde{Q}^{bac}_{ed} = \tilde{R}^{ba}_e$.

# E Solving the graph braid polygon equations

Solving the graph braid equations for a graph $\Gamma$ and a set of anyons, whose fusion theory is described by a given set of $F$-symbols, comes down to solving a system of polynomial equations. These are determined by the various commuting diagrams, in particular

(a) if $\Gamma$ is the lollipop graph, equations (13), (14), (34), and (29) must be satisfied, while

(b) if $\Gamma$ is the trijunction, equations (13) and (14) must be satisfied (for three particles) together with some extra equations for four particles: (62), (60), (61), and (63).

In what follows we will assume that the $F$-symbols are given a priori. To solve the graph braid equations we implemented algorithms in the Wolfram Language that can be broken down into the following routines.

1. Setting up the relevant equations.

2. Solving all (monomial) equations of the form $m_1 = m_2$, where $m_i$ are monomials.

3. Substituting solutions from the previous step in the remaining equations. If some equations become monomial equations, return to the previous step and repeat. If there are no new monomial equations the remaining system is solved using the built-in Solve function of the Wolfram Language.

Since for both the lollipop and trijunction graphs, equations (13) and (14) need to be satisfied, it is beneficial to start by searching for admissible sets of $P, Q,$ and $R$-symbols for each given set of $F$-symbols. By inverting some of the arrows and going around the whole hexagon, these equations can be re-expressed in terms of $P$ and $Q$ as follows

$$P_{ed}^{cab} \delta_{ee'} = \sum_{f,g} \left[ (F_d^{acb})^{-1} \right]_{ge'} \left[ F_d^{cab} \right]_{ef} R_d^{cf} \left[ F_d^{abc} \right]_{fg} (R_g^{cb})^{-1}, \tag{57}$$

$$Q_{ed}^{bac} \delta_{ee'} = \sum_{f,g} \left[ F_d^{bac} \right]_{e'g} (R_g^{ca})^{-1} \left[ (F_d^{bca})^{-1} \right]_{gf} R_d^{fa} \left[ (F_d^{abc})^{-1} \right]_{fe}. \tag{58}$$

Here the $\delta_{ee'}$ appears as a consequence of the fact that we demand $P$ and $Q$ to preserve the charge $e$. For $e \neq e'$ we get a consistency equation on the $R$-symbols which can be solved in terms of $R$. When $e = e'$ we get a definition for the $P$ and $Q$ symbols in terms of the $R$-symbols we solved for.

For the trijunction with three particles, no extra equations need to be added. For systems with four particles, there are four new simple braid generators $\sigma_3^{(1,2,1,2)}, \sigma_3^{(2,1,1,2)}, \sigma_3^{(1,1,1,2)},$ and $\sigma_3^{(2,2,1,2)}$ for which we use the symbols $A, B, X, Y$ respectively.

In the following, we will work with the following convention for fusion labels for four particles

$$\begin{array}{lllll} a \times b = f, & a \times c = n, & a \times d = m, & a \times b \times c = g, & a \times b \times d = j, \\ b \times c = h, & b \times d = y, & c \times d = l, & a \times c \times d = r, & b \times c \times d = k. \end{array} \tag{59}$$

First of all, we need to take into account the fact that both $X$ and $Y$ can be expressed in terms of $P$ and $Q$ after a change of basis. Equation (18), together with its counterpart for $Y$, can be rewritten as

$$X_{fge}^{bacd} \delta_{gg'} = \sum_l \left[ F_e^{fcd} \right]_{gl} P_{fe}^{bal} \left[ (F_e^{fcd})^{-1} \right]_{lg'}, \tag{60}$$

$$Y_{fge}^{bacd} \delta_{gg'} = \sum_l \left[ F_e^{fcd} \right]_{gl} Q_{fe}^{bal} \left[ (F_e^{fcd})^{-1} \right]_{lg'}. \tag{61}$$

Second, there is the pseudocommutative relation (21) which can be rewritten as the following demand

$$A_{fje}^{badc} \delta_{jj'} = \sum_{l,g,l'} \left[ F_e^{fdc} \right]_{jl} R_l^{dc} \left[ (F_e^{fcd})^{-1} \right]_{lg} B_{fge}^{bacd} \left[ F_e^{fcd} \right]_{gl'} (R_{l'}^{dc})^{-1} \left[ (F_e^{fdc})^{-1} \right]_{l'j'}. \tag{62}$$

Last, we need to take account of equations (23). Although there are four equations in total, three of these are satisfied once we demand that equations (60), (61), (62) together with the $N = 3$ P- and Q-hexagons hold. The only independent equation that remains is then

$$\delta_{nn'}\delta_{gg'}B_{nge}^{cabd} = \sum_{f,h,k}\left[F_g^{cab}\right]_{nf}Q_{ge}^{cfd}\left[F_g^{abc}\right]_{fh}\left[F_e^{ahd}\right]_{gk}(Q_{hk}^{cbd})^{-1}\left[(F_e^{ahd})^{-1}\right]_{kg'}\left[(F_{g'}^{acb})^{-1}\right]_{hn'}. \quad (63)$$

Just like for the $P$ and $Q$ symbols some of these equations can be used as defining equations and others as consistency requirements. Interestingly, once the $R$, $P$, and $Q$ symbols are known, no equations need to be solved since all the new equations have a right-hand side that is completely determined by the values of $R$, $P$, and $Q$. Since there are now multiple equations that define the same symbols one should also check that the definitions are consistent with one another.

For the lollipop graph, extra constraints on the $P$ and $R$-symbols (34) together with equations (29) for the $D$-symbols, need to be added. The equations for the $D$-symbols can be solved separately. If there is no gauge freedom left after fixing the $F$-symbols then the solutions to the lollipop equations consist of all possible combinations of solutions to the circle equations (29) with solutions to equations (13), (14), and (34). Of all the anyon models we investigated there is only one model which has gauge freedom left after fixing the $F$-symbols: $\mathbb{Z}_2 \times \mathbb{Z}_2$. The method with which we constructed solutions to the lollipop equations for $\mathbb{Z}_2 \times \mathbb{Z}_2$ is described in H.3. Once the lollipop equations were solved we checked the planarity of the solutions by checking whether $R_e^{ab} \equiv P_{ed}^{abc} \equiv Q_{ed}^{abc}$ and $D_e^{ab} \equiv R_e^{ab}D_b^{1b}$.

Normally, before solving the equations, it is beneficial to break gauge symmetry. As explained in Section (2), given a solution for the graph braid equations one can create an infinite set of other solutions by acting with a gauge transformation. Breaking such symmetry greatly reduces computation time as the number of variables, equations, and solutions decreases. Apart from the $F$-symbols, all symbols correspond to exchanging and therefore transform in the same way:

$$S' = \frac{u_x^{ab}}{u_x^{ba}}S. \quad (64)$$

Since the $F$-symbols are given a priori we do have the extra demand that the $F$-symbols are invariant under such transforms. Interestingly, for all sets of $F$-symbols we considered with the exception of $\mathbb{Z}_2 \times \mathbb{Z}_2$, demanding gauge invariance results in removing all gauge freedom in the remaining symbols. For $\mathbb{Z}_2 \times \mathbb{Z}_2$ the remaining gauge transforms form a $\mathbb{Z}_2$ group and we removed this symmetry after solving the equations.

Solving systems of monomial equations can be done using linear algebra. Indeed, because none of the variables appearing in the graph braid equations are allowed to be 0, one can take a logarithm of the monomial equations to convert them to a set of linear equations with integer coefficients. Each equation only holds modulo an integer times $2\pi i$ due to the multivaluedness of the logarithm. Solving a system of linear equations modulo a discrete subspace can be done by computing a Smith normal form [56]. Once the linear system has been solved one can exponentiate the solutions, which typically contain some continuous freedom, and substitute them back in the original equations. The system of equations then reduces to a smaller system which might contain new monomial equations. If so, one only needs to repeat the above procedure until no monomial equations are left. To solve the remaining equations the built-in Solve command from Mathematica was used.

For the rings of type $SU(2)_k$ and $PSU(2)_k$, we did not have access to all solutions to the pentagon equations and made do with a single solution, obtained using the methods in [7]. Moreover, the specific form of the solutions for $k$ an odd number were too complicated to derive the solutions symbolically. Eventually, we did find symbolic solutions by solving the systems numerically and reverting the numeric solutions to roots of polynomial equations. All solutions obtained this way were found to be correct with an accuracy of 1000 decimal digits, and an infinite precision (meaning the computer used as many internal extra digits as needed to ensure all 1000 digits are correct).

# F   Graph braiding obstruction for Tambara-Yamagami models

In this section we will describe the obstruction to a solution of the $d$ valent star graph braiding hexagon equations for Tambara Yamagami over $G$, unless $G$ is $\mathbb{Z}_2$ to some power. We will focus on $d = 3$, a trijunction,

however, the analysis in [18], shows that this is true for any valence. We will examine the expressions for $P_{g\sigma}^{\sigma\sigma\sigma}$ to deduce that $\overline{\chi}(g_1, g_2) = \chi(g_1, g_2)$. This then implies that $\chi(g_1^2, g_2) = 1$ for all $g_1, g_2 \in G$ and since $\chi$ is a non-degenerate bi-character we must have that $g_1^2$ is the group unit for all $g_1$. We begin with (14) and $a = b = c = d = \sigma$

$$P_{g\sigma}^{\sigma\sigma\sigma} \kappa\tau^{-1}\overline{\chi}(g,f)R_f^{\sigma\sigma} = \kappa^2\tau^{-2}\sum_e \overline{\chi}(g,e)R_\sigma^{\sigma e}\,\overline{\chi}(e,f). \tag{65}$$

The values for $g, f$ and $e$ are group elements since they come from the multiplication of two $\sigma$ particles, we denote them as $g_1$, $g_2$ and $g_3$ respectively. Additionally, the only label that will matter for $P_{g_1\sigma}^{\sigma\sigma\sigma}$ is $g_1$ so we will write $P_{g_1\sigma}^{\sigma\sigma\sigma} := P(g_1)$. The equation for $P$ using this notation and configuration of anyons is given by

$$P(g_1) = \kappa\tau^{-1}\chi(g_1, g_2)\,\overline{R_{g_2}^{\sigma\sigma}}\sum_{g_3}\overline{\chi}(g_1 g_2, g_3)\,R_\sigma^{\sigma g_3}, \quad \forall g_2 \in G. \tag{66}$$

Now substitute $g_1 h$ for $g_1$ and $h^{-1}g_2$ for $g_2$, where $h$ is an arbitrary element of $G$. Then we get that,

$$P(g_1 h) = \kappa\tau^{-1}\chi(g_1 h, h^{-1}g_2)\overline{R_{h^{-1}g_2}^{\sigma\sigma}}\sum_{g_3}\overline{\chi}(g_1 g_2, g_3)\,R_\sigma^{\sigma g_3}, \quad \forall h, g_2 \in G. \tag{67}$$

All of the left-hand-sides and right-hand-sides of expressions (66) and (67) are non zero complex numbers so we can perform the quotient. Since both have the same terms in the sum over $g_3$, we get the following expression,

$$\frac{P(g_1)}{P(g_1 h)} = \frac{\chi(g_1, g_2)\overline{R_{g_2}^{\sigma\sigma}}}{\chi(g_1 h, h^{-1}g_2)\overline{R_{h^{-1}g_2}^{\sigma\sigma}}} \tag{68}$$

$$= \frac{\chi(g_1, g_2)\overline{R_{g_2}^{\sigma\sigma}}}{\chi(g_1, g_2)\chi(g, h^{-1})\chi(h, h^{-1})\chi(h, g_2)\overline{R_{h^{-1}g_2}^{\sigma\sigma}}} \tag{69}$$

$$= R_{h^{-1}g_2}^{\sigma\sigma}\,\overline{R_{g_2}^{\sigma\sigma}}\,\overline{\chi}(g_1 h g_2^{-1}, h^{-1}), \quad \forall h, g_2 \in G. \tag{70}$$

where we have used the fact that $\chi$ is a symmetric bicharacter to expand, and then simplify, the denominator.

Furthermore, since $h$ is arbitrary, we can fix $h = g_1^{-1}$ to get

$$\frac{P(g_1)}{P(1)} = R_{g_1 g_2}^{\sigma\sigma}\,\overline{R_{g_2}^{\sigma\sigma}}\,\overline{\chi}(g_2^{-1}, g_1), \quad \forall g_2 \in G. \tag{71}$$

In particular for $g_2 = 1$, the vacuum charge, this expression simplifies to,

$$\frac{P(g_1)}{P(1)} = R_{g_1}^{\sigma\sigma}\,\overline{R_1^{\sigma\sigma}}. \tag{72}$$

We can combine equation (71) with (72) to get the following equation

$$R_{g_1 g_2}^{\sigma\sigma}\,\overline{R_{g_2}^{\sigma\sigma}}\,\overline{\chi}(g_2^{-1}, g_1) = R_{g_1}^{\sigma\sigma}\,\overline{R_1^{\sigma\sigma}}, \quad \forall g_2 \in G. \tag{73}$$

Using the fact that $\chi$ is a symmetric bicharacter, we can rearrange this equation to get

$$R_{g_1 g_2}^{\sigma\sigma} = \frac{R_{g_1}^{\sigma\sigma} R_{g_2}^{\sigma\sigma}}{R_1^{\sigma\sigma}\chi(g_1, g_2)}. \tag{74}$$

Since the expressions for $Q_{g_1\sigma}^{\sigma\sigma\sigma}$ in equation (13) have inverse $F$- symbols, if we follow the same steps we get,

$$R_{g_1 g_2}^{\sigma\sigma} = \frac{R_{g_1}^{\sigma\sigma} R_{g_2}^{\sigma\sigma}}{R_1^{\sigma\sigma}\overline{\chi}(g_1, g_2)} \tag{75}$$

But both of these expressions must be simultaneously true so we can equate them to deduce;

$$\chi(g_1, g_2) = \overline{\chi}(g_1, g_2). \tag{76}$$

Therefore there are only solutions for the graph braiding hexagon equations for the Tambara-Yamagami fusion category if $G$ is $\mathbb{Z}_2$ to some power.

# G  Solutions for the Ising model

The Ising fusion ring has 3 particles, $1, \psi, \sigma$ subject to the multiplication rules

$$1 \times a = a \times 1 = a, \quad \forall a \in \{1, \psi, \sigma\}, \tag{77}$$

$$\psi \times \psi = 1, \quad \sigma \times \psi = \psi \times \sigma = \sigma, \quad \sigma \times \sigma = 1 + \psi. \tag{78}$$

In the following section we will list the solutions to various equations for the Ising model. To save space we omit any well-defined symbol equal to 1. By well-defined we mean that the fusion tree corresponding to the symbol exists.

## G.1  Solutions to the pentagon equations

There are two solutions to the pentagon equations for the Ising fusion ring. Both solutions share the same values for the following $F$ symbols

$$[F_\sigma^{\psi\sigma\psi}]_{\sigma\sigma} = [F_1^{\sigma\psi\sigma}]_{\sigma\sigma} = [F_1^{\sigma\sigma\psi}]_{\psi\sigma} = [F_\psi^{\sigma\sigma\psi}]_{1\sigma} = -1, \tag{79}$$

but have a different sign for the $F$-matrix

$$\left[F_\sigma^{\sigma\sigma\sigma}\right] = \pm\frac{1}{\sqrt{2}}\begin{bmatrix} 1 & -1 \\ 1 & 1 \end{bmatrix}. \tag{80}$$

We will denote these solutions by $\mathcal{F}_\kappa$ where $\kappa = \pm 1$. Note that some of the $F$-symbols in these solutions are gauge dependent and so they may differ from those in other works. In, e.g. [51, 58] and [73], a gauge is used such that

$$\left[F_\sigma^{\sigma\sigma\sigma}\right] = \pm\frac{1}{\sqrt{2}}\begin{bmatrix} 1 & 1 \\ 1 & -1 \end{bmatrix}. \tag{81}$$

## G.2  Solutions to the planar hexagon equations

Each set of $F$-symbols allows four solutions to the planar hexagon equations. They can be parameterized as follows:

$$R_1^{\psi\psi} = -1, \quad R_\sigma^{\psi\sigma} = R_\sigma^{\sigma\psi} = \varepsilon_1 i, \quad R_1^{\sigma\sigma} = \left(\frac{\kappa\varepsilon_1}{i}\right)^{\frac{1+\kappa}{2}} i^{\varepsilon_2+1} e^{-\varepsilon_1 \frac{i\pi}{8}}, \quad R_1^{\sigma\sigma} = \left(\frac{\kappa\varepsilon_1}{i}\right)^{\frac{1-\kappa}{2}} i^{\varepsilon_2+1} e^{-\varepsilon_1 \frac{i\pi}{8}},$$

where $\varepsilon_i \in \{-1, 1\}$.

## G.3  Solutions to the trijunction equations

### G.3.1  Three particles

For three particles each solution to the pentagon equation gives rise to two classes of solutions to the trijunction equations. Each class of solutions are parameterized by two complex phases $z_1, z_2$. To save space we will not denote the symbols $P_{ed}^{ab1} \equiv Q_{ed}^{ab1}$ since these are equal to the $R$-symbols $R_e^{ab}$. The four combinations of $F$-and $R$-symbols have the following form

$$\begin{aligned}
&P_{1\psi}^{\psi\psi\psi} = \tfrac{1}{z_1}, && P_{1\sigma}^{\psi\psi\sigma} = -1, && P_{\sigma\sigma}^{\psi\sigma\psi} = \tfrac{-\varepsilon i}{z_1}, && P_{\sigma 1}^{\psi\sigma\sigma} = -\varepsilon i z_1, && P_{\sigma\psi}^{\psi\sigma\sigma} = \varepsilon i, && P_{\sigma\sigma}^{\sigma\psi\psi} = \varepsilon i, \\
&P_{\sigma 1}^{\sigma\psi\sigma} = \varepsilon i, && P_{\sigma\psi}^{\sigma\psi\sigma} = \varepsilon i, && P_{1\psi}^{\sigma\sigma\psi} = z_2, && P_{\psi 1}^{\sigma\sigma\psi} = \varepsilon i z_2, && P_{1\sigma}^{\sigma\sigma\sigma} = \tfrac{\kappa}{z_2} e^{-\frac{\varepsilon i\pi}{4}}, && P_{\psi\sigma}^{\sigma\sigma\sigma} = \tfrac{\kappa}{z_2} e^{\frac{\varepsilon i\pi}{4}}, \\
&Q_{1\psi}^{\psi\psi\psi} = \tfrac{1}{z_1}, && Q_{1\sigma}^{\psi\psi\sigma} = -1, && Q_{\sigma\sigma}^{\psi\sigma\psi} = \varepsilon i, && Q_{\sigma 1}^{\psi\sigma\sigma} = \varepsilon i, && Q_{\sigma\psi}^{\psi\sigma\sigma} = \varepsilon i, && Q_{\sigma\sigma}^{\sigma\psi\psi} = \tfrac{-\varepsilon i}{z_1}, \\
&Q_{\sigma 1}^{\sigma\psi\sigma} = -\varepsilon i z_1, && Q_{\sigma\psi}^{\sigma\psi\sigma} = \varepsilon i, && Q_{1\psi}^{\sigma\sigma\psi} = z_2, && Q_{\psi 1}^{\sigma\sigma\psi} = \varepsilon i z_2, && Q_{1\sigma}^{\sigma\sigma\sigma} = \tfrac{\kappa}{z_2} e^{-\frac{\varepsilon i\pi}{4}}, && Q_{\psi\sigma}^{\sigma\sigma\sigma} = \tfrac{\kappa}{z_2} e^{\frac{\varepsilon i\pi}{4}}, \\
&R_1^{\psi\psi} = z_1, && R_\sigma^{\psi\sigma} = \varepsilon i, && R_\sigma^{\sigma\psi} = \varepsilon i, && R_1^{\sigma\sigma} = z_2, && R_\psi^{\sigma\sigma} = \varepsilon i z_2,
\end{aligned}$$

where $\varepsilon \in \{-1, 1\}$. We will label each solution by $\mathcal{T}_{\kappa,\varepsilon}^{(3)}$.

### G.3.2 Four particles

For four particles, the trijunction equations can only be satisfied if $z_1 = -1$ (which implies $R_1^{\psi\psi} = -1$) and therefore the solutions have the property that $P \equiv Q$. Note that this does not necessarily imply that $P \equiv R$, i.e. that the solutions are planar. The solutions are then described by adding to the $\mathcal{T}_{\kappa,\varepsilon}^{(3)}$ the respective values of the $A, B, X$, and $Y$ symbols which we will describe here. All symbols with a 1 as the third or fourth top label are $P$, $Q$, or $R$ symbols and will therefore not be listed.

For the Ising model, it turns out that all symbols with the same labels are equal to each other. We can thus write the solutions in terms of the symbol $M$, where $M$ could be any of $A, B, X, Y$. The solutions then read

$$M_{fge}^{\psi\psi cd} \equiv -1 \tag{82}$$

$$M_{fge}^{\sigma\psi cd} \equiv M_{fge}^{\psi\sigma cd} \equiv \varepsilon i \tag{83}$$

$$M_{fge}^{\sigma\sigma cd} = \begin{cases} z_2 & \text{if } c = d \text{ and } f = 1 \\ \varepsilon i z_2 & \text{if } c = d \text{ and } f = \psi \\ \frac{\kappa}{z_2} \exp\left(\frac{-\varepsilon i\pi}{4}\right) & \text{if } c \neq d \text{ and } f = 1 \\ \frac{\kappa}{z_2} \exp\left(\frac{\varepsilon i\pi}{4}\right) & \text{if } c \neq d \text{ and } f = \psi \end{cases} \tag{84}$$

where $c, d \in \{\psi, \sigma\}$, $f, g, e \in \{1, \psi, \sigma\}$, and the value of $\kappa$ and $\varepsilon$ are fixed by the choice of $\mathcal{T}_{\kappa,\varepsilon}^{(3)}$.

## G.4 Solutions to the lollipop equations

### G.4.1 Lollipop trijunction

For the Ising model on the trijunction, on the lollipop, the demand that $P_{ed}^{abc} \equiv R_e^{ab}$ implies that the solutions must be planar. In particular, the solutions are the four solutions for the planar hexagon equations with the addition of the $P$-and $Q$ symbols, which obey $P_{ed}^{abc} \equiv Q_{ed}^{abc} \equiv R_e^{ab}$.

### G.4.2 Circle solutions

There are sixteen solutions to the circle equations for each set of $F$-symbols. They can be written as

$$D_\psi^{1\psi} = -1, \ D_\sigma^{1\sigma} = \exp\left(i\pi \frac{-2 - \nu_1 + 4\nu_2 - 2\kappa}{8}\right), \ D_\sigma^{\psi\sigma} = -\nu_1 \exp\left(i\pi \frac{2 - \nu_1 + 4\nu_2 - 2\kappa}{8}\right),$$
$$D_\sigma^{\sigma\psi} = \nu_1 i, \ D_1^{\sigma\sigma} = \nu_3, \ D_\psi^{\sigma\sigma} = \nu_4 i,$$

where the $\nu_i \in \{-1, 1\}$ and $\kappa$ is fixed by the choice of $F$-symbols. In particular we find that, per set of $F$-symbols, there are four possible values for the generalized topological spins. These coincide with the values of the topological spins for planar Ising anyons.

### G.4.3 Full lollipop solutions

There are 32 solutions to the full lollipop equations per set of $F$-symbols. Because there is no gauge freedom left after fixing a set of $F$-symbols, for a given set of $F$-symbols any solution can be found by combining a solution to the lollipop trijunction equations with matching label $\kappa$ with a solution to the circle equations with matching label $\kappa$.

# H  Solutions for the quantum double of $\mathbb{Z}_2$

The quantum double of $\mathbb{Z}_2$ is a model with four anyons $1, e, m, \varepsilon$ that follow the fusion rules of $\mathbb{Z}_2 \times \mathbb{Z}_2$ (via, e.g., the identification $1 = (0,0), e = (1,0), m = (0,1), \varepsilon = (1,1)$) and for which $[F_d^{abc}]_{ef} \equiv 1$ for each well-defined $F$-symbol. This model arises as the excitations in the Toric code model with gauge group $\mathbb{Z}_2$ [40].

## H.1 Solutions to the planar hexagon equations

There are eight gauge-independent planar hexagon solutions:

$$R_1^{\varepsilon\varepsilon} = \nu_1, \ R_m^{\varepsilon e} = \nu_2, \ R_e^{\varepsilon m} = \nu_1\nu_2, \ R_1^{ee} = \nu_3, \ R_{\varepsilon}^{em} = \nu_3, \ R_e^{m\varepsilon} = \nu_1, \ R_{\varepsilon}^{me} = \nu_2\nu_3, \ R_1^{mm} = \nu_1\nu_2\nu_3,$$

where $\nu_i \in \{-1, 1\}$.

## H.2 Solutions to the trijunction equations

The trijunction equations that impose constraints on any of the $R$-symbols are trivially satisfied. Therefore, we find that all non-trivial $R$-symbols are free parameters

$$R_1^{\varepsilon\varepsilon} = z_1, \ R_m^{\varepsilon e} = z_2, \ R_e^{\varepsilon m} = z_3, \ R_m^{e\varepsilon} = z_4, \ R_1^{ee} = z_5, \ R_{\varepsilon}^{em} = z_6, \ R_e^{m\varepsilon} = z_7, \ R_{\varepsilon}^{me} = z_8, \ R_1^{mm} = z_9,$$

and all other symbols can be expressed in terms of these free parameters. To save space we will omit the symbols $P_{ed}^{ab1} \equiv Q_{ed}^{ab1}$ since these are equal to the $R$-symbols $R_e^{ab}$. The $P$-and $Q$ symbols are the following

$$
\begin{array}{cccccc}
P_{1\varepsilon}^{\varepsilon\varepsilon\varepsilon} = \frac{1}{z_1}, & P_{me}^{e\varepsilon\varepsilon} = \frac{1}{z_4}, & P_{em}^{m\varepsilon\varepsilon} = \frac{1}{z_7}, & Q_{1\varepsilon}^{\varepsilon\varepsilon\varepsilon} = \frac{1}{z_1}, & Q_{me}^{e\varepsilon\varepsilon} = \frac{z_7}{z_1}, & Q_{em}^{m\varepsilon\varepsilon} = \frac{z_4}{z_1}, \\[8pt]
P_{1e}^{\varepsilon\varepsilon e} = \frac{z_3}{z_2}, & P_{m\varepsilon}^{e\varepsilon e} = \frac{z_6}{z_5}, & P_{e1}^{m\varepsilon e} = \frac{z_9}{z_8}, & Q_{1e}^{\varepsilon\varepsilon e} = \frac{z_7}{z_4}, & Q_{m\varepsilon}^{e\varepsilon e} = \frac{1}{z_4}, & Q_{e1}^{m\varepsilon e} = \frac{z_1}{z_4}, \\[8pt]
P_{1m}^{\varepsilon\varepsilon m} = \frac{z_2}{z_3}, & P_{m1}^{e\varepsilon m} = \frac{z_5}{z_6}, & P_{ee}^{m\varepsilon m} = \frac{z_8}{z_9}, & Q_{1m}^{\varepsilon\varepsilon m} = \frac{z_4}{z_7}, & Q_{m1}^{e\varepsilon m} = \frac{z_1}{z_7}, & Q_{ee}^{m\varepsilon m} = \frac{1}{z_7}, \\[8pt]
P_{me}^{\varepsilon e\varepsilon} = \frac{z_3}{z_1}, & P_{1\varepsilon}^{ee\varepsilon} = \frac{z_6}{z_4}, & P_{\varepsilon1}^{me\varepsilon} = \frac{z_9}{z_7}, & Q_{me}^{\varepsilon e\varepsilon} = \frac{1}{z_2}, & Q_{1\varepsilon}^{ee\varepsilon} = \frac{z_8}{z_2}, & Q_{\varepsilon1}^{me\varepsilon} = \frac{z_5}{z_2}, \\[8pt]
P_{m\varepsilon}^{\varepsilon ee} = \frac{1}{z_2}, & P_{1e}^{eee} = \frac{1}{z_5}, & P_{\varepsilon m}^{mee} = \frac{1}{z_8}, & Q_{m\varepsilon}^{\varepsilon ee} = \frac{z_8}{z_5}, & Q_{1e}^{eee} = \frac{1}{z_5}, & Q_{\varepsilon m}^{mee} = \frac{z_2}{z_5}, \\[8pt]
P_{m1}^{\varepsilon em} = \frac{z_1}{z_3}, & P_{1m}^{eem} = \frac{z_4}{z_6}, & P_{ee}^{mem} = \frac{z_7}{z_9}, & Q_{m1}^{\varepsilon em} = \frac{z_5}{z_8}, & Q_{1m}^{eem} = \frac{z_2}{z_8}, & Q_{ee}^{mem} = \frac{1}{z_8}, \\[8pt]
P_{em}^{\varepsilon m\varepsilon} = \frac{z_2}{z_1}, & P_{\varepsilon1}^{em\varepsilon} = \frac{z_5}{z_4}, & P_{1\varepsilon}^{mm\varepsilon} = \frac{z_8}{z_7}, & Q_{em}^{\varepsilon m\varepsilon} = \frac{1}{z_3}, & Q_{\varepsilon1}^{em\varepsilon} = \frac{z_9}{z_3}, & Q_{1\varepsilon}^{mm\varepsilon} = \frac{z_6}{z_3}, \\[8pt]
P_{e1}^{\varepsilon me} = \frac{z_1}{z_2}, & P_{\varepsilon m}^{eme} = \frac{z_4}{z_5}, & P_{1e}^{mme} = \frac{z_7}{z_8}, & Q_{e1}^{\varepsilon me} = \frac{z_9}{z_6}, & Q_{\varepsilon m}^{eme} = \frac{1}{z_6}, & Q_{1e}^{mme} = \frac{z_3}{z_6}, \\[8pt]
P_{ee}^{\varepsilon mm} = \frac{1}{z_3}, & P_{\varepsilon e}^{emm} = \frac{1}{z_6}, & P_{1m}^{mmm} = \frac{1}{z_9}, & Q_{ee}^{\varepsilon mm} = \frac{z_6}{z_9}, & Q_{\varepsilon e}^{emm} = \frac{z_3}{z_9}, & Q_{1m}^{mmm} = \frac{1}{z_9}.
\end{array}
$$

We can observe some interesting features in this table, namely when all of the particles are of the same type we find $P^{aaa} = Q^{aaa}$.

### H.2.1 Four particles

For four particles we have the following solutions.

|  | Value of $M$ |  |  |  |  | Value of $M$ |  |  |  |  | Value of $M$ |  |  |  |
|---|---|---|---|---|---|---|---|---|---|---|---|---|---|---|
|  | $A$ | $B$ | $X$ | $Y$ |  | $A$ | $B$ | $X$ | $Y$ |  | $A$ | $B$ | $X$ | $Y$ |
| $M_{1\varepsilon1}^{\varepsilon\varepsilon\varepsilon\varepsilon}$ | $z_1$ | $z_1$ | $z_1$ | $z_1$ | $M_{mem}^{eeee}$ | $\frac{z_1}{z_7}$ | $\frac{z_1}{z_7}$ | $z_4$ | $z_4$ | $M_{eme}^{m\varepsilon\varepsilon\varepsilon}$ | $\frac{z_1}{z_4}$ | $\frac{z_1}{z_4}$ | $z_7$ | $z_7$ |
| $M_{1\varepsilon m}^{\varepsilon\varepsilon\varepsilon e}$ | $\frac{z_2}{z_3}$ | $\frac{z_4}{z_7}$ | $\frac{z_2}{z_3}$ | $\frac{z_4}{z_7}$ | $M_{me1}^{eeee}$ | $\frac{z_2 z_9}{z_3 z_8}$ | $z_4$ | $\frac{z_5}{z_6}$ | $\frac{z_1}{z_7}$ | $M_{eme}^{m\varepsilon\varepsilon e}$ | $\frac{z_2 z_6}{z_3 z_5}$ | $\frac{z_4}{z_1}$ | $\frac{z_8}{z_9}$ | $\frac{1}{z_7}$ |
| $M_{1\varepsilon\varepsilon}^{\varepsilon\varepsilon\varepsilon m}$ | $\frac{z_3}{z_2}$ | $\frac{z_7}{z_4}$ | $\frac{z_3}{z_2}$ | $\frac{z_7}{z_4}$ | $M_{me\varepsilon}^{eeem}$ | $\frac{z_3 z_8}{z_2 z_9}$ | $\frac{z_7}{z_1}$ | $\frac{z_6}{z_5}$ | $\frac{1}{z_4}$ | $M_{em1}^{m\varepsilon em}$ | $\frac{z_3 z_5}{z_2 z_6}$ | $z_7$ | $\frac{z_9}{z_8}$ | $\frac{z_1}{z_4}$ |
| $M_{1em}^{\varepsilon\varepsilon\varepsilon\varepsilon}$ | $\frac{z_4}{z_7}$ | $\frac{z_2}{z_3}$ | $\frac{z_2}{z_3}$ | $\frac{z_4}{z_7}$ | $M_{me1}^{eeee}$ | $z_4$ | $\frac{z_2 z_9}{z_3 z_8}$ | $\frac{z_5}{z_6}$ | $\frac{z_1}{z_7}$ | $M_{e1\varepsilon}^{m\varepsilon\varepsilon\varepsilon}$ | $\frac{z_4}{z_1}$ | $\frac{z_2 z_6}{z_3 z_5}$ | $\frac{z_8}{z_9}$ | $\frac{1}{z_7}$ |
| $M_{1e1}^{\varepsilon\varepsilon\varepsilon\varepsilon}$ | $\frac{z_5 z_9}{z_6 z_8}$ | $\frac{z_5 z_9}{z_6 z_8}$ | $z_1$ | $z_1$ | $M_{mem}^{eeee}$ | $\frac{z_5}{z_6}$ | $\frac{z_5}{z_6}$ | $z_4$ | $z_4$ | $M_{e1e}^{m\varepsilon\varepsilon e}$ | $\frac{z_3 z_5}{z_2 z_6}$ | $\frac{z_3 z_5}{z_2 z_6}$ | $z_7$ | $z_7$ |
| $M_{1e\varepsilon}^{\varepsilon\varepsilon\varepsilon m}$ | $\frac{z_6 z_8}{z_5 z_9}$ | $\frac{z_6 z_8}{z_5 z_9}$ | $\frac{1}{z_1}$ | $\frac{1}{z_1}$ | $M_{me\varepsilon}^{eeem}$ | $\frac{z_6}{z_5}$ | $\frac{z_3 z_8}{z_2 z_9}$ | $\frac{1}{z_4}$ | $\frac{z_7}{z_1}$ | $M_{e1m}^{m\varepsilon em}$ | $\frac{z_2 z_6}{z_3 z_5}$ | $\frac{z_8}{z_9}$ | $\frac{1}{z_7}$ | $\frac{z_4}{z_1}$ |
| $M_{1me}^{\varepsilon\varepsilon m\varepsilon}$ | $\frac{z_7}{z_4}$ | $\frac{z_3}{z_2}$ | $\frac{z_3}{z_2}$ | $\frac{z_7}{z_4}$ | $M_{m1\varepsilon}^{eeme}$ | $\frac{z_7}{z_1}$ | $\frac{z_3 z_8}{z_2 z_9}$ | $\frac{z_6}{z_5}$ | $\frac{1}{z_4}$ | $M_{e\varepsilon1}^{m\varepsilon m\varepsilon}$ | $z_7$ | $\frac{z_3 z_5}{z_2 z_6}$ | $\frac{z_9}{z_8}$ | $\frac{z_1}{z_4}$ |
| $M_{1m\varepsilon}^{\varepsilon\varepsilon me}$ | $\frac{z_6 z_8}{z_5 z_9}$ | $\frac{z_6 z_8}{z_5 z_9}$ | $\frac{1}{z_1}$ | $\frac{1}{z_1}$ | $M_{m1e}^{eeme}$ | $\frac{z_3 z_8}{z_2 z_9}$ | $\frac{z_6}{z_5}$ | $\frac{1}{z_4}$ | $\frac{z_7}{z_1}$ | $M_{eem}^{meme}$ | $\frac{z_8}{z_9}$ | $\frac{z_2 z_6}{z_3 z_5}$ | $\frac{1}{z_7}$ | $\frac{z_4}{z_1}$ |
| $M_{1m1}^{\varepsilon\varepsilon mm}$ | $\frac{z_5 z_9}{z_6 z_8}$ | $\frac{z_5 z_9}{z_6 z_8}$ | $z_1$ | $z_1$ | $M_{m1m}^{eemm}$ | $\frac{z_2 z_9}{z_3 z_8}$ | $\frac{z_2 z_9}{z_3 z_8}$ | $z_4$ | $z_4$ | $M_{e\varepsilon e}^{memm}$ | $\frac{z_9}{z_8}$ | $\frac{z_9}{z_8}$ | $z_7$ | $z_7$ |
| $M_{mem}^{\varepsilon\varepsilon\varepsilon\varepsilon}$ | $\frac{z_1}{z_3}$ | $\frac{z_1}{z_3}$ | $z_2$ | $z_2$ | $M_{1\varepsilon1}^{eeee}$ | $\frac{z_1 z_9}{z_3 z_7}$ | $\frac{z_1 z_9}{z_3 z_7}$ | $z_5$ | $z_5$ | $M_{\varepsilon1\varepsilon}^{me\varepsilon\varepsilon}$ | $\frac{z_1 z_6}{z_3 z_4}$ | $\frac{z_1 z_6}{z_3 z_4}$ | $z_8$ | $z_8$ |
| $M_{me1}^{\varepsilon\varepsilon\varepsilon\varepsilon}$ | $z_2$ | $\frac{z_4 z_9}{z_6 z_7}$ | $\frac{z_1}{z_3}$ | $\frac{z_5}{z_8}$ | $M_{1\varepsilon m}^{eeee}$ | $\frac{z_2}{z_8}$ | $\frac{z_4}{z_6}$ | $\frac{z_4}{z_6}$ | $\frac{z_2}{z_8}$ | $M_{\varepsilon1e}^{me\varepsilon e}$ | $\frac{z_2}{z_5}$ | $\frac{z_3 z_4}{z_1 z_6}$ | $\frac{z_7}{z_9}$ | $\frac{1}{z_8}$ |
| $M_{me\varepsilon}^{\varepsilon\varepsilon\varepsilon m}$ | $\frac{z_3}{z_1}$ | $\frac{z_6 z_7}{z_4 z_9}$ | $\frac{1}{z_2}$ | $\frac{z_8}{z_5}$ | $M_{1\varepsilon\varepsilon}^{eeem}$ | $\frac{z_3 z_7}{z_1 z_9}$ | $\frac{z_3 z_7}{z_1 z_9}$ | $\frac{1}{z_5}$ | $\frac{1}{z_5}$ | $M_{\varepsilon1m}^{meem}$ | $\frac{z_3 z_4}{z_1 z_6}$ | $\frac{z_7}{z_9}$ | $\frac{1}{z_8}$ | $\frac{z_2}{z_5}$ |
| $M_{me1}^{\varepsilon\varepsilon\varepsilon\varepsilon}$ | $\frac{z_4 z_9}{z_6 z_7}$ | $z_2$ | $\frac{z_1}{z_3}$ | $\frac{z_5}{z_8}$ | $M_{1em}^{eeee}$ | $\frac{z_4}{z_6}$ | $\frac{z_2}{z_8}$ | $\frac{z_4}{z_6}$ | $\frac{z_2}{z_8}$ | $M_{eme}^{me\varepsilon\varepsilon}$ | $\frac{z_3 z_4}{z_1 z_6}$ | $\frac{z_2}{z_5}$ | $\frac{z_7}{z_9}$ | $\frac{1}{z_8}$ |
| $M_{mem}^{\varepsilon\varepsilon\varepsilon e}$ | $\frac{z_5}{z_8}$ | $\frac{z_5}{z_8}$ | $z_2$ | $z_2$ | $M_{1e1}^{eeee}$ | $z_5$ | $z_5$ | $z_5$ | $z_5$ | $M_{\varepsilon m\varepsilon}^{meee}$ | $\frac{z_5}{z_2}$ | $\frac{z_5}{z_2}$ | $z_8$ | $z_8$ |
| $M_{me\varepsilon}^{\varepsilon\varepsilon\varepsilon m}$ | $\frac{z_6 z_7}{z_4 z_9}$ | $\frac{z_8}{z_5}$ | $\frac{z_3}{z_1}$ | $\frac{1}{z_2}$ | $M_{1e\varepsilon}^{eeem}$ | $\frac{z_6}{z_4}$ | $\frac{z_8}{z_2}$ | $\frac{z_6}{z_4}$ | $\frac{z_8}{z_2}$ | $M_{em1}^{meem}$ | $\frac{z_1 z_6}{z_3 z_4}$ | $z_8$ | $\frac{z_9}{z_7}$ | $\frac{z_5}{z_2}$ |
| $M_{m1\varepsilon}^{\varepsilon\varepsilon m\varepsilon}$ | $\frac{z_6 z_7}{z_4 z_9}$ | $\frac{z_3}{z_1}$ | $\frac{1}{z_2}$ | $\frac{z_8}{z_5}$ | $M_{1me}^{eeme}$ | $\frac{z_3 z_7}{z_1 z_9}$ | $\frac{z_3 z_7}{z_1 z_9}$ | $\frac{1}{z_5}$ | $\frac{1}{z_5}$ | $M_{eem}^{mem\varepsilon}$ | $\frac{z_7}{z_9}$ | $\frac{z_3 z_4}{z_1 z_6}$ | $\frac{1}{z_8}$ | $\frac{z_2}{z_5}$ |
| $M_{m1e}^{\varepsilon\varepsilon me}$ | $\frac{z_8}{z_5}$ | $\frac{z_6 z_7}{z_4 z_9}$ | $\frac{z_3}{z_1}$ | $\frac{1}{z_2}$ | $M_{1m\varepsilon}^{eeme}$ | $\frac{z_8}{z_2}$ | $\frac{z_6}{z_4}$ | $\frac{z_6}{z_4}$ | $\frac{z_8}{z_2}$ | $M_{ee1}^{meme}$ | $z_8$ | $\frac{z_1 z_6}{z_3 z_4}$ | $\frac{z_9}{z_7}$ | $\frac{z_5}{z_2}$ |
| $M_{m1m}^{\varepsilon\varepsilon mm}$ | $\frac{z_4 z_9}{z_6 z_7}$ | $\frac{z_4 z_9}{z_6 z_7}$ | $z_2$ | $z_2$ | $M_{1m1}^{eemm}$ | $\frac{z_1 z_9}{z_3 z_7}$ | $\frac{z_1 z_9}{z_3 z_7}$ | $z_5$ | $z_5$ | $M_{e\varepsilon\varepsilon}^{memm}$ | $\frac{z_9}{z_7}$ | $\frac{z_9}{z_7}$ | $z_8$ | $z_8$ |
| $M_{eme}^{\varepsilon m\varepsilon\varepsilon}$ | $\frac{z_1}{z_2}$ | $\frac{z_1}{z_2}$ | $z_3$ | $z_3$ | $M_{\varepsilon1\varepsilon}^{em\varepsilon\varepsilon}$ | $\frac{z_1 z_8}{z_2 z_7}$ | $\frac{z_1 z_8}{z_2 z_7}$ | $z_6$ | $z_6$ | $M_{1\varepsilon1}^{mm\varepsilon\varepsilon}$ | $\frac{z_1 z_5}{z_2 z_4}$ | $\frac{z_1 z_5}{z_2 z_4}$ | $z_9$ | $z_9$ |
| $M_{eme}^{\varepsilon mee}$ | $\frac{z_2}{z_1}$ | $\frac{z_4 z_8}{z_5 z_7}$ | $\frac{1}{z_3}$ | $\frac{z_6}{z_9}$ | $M_{\varepsilon1e}^{emee}$ | $\frac{z_2 z_7}{z_1 z_8}$ | $\frac{z_4}{z_5}$ | $\frac{1}{z_6}$ | $\frac{z_3}{z_9}$ | $M_{1em}^{mm\varepsilon e}$ | $\frac{z_2 z_4}{z_1 z_5}$ | $\frac{z_2 z_4}{z_1 z_5}$ | $\frac{1}{z_9}$ | $\frac{1}{z_9}$ |
| $M_{em1}^{\varepsilon mem}$ | $z_3$ | $\frac{z_5 z_7}{z_4 z_8}$ | $\frac{z_1}{z_2}$ | $\frac{z_9}{z_6}$ | $M_{\varepsilon1m}^{emem}$ | $\frac{z_3}{z_9}$ | $\frac{z_2 z_7}{z_1 z_8}$ | $\frac{z_4}{z_5}$ | $\frac{1}{z_6}$ | $M_{1\varepsilon\varepsilon}^{mmem}$ | $\frac{z_3}{z_6}$ | $\frac{z_7}{z_8}$ | $\frac{z_7}{z_8}$ | $\frac{z_3}{z_6}$ |
| $M_{e1\varepsilon}^{\varepsilon m\varepsilon\varepsilon}$ | $\frac{z_4 z_8}{z_5 z_7}$ | $\frac{z_2}{z_1}$ | $\frac{1}{z_3}$ | $\frac{z_6}{z_9}$ | $M_{eme}^{emee}$ | $\frac{z_4}{z_5}$ | $\frac{z_2 z_7}{z_1 z_8}$ | $\frac{1}{z_6}$ | $\frac{z_3}{z_9}$ | $M_{1em}^{mmee}$ | $\frac{z_2 z_4}{z_1 z_5}$ | $\frac{z_2 z_4}{z_1 z_5}$ | $\frac{1}{z_9}$ | $\frac{1}{z_9}$ |
| $M_{e1e}^{\varepsilon mee}$ | $\frac{z_5 z_7}{z_4 z_8}$ | $\frac{z_5 z_7}{z_4 z_8}$ | $z_3$ | $z_3$ | $M_{\varepsilon m\varepsilon}^{emee}$ | $\frac{z_5}{z_4}$ | $\frac{z_5}{z_4}$ | $z_6$ | $z_6$ | $M_{1e1}^{mmee}$ | $\frac{z_1 z_5}{z_2 z_4}$ | $\frac{z_1 z_5}{z_2 z_4}$ | $z_9$ | $z_9$ |
| $M_{e1m}^{\varepsilon mem}$ | $\frac{z_6}{z_9}$ | $\frac{z_4 z_8}{z_5 z_7}$ | $\frac{z_2}{z_1}$ | $\frac{1}{z_3}$ | $M_{em1}^{emem}$ | $z_6$ | $\frac{z_1 z_8}{z_2 z_7}$ | $\frac{z_5}{z_4}$ | $\frac{z_9}{z_3}$ | $M_{1e\varepsilon}^{mmem}$ | $\frac{z_6}{z_3}$ | $\frac{z_8}{z_7}$ | $\frac{z_8}{z_7}$ | $\frac{z_6}{z_3}$ |
| $M_{ee1}^{\varepsilon mm\varepsilon}$ | $\frac{z_5 z_7}{z_4 z_8}$ | $z_3$ | $\frac{z_1}{z_2}$ | $\frac{z_9}{z_6}$ | $M_{\varepsilon em}^{emm\varepsilon}$ | $\frac{z_2 z_7}{z_1 z_8}$ | $\frac{z_3}{z_9}$ | $\frac{z_4}{z_5}$ | $\frac{1}{z_6}$ | $M_{1me}^{mmm\varepsilon}$ | $\frac{z_7}{z_8}$ | $\frac{z_3}{z_6}$ | $\frac{z_7}{z_8}$ | $\frac{z_3}{z_6}$ |
| $M_{eem}^{\varepsilon mme}$ | $\frac{z_4 z_8}{z_5 z_7}$ | $\frac{z_6}{z_9}$ | $\frac{z_2}{z_1}$ | $\frac{1}{z_3}$ | $M_{\varepsilon e1}^{emme}$ | $\frac{z_1 z_8}{z_2 z_7}$ | $z_6$ | $\frac{z_5}{z_4}$ | $\frac{z_9}{z_3}$ | $M_{1m\varepsilon}^{mmme}$ | $\frac{z_8}{z_7}$ | $\frac{z_6}{z_3}$ | $\frac{z_8}{z_7}$ | $\frac{z_6}{z_3}$ |
| $M_{ee\varepsilon}^{\varepsilon mmm}$ | $\frac{z_9}{z_6}$ | $\frac{z_9}{z_6}$ | $z_3$ | $z_3$ | $M_{\varepsilon e\varepsilon}^{emmm}$ | $\frac{z_9}{z_3}$ | $\frac{z_9}{z_3}$ | $z_6$ | $z_6$ | $M_{1m1}^{mmmm}$ | $z_9$ | $z_9$ | $z_9$ | $z_9$ |

We can notice here again, when all of the particles are the same type the graph braid symbols are equal, i.e.
$X^{aaaa} = Y^{aaaa} = A^{aaaa} = B^{aaaa}$.

### H.3 Solutions to the lollipop equations

### H.3.1 Lollipop trijunction solutions

In contrast to the Ising model, demanding that $P_{ed}^{abc} \equiv R_e^{ab}$ does not necessarily imply that the solutions must be planar. There are 32 solutions in total which can be presented as follows:

$$R_1^{22} = v_1, R_4^{23} = v_2, R_3^{24} = v_1 v_2, R_4^{32} = v_3, R_1^{33} = v_4, R_2^{34} = v_3 v_4, R_3^{42} = -1, R_2^{43} = v_5, R_1^{44} = -v_5$$

and

| | Value of $M$ | | | Value of $M$ | | | Value of $M$ | |
|---|---|---|---|---|---|---|---|---|
| | $P$ | $Q$ | | $P$ | $Q$ | | $P$ | $Q$ |
| $M_{1\varepsilon}^{\varepsilon\varepsilon\varepsilon}$ | $v_1$ | $v_1$ | $M_{me}^{\varepsilon\varepsilon\varepsilon}$ | $v_3$ | $-v_1$ | $M_{em}^{m\varepsilon\varepsilon}$ | $-1$ | $v_1 v_3$ |
| $M_{1e}^{\varepsilon\varepsilon\varepsilon}$ | $v_1$ | $-v_3$ | $M_{m\varepsilon}^{\varepsilon\varepsilon\varepsilon}$ | $v_3$ | $v_3$ | $M_{e1}^{m\varepsilon\varepsilon}$ | $-1$ | $v_1 v_3$ |
| $M_{1m}^{\varepsilon\varepsilon m}$ | $v_1$ | $-v_3$ | $M_{m1}^{\varepsilon\varepsilon m}$ | $v_3$ | $-v_1$ | $M_{e\varepsilon}^{m\varepsilon m}$ | $-1$ | $-1$ |
| $M_{me}^{\varepsilon\varepsilon\varepsilon}$ | $v_2$ | $v_2$ | $M_{1\varepsilon}^{\varepsilon\varepsilon\varepsilon}$ | $v_4$ | $v_2 v_5$ | $M_{\varepsilon 1}^{m\varepsilon\varepsilon}$ | $v_5$ | $v_2 v_4$ |
| $M_{m\varepsilon}^{\varepsilon\varepsilon\varepsilon}$ | $v_2$ | $v_4 v_5$ | $M_{1e}^{\varepsilon\varepsilon\varepsilon}$ | $v_4$ | $v_4$ | $M_{\varepsilon m}^{m\varepsilon\varepsilon}$ | $v_5$ | $v_2 v_4$ |
| $M_{m1}^{\varepsilon\varepsilon m}$ | $v_2$ | $v_4 v_5$ | $M_{1m}^{\varepsilon\varepsilon m}$ | $v_4$ | $v_2 v_5$ | $M_{\varepsilon e}^{m\varepsilon m}$ | $v_5$ | $v_5$ |
| $M_{em}^{\varepsilon m\varepsilon}$ | $v_1 v_2$ | $v_1 v_2$ | $M_{\varepsilon 1}^{\varepsilon m\varepsilon}$ | $v_3 v_4$ | $-v_1 v_2 v_5$ | $M_{1\varepsilon}^{mm\varepsilon}$ | $-v_5$ | $v_1 v_2 v_3 v_4$ |
| $M_{e1}^{\varepsilon m\varepsilon}$ | $v_1 v_2$ | $-v_3 v_4 v_5$ | $M_{\varepsilon m}^{\varepsilon m\varepsilon}$ | $v_3 v_4$ | $v_3 v_4$ | $M_{1e}^{mm\varepsilon}$ | $-v_5$ | $v_1 v_2 v_3 v_4$ |
| $M_{e\varepsilon}^{\varepsilon mm}$ | $v_1 v_2$ | $-v_3 v_4 v_5$ | $M_{\varepsilon e}^{\varepsilon mm}$ | $v_3 v_4$ | $-v_1 v_2 v_5$ | $M_{1m}^{mmm}$ | $-v_5$ | $-v_5$ |

(85)

where $v_i \in \{-1, 1\}$. Demanding planarity then comes down to demanding that $v_1 = -v_3$ and $v_2 = v_4 v_5$. The loss of two binary degrees of freedom thus implies that only one out of four solutions are planar. We find here again $P^{aaa} = Q^{aaa}$.

### H.3.2 Circle solutions

There are 128 solutions to the circle equations. They can be presented as follows:

$$D_\varepsilon^{1\varepsilon} = \mu_1, \qquad D_e^{1e} = \mu_2, \quad D_m^{1m} = \mu_3, \qquad D_1^{\varepsilon\varepsilon} = \mu_4, \qquad D_m^{\varepsilon e} = \mu_5, \quad D_e^{\varepsilon m} = \mu_6,$$
$$D_m^{e\varepsilon} = \mu_3 \mu_5, \quad D_1^{ee} = \mu_7, \quad D_\varepsilon^{em} = -\mu_1, \quad D_e^{m\varepsilon} = \mu_2 \mu_6, \quad D_\varepsilon^{me} = -1, \quad D_1^{mm} = \mu_4 \mu_7,$$

(86)

where $\mu_i \in \{-1, 1\}$. The twist factors are the same as in the planar case.

### H.3.3 Full lollipop solutions

In contrast to the Ising model, after fixing the $F$-symbols, there is a discrete $\mathbb{Z}_2$ gauge symmetry left that has the following form:

$$M_m^{\varepsilon e} \mapsto -M_m^{\varepsilon e}, \qquad M_e^{\varepsilon m} \mapsto -M_e^{\varepsilon m}, \qquad M_m^{e\varepsilon} \mapsto -M_m^{e\varepsilon},$$
$$M_\varepsilon^{em} \mapsto -M_\varepsilon^{em}, \qquad M_e^{m\varepsilon} \mapsto -M_e^{m\varepsilon}, \qquad M_\varepsilon^{me} \mapsto -M_\varepsilon^{me},$$

for $M = R$ and $M = D$, and

$$M_{md}^{\varepsilon ec} \mapsto -M_{md}^{\varepsilon ec}, \qquad M_{ed}^{\varepsilon mc} \mapsto -M_{ed}^{\varepsilon mc}, \qquad M_{md}^{e\varepsilon c} \mapsto -M_{md}^{e\varepsilon c},$$
$$M_{\varepsilon d}^{emc} \mapsto -M_{\varepsilon d}^{emc}, \qquad M_{ed}^{m\varepsilon c} \mapsto -M_{ed}^{m\varepsilon c}, \qquad M_{\varepsilon d}^{mec} \mapsto -M_{\varepsilon d}^{mec},$$

for $M = P$ and $M = Q$. For the solutions to the lollipop trijunction equations and the circle equations, described in sections H.3.1 and H.3.2, this gauge symmetry has been removed. To construct the full solution set to the lollipop equations one should therefore re-introduce these gauge equivalent solutions, construct all products between solutions to the trijunction lollipop equations and circle equations, and finally remove this gauge

symmetry again. This set has twice the size of the product set of gauge-inequivalent solutions. It can easily be constructed by taking the product of the lollipop trijunction solutions and the following set of non-reduced solutions to the circle equations

$$D_\varepsilon^{1\varepsilon} = \mu_1, \qquad D_e^{1e} = \mu_2, \quad D_m^{1m} = \mu_3, \qquad D_1^{\varepsilon\varepsilon} = \mu_4, \qquad D_m^{\varepsilon e} = \sigma\mu_5, \quad D_e^{\varepsilon m} = \sigma\mu_6,$$

$$D_m^{e\varepsilon} = \sigma\mu_3\mu_5, \quad D_1^{ee} = \mu_7, \quad D_\varepsilon^{em} = -\sigma\mu_1, \quad D_e^{m\varepsilon} = \sigma\mu_2\mu_6, \quad D_\varepsilon^{me} = -\sigma, \quad D_1^{mm} = \mu_4\mu_7, \tag{87}$$

where $\sigma \in \{-1, 1\}$ reintroduces the $\mathbb{Z}_2$ gauge freedom.

# I  Solutions for the TY($\mathbb{Z}_3$) model

The TY($\mathbb{Z}_3$) fusion ring has 4 particles, $1, \psi_1, \psi_2, \sigma$, where $\{1, \psi_1, \psi_2\}$ form a $\mathbb{Z}_3$ subgroup, and

$$1 \times a \;=\; a \times 1 = a, \quad \forall a \in \{1, \psi_1, \psi_2, \sigma\}, \tag{88}$$

$$\sigma \times b \;=\; b \times \sigma = \sigma, \quad \forall a \in \{1, \psi_1, \psi_2\}, \tag{89}$$

$$\sigma \times \sigma \;=\; 1 + \psi_1 + \psi_2. \tag{90}$$

In the following sections we will list the solutions to the pentagon equations as well as the circle equations. All other equations admit no solutions. We omit any well-defined symbol equal to 1.

## I.1  Solutions to the pentagon equations

There are four solutions to the pentagon equations which can be presented as follows:

$$[F_{\psi_1}^{\sigma\sigma\psi_1}]_{1\sigma} = \kappa_1, \qquad [F_{\psi_2}^{\sigma\sigma\psi_2}]_{1\sigma} = \kappa_1, \qquad [F_{\psi_1}^{\sigma\psi_1\sigma}]_{\sigma\sigma} = e^{-\frac{2}{3}i\pi\kappa_1\kappa_2}, \quad [F_{\psi_2}^{\sigma\psi_1\sigma}]_{\sigma\sigma} = e^{\frac{2}{3}i\pi\kappa_1\kappa_2},$$

$$[F_\sigma^{\sigma\psi_1\psi_2}]_{\sigma 1} = \kappa_1, \qquad [F_{\psi_1}^{\sigma\psi_2\sigma}]_{\sigma\sigma} = e^{\frac{2}{3}i\pi\kappa_1\kappa_2}, \quad [F_{\psi_2}^{\sigma\psi_2\sigma}]_{\sigma\sigma} = e^{-\frac{2}{3}i\pi\kappa_1\kappa_2}, \quad [F_\sigma^{\sigma\psi_2\psi_1}]_{\sigma 1} = \kappa_1,$$

$$[F_{\psi_1}^{\psi_1\sigma\sigma}]_{\sigma 1} = \kappa_1, \qquad [F_\sigma^{\psi_1\sigma\psi_1}]_{\sigma\sigma} = e^{-\frac{2}{3}i\pi\kappa_1\kappa_2}, \quad [F_\sigma^{\psi_1\sigma\psi_2}]_{\sigma\sigma} = e^{\frac{2}{3}i\pi\kappa_1\kappa_2}, \quad [F_{\psi_1}^{\psi_1\psi_1\psi_2}]_{\psi_2 1} = \kappa_1,$$

$$[F_\sigma^{\psi_1\psi_2\sigma}]_{1\sigma} = \kappa_1, \qquad [F_{\psi_2}^{\psi_1\psi_2\psi_2}]_{1\psi_1} = \kappa_1, \qquad [F_{\psi_2}^{\psi_2\sigma\sigma}]_{\sigma 1} = \kappa_1, \qquad [F_\sigma^{\psi_2\sigma\psi_1}]_{\sigma\sigma} = e^{\frac{2}{3}i\pi\kappa_1\kappa_2},$$

$$[F_\sigma^{\psi_2\sigma\psi_2}]_{\sigma\sigma} = e^{-\frac{2}{3}i\pi\kappa_1\kappa_2}, \quad [F_\sigma^{\psi_2\psi_1\sigma}]_{1\sigma} = \kappa_1, \qquad [F_{\psi_1}^{\psi_2\psi_1\psi_1}]_{1\psi_2} = \kappa_1, \qquad [F_{\psi_2}^{\psi_2\psi_2\psi_1}]_{\psi_1 1} = \kappa_1,$$

$$\left[F_\sigma^{\sigma\sigma\sigma}\right] = \frac{1}{\sqrt{3}} \begin{bmatrix} \kappa_1 & 1 & 1 \\ 1 & e^{i\pi(\frac{\kappa_1}{6}+\frac{1}{2})\kappa_2} & e^{-i\pi(\frac{\kappa_1}{6}+\frac{1}{2})\kappa_2} \\ 1 & e^{-i\pi(\frac{\kappa_1}{6}+\frac{1}{2})\kappa_2} & e^{i\pi(\frac{\kappa_1}{6}+\frac{1}{2})\kappa_2} \end{bmatrix},$$

where $\kappa_1, \kappa_2 \in \{-1, 1\}$ and the matrix indices of $\left[F_\sigma^{\sigma\sigma\sigma}\right]$ range over $(1, \psi_1, \psi_2)$.

## I.2  Solutions to the circle equations

In contrast to the planar hexagon equations, we now find there are 48 solutions, per set of $F$-symbols, to the circle equations. Let $\varepsilon_i \in \{-1, 1\}$ and $\nu \in \{0, 1, 2\}$, then they can be presented as follows. If $(\kappa_1, \kappa_2) = (-1, -1)$ then

$$D_\sigma^{1\sigma} = \varepsilon_1 e^{\frac{i\pi}{12}(7-2\nu(\nu+1))}, \quad D_{\psi_1}^{1\psi_1} = e^{-\frac{2i\pi}{3}}, \qquad\qquad D_{\psi_2}^{1\psi_2} = e^{-\frac{2i\pi}{3}},$$

$$D_1^{\sigma\sigma} = \varepsilon_2, \qquad\qquad D_{\psi_1}^{\sigma\sigma} = e^{i\pi(\frac{\varepsilon_3}{2}+\frac{1}{6})}, \qquad\qquad D_{\psi_2}^{\sigma\sigma} = e^{i\pi(\frac{\varepsilon_4}{2}+\frac{1}{6})},$$

$$D_\sigma^{\sigma\psi_1} = e^{\frac{2i\pi}{3}((\nu-1)^2\varepsilon_1-1)}, \quad D_\sigma^{\sigma\psi_2} = e^{-\frac{2i\pi}{3}((\nu-1)^2\varepsilon_1+1)}, \qquad D_\sigma^{\psi_1\sigma} = e^{-\frac{i\pi}{12}(2\nu^2+2\nu-9+2\varepsilon_1(4\nu^2-8\nu+1))},$$

$$D_{\psi_2}^{\psi_1\psi_1} = e^{\frac{2i\pi}{3}}, \qquad\qquad D_\sigma^{\psi_2\sigma} = e^{-\frac{i\pi}{12}(2\nu^2-10\nu+3+\varepsilon_1(4\nu^2-8\nu-2))}, \quad D_{\psi_1}^{\psi_2\psi_2} = e^{\frac{2i\pi}{3}}.$$

If $(\kappa_1, \kappa_2) = (-1, 1)$ then

$$D_\sigma^{1\sigma} = \varepsilon_1 e^{\frac{i\pi}{12}(5+2\nu(\nu+1))}, \quad D_{\psi_1}^{1\psi_1} = e^{\frac{2i\pi}{3}}, \qquad\qquad D_{\psi_2}^{1\psi_2} = e^{\frac{2i\pi}{3}},$$

$$D_1^{\sigma\sigma} = \varepsilon_2, \qquad\qquad D_{\psi_1}^{\sigma\sigma} = e^{-i\pi\left(\frac{1}{6}-\frac{\varepsilon_3}{2}\right)}, \qquad D_{\psi_2}^{\sigma\sigma} = e^{-i\pi\left(\frac{1}{6}-\frac{\varepsilon_4}{2}\right)},$$

$$D_\sigma^{\sigma\psi_1} = e^{\frac{2i\pi}{3}(3-2\nu)}, \qquad D_\sigma^{\sigma\psi_2} = e^{\frac{2i\pi}{3}(2\nu-1)}, \qquad D_\sigma^{\psi_1\sigma} = e^{\frac{i\pi}{12}\left(2\nu^2-6\nu-1+6\varepsilon_1\right)},$$

$$D_{\psi_2}^{\psi_1\psi_1} = e^{-\frac{2i\pi}{3}}, \qquad D_\sigma^{\psi_2\sigma} = e^{\frac{i\pi}{12}\left(2\nu^2-2\nu-5-6\varepsilon_1\left(2\nu^2-4\nu+1\right)\right)}, \quad D_{\psi_1}^{\psi_2\psi_2} = e^{-\frac{2i\pi}{3}}.$$

If $(\kappa_1, \kappa_2) = (1, -1)$ then

$$D_\sigma^{1\sigma} = \varepsilon_1 e^{-\frac{i\pi}{12}(-1+2\nu(\nu+1))}, \quad D_{\psi_1}^{1\psi_1} = e^{-\frac{2i\pi}{3}}, \qquad\qquad D_{\psi_2}^{1\psi_2} = e^{-\frac{2i\pi}{3}},$$

$$D_1^{\sigma\sigma} = \varepsilon_2, \qquad\qquad D_{\psi_1}^{\sigma\sigma} = e^{i\pi\left(\frac{\varepsilon_3}{2}+\frac{1}{6}\right)}, \qquad D_{\psi_2}^{\sigma\sigma} = e^{i\pi\left(\frac{\varepsilon_4}{2}+\frac{1}{6}\right)},$$

$$D_\sigma^{\sigma\psi_1} = e^{\frac{2i\pi}{3}\left((\nu-1)^2\varepsilon_1-1\right)}, \quad D_\sigma^{\sigma\psi_2} = e^{\frac{2i\pi}{3}\left((\nu-1)^2(-\varepsilon_1)-1\right)}, \quad D_\sigma^{\psi_1\sigma} = e^{-\frac{i\pi}{12}\left(2\nu^2-10\nu-3-\varepsilon_1\left(4\nu^2-8\nu-2\right)\right)},$$

$$D_{\psi_2}^{\psi_1\psi_1} = e^{\frac{2i\pi}{3}}, \qquad D_\sigma^{\psi_2\sigma} = e^{-\frac{i\pi}{12}\left(2\nu^2+2\nu-15-2\varepsilon_1\left(4\nu^2-8\nu+1\right)\right)}, \quad D_{\psi_1}^{\psi_2\psi_2} = e^{\frac{2i\pi}{3}}.$$

If $(\kappa_1, \kappa_2) = (1, 1)$ then

$$D_\sigma^{1\sigma} = \varepsilon_1 e^{\frac{i\pi}{12}(-1+2\nu(\nu+1))}, \quad D_{\psi_1}^{1\psi_1} = e^{\frac{2i\pi}{3}}, \qquad\qquad D_{\psi_2}^{1\psi_2} = e^{\frac{2i\pi}{3}},$$

$$D_1^{\sigma\sigma} = \varepsilon_2, \qquad\qquad D_{\psi_1}^{\sigma\sigma} = e^{-i\pi\left(\frac{1}{6}-\frac{\varepsilon_3}{2}\right)}, \qquad D_{\psi_2}^{\sigma\sigma} = e^{-i\pi\left(\frac{1}{6}-\frac{\varepsilon_4}{2}\right)},$$

$$D_\sigma^{\sigma\psi_1} = e^{\frac{2i\pi}{3}(3-2\nu)}, \qquad D_\sigma^{\sigma\psi_2} = e^{\frac{2i\pi}{3}(2\nu-1)}, \qquad D_\sigma^{\psi_1\sigma} = e^{\frac{i\pi}{12}\left(2\nu^2-6\nu+5-6\varepsilon_1\right)},$$

$$D_{\psi_2}^{\psi_1\psi_1} = e^{-\frac{2i\pi}{3}}, \qquad D_\sigma^{\psi_2\sigma} = e^{\frac{i\pi}{12}\left(2\nu^2-2\nu+1+6\varepsilon_1\left(2\nu^2-4\nu+1\right)\right)}, \quad D_{\psi_1}^{\psi_2\psi_2} = e^{-\frac{2i\pi}{3}}.$$

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
