# Peer review of "Extending the planar theory of anyons to quantum wire networks"

_SciPost Physics_

## Round 1 · Referee Report · Anonymous (Referee 1) · 2024-1-9

Strengths

1- The study of anyons on wire networks is well-motivated from a physics point of view and the authors have done some important foundational work towards building the general theory.

2- There are many examples worked out to support and illustrate the new theory.

3-This effort is original and also very timely, with the development of the graph braid group being relatively recent and there being other recent physics papers on the subject of wire networks.

Weaknesses

1- There are a few places where mathematical aspects of fusion categories versus anyon models get slightly confused, but it is nothing that can't be sorted out quickly.

2- A clear description of the graph braid group via generators and presentations for different types of graphs is central to the development of the theory but lacking.

3- A clearer delineation of the foundational assumptions of the theory (e.g. graph fusion, unitarity, compatibility of graph fusion and graph braiding) and what is not fully worked out (non-degeneracy of graph-braiding, topological twists) would improve the organization of the paper.

Report

I think this paper is a great fit for SciPost that will be ready to publish after the authors address some of the points raised in this report.

Requested changes

1- I think it is important to state that the results of the paper are for multiplicity-free anyon models somewhere in the abstract and again in the introduction, since otherwise it is not stated explicitly until Section 2.

2- I would have liked to see an explanation of why it is okay to assume that the pentagon equations for anyon on wire networks should be the same as the usual pentagon equations, either using a mathematical or physical argument. Especially since later in Section 3.1 the authors seem to argue that unlike in the planar case, not all choices of bases of the Hilbert space of states associated to a collection of anyons in a wire network are valid.

3- This is an annoying point, but typically when people call something Ocneanu rigidity they are only referring to the statement that there are only finitely many fusion categories with a given fusion rule. I realize that the authors are quoting directly from Kitaev's "exactly solvable models" paper though, so it is not necessarily to change it.

4- Since the definition of the graph braid group is so central to the theory developed by the authors I would have found it helpful to include a bit more background at the beginning of Section 3 and not just relegated to an Appendix, although I understand this is the prevailing style in paper in this journal. My bigger issue is that while the generators are clearly defined, the relations (or lack of!) are a bit obscured. For example, the fact that the three-strand braid group of a trijunction is free is not explicitly stated in the Appendix.

5- The topological twist is only defined ( and similarly Equation (6) only holds) if the category has a ribbon structure, so it could be good to mention much earlier that everything is unitary (as I'm sure is intended because the authors later mention so on page 8).

6- Several times the graph braid generators and their matrix representations are conflated, for example in the sentences before Equation 59. On this subject, my opinion is that it is not ideal that A,B,X, and Y were called A-symbols, B-symbols etc. This puts them on the same level semantically as the fundamental data of the graph-braided anyon model, like the R-symbols, Q-symbols, P-symbols etc., when the representations of the graph-braids are really just a consequence of R,Q, P, and so on.

7- In the last paragraph before Section 3 on page 8 the authors list Rep(G) as an anyon model, but this is a symmetric fusion category and not a modular fusion category. Probably you meant its Drinfeld center? But then later in Table 1 there is a UCC in the column for "Planar?" in the table of graph anyon models. But Rep(D_4) is not a planar anyon model, so I'm a bit confused what is meant.

8- In Section 3.1 there is an emphasis on picking good bases for the matrix representations of the graph braid generators. In order to characterize the image of the subgroup of the unitary group (set of quantum gates) generated by the graph braids, don't they all need to be in the same basis? Overall I found the discussion of good vs. bad choices of bases hard to follow.

9- This is only an opinion but the notation e.g. $f=b \times c$ and calling $f$ a composite anyon is a little misleading in the anyon graph braids in Figure 3, since in the Q-hexagon equations there is a sum over f. Rather $f$ is an anyon in the fusion channel of $b$ and $c$. This is again confusing in Equation 59. Later however I am worried this is important, for example at the top of page 14, where a composite anyon is used as an index for one of the graph braid generators. In the planar setting it is not valid to use composite anyons as labels of fusion trees, so it would be good to see some more explanation of what is meant by this notation.

10- Is the label in the superscript ofthe graph braid $\sigma_3^{1_d,2_c,1_b,2_a}$ supposed to match the one in Figure 5(b)?

11- An investigation of what is meant by non-degeneracy of graph-braiding seems to be missing. Depending on the authors' aims it is not so important but it could be good to mention.

12- It is not obvious to me that a given solution to the graph hexagon equations uniquely specifies a wire network anyon model. Does one need to find solutions to graph ribbon equations as well? In general I was a bit confused about the role of topological twists, independently of the underlying graph topology. This seems to be treated at the end of Section 4, where there is a definition given of topological twist and maybe partially in Appendix D but I was unclear about the big picture from the main body of the paper alone.

13- If Appendix A is meant to be a stand-alone section containing the mathematical background of the graph-braid group it may best best if it did not make any reference to anyons, since the graph braid group is a purely mathematical notion. A presentation of all generators and relations for each type of graph under consideration should be clearly stated.

14-In Appendix D it may be best to avoid the use of the word "morphism" in the description of half-twists in graph-braided anyon models, since (apart from the underlying fusion category that exists by fiat) the authors have not shown that the graph-braided anyon model described a category.

15- In Appendix E there are some sentences that allude to demanding that the F-symbols should be gauge invariant but this is not physically justified and it will rule out many interesting theories. I think this was stated unintentionally based on the content of later appendices.

  • validity: good
  • significance: high
  • originality: high
  • clarity: ok
  • formatting: good
  • grammar: excellent

Author:  Mia Conlon  on 2024-10-10  [id 4856]

(in reply to Report 1 on 2024-01-09)

We thank Referee 1 for providing a very useful critical review of our work. We will address each of their points in order and we have updated our manuscript accordingly.

1 - I think it is important to state that the results of the paper are for multiplicity-free anyon models somewhere in the abstract and again in the introduction, since otherwise it is not stated explicitly until Section

Thank you for this comment, we have amended our paper to explicitly include it in the Abstract, the Introduction and the Conclusion.

2 - I would have liked to see an explanation of why it is okay to assume that the pentagon equations for anyon on wire networks should be the same as the usual pentagon equations, either using a mathematical or physical argument. Especially since later in Section 3.1 the authors seem to argue that unlike in the planar case, not all choices of bases of the Hilbert space of states associated to a collection of anyons in a wire network are valid.

Thank you for this interesting question and one we thought about quite a bit ourselves. Firstly, note that we only implement $F$-moves while the particles are on the \textbf{same} edge and not in motion. The action of a graph braid generator takes our anyons from a fixed edge of the graph, shuttle through junctions or cycles them, then returns them back to their initial location. As the $F$-moves are only utilised when the anyons are not in motion and when they are all on the same edge, this is essentially equivalent to $F$ moves on a surface. Hopefully this clarifies why it is okay to use the same pentagon equations. Regarding Section 3.1, firstly the action of the graph braid symbols must be defined with respect to a basis of the Hilbert space (certain fusion trees). For four particles, there are two choices (since the two particles furthest from the junction, which are being exchanged, must be joined by a fusion tree), so the choice remains: what is the fusion tree of the other two particles. At first, it seems that this should make no difference, since we can always change the fusion basis using an appropriate $F$-matrix. However, it is important for the braiding-fusion theory that we are building to respect the so-called {\it no transmutation principle}. It turns out that the no transmutation principle for graph braids is most convenient to express mathematically only in certain fusion bases. We have rewritten the explanation of this fact and replaced Figure 5 with a different one which hopefully illustrates this point a bit better.

3 - This is an annoying point, but typically when people call something Ocneanu rigidity they are only referring to the statement that there are only finitely many fusion categories with a given fusion rule. I realize that the authors are quoting directly from Kitaev's "exactly solvable models" paper though, so it is not necessarily to change it.

Thank you for this comment. This was also pointed out to us by the Referee 2.

4 - Since the definition of the graph braid group is so central to the theory developed by the authors I would have found it helpful to include a bit more background at the beginning of Section 3 and not just relegated to an Appendix, although I understand this is the prevailing style in paper in this journal. My bigger issue is that while the generators are clearly defined, the relations (or lack of!) are a bit obscured. For example, the fact that the three-strand braid group of a trijunction is free is not explicitly stated in the Appendix.

This is a very fair comment. We did consider including the material from Appendix A at the start of Section 3, however we felt it interrupted the flow of the paper. We felt that the majority of our readership will be familiar with the planar theory of anyons and using our graphical diagrams for the action of the graph braid generators, be able to reach our results easier than if we dedicated a large portion of our first section of new results to mathematical theory. We have amended Section 3 to include the generators of the four strand graph braid group and explicitly show the relation they are subject to. Hopefully this may help. As a general remark, finding the relators in presentations of graph braid groups is more difficult than finding the generators. As the Referee has pointed out, there exists a universal set of geometric generators for graph braid groups (derived in the paper [B. H. An, T. Maciazek, Communications in Mathematical Physics, 384(2):1109–1140, 2021]). In order to find the {\bf{complete}} set of relators for a given graph and a given number of particles, one needs to run the general algorithm from the paper [L. Sabalka, D. Farley Alg \& Geom Topology 5:1075-1109, 2005] which relies on the discrete Morse theory. The computational complexity of this algorithm grows very fast with the size of the graph and the number of particles. Fortunately, in order to derive the results presented in our paper, it suffices to use a sub-complete set of relators. Such relators are analysed case-by-case in our paper and are proved pictorially. This way, we hope to keep the results more accessible to people who are not familiar with the discrete Morse theory. In the case of the $H$-graph, we have listed all the $n=3$ and $n=4$ relators. The complete set of relators for the $T$-junction has been listed in Appendix C (these are just the pseudocommutative relations). The compete set of relations for higher-order junctions (having more than three legs) can be found in Proposition 2 of the paper [B. H. An, T. Maciazek, Communications in Mathematical Physics, 384(2):1109–1140, 2021].

5 - The topological twist is only defined ( and similarly Equation (6) only holds) if the category has a ribbon structure, so it could be good to mention much earlier that everything is unitary (as I'm sure is intended because the authors later mention so on page 8).

Thank you for this comment - this is something we should have been more clear about earlier and throughout our paper. We have amended our manuscript to mention in the Introduction and Abstract that we are specifically interested in taking unitary solutions to the pentagon equation.

6 - Several times the graph braid generators and their matrix representations are conflated, for example in the sentences before Equation 59. On this subject, my opinion is that it is not ideal that A,B,X, and Y were called A-symbols, B-symbols etc. This puts them on the same level semantically as the fundamental data of the graph-braided anyon model, like the R-symbols, Q-symbols, P-symbols etc., when the representations of the graph-braids are really just a consequence of R,Q, P, and so on.

Thank you for alerting us to this. We have tried to consistently use a capitalised latin letter, i.e. A,B,X,P,Q,R to represent the matrix elements of the braiding operator corresponding to a graph braid on the Hilbert space and linguistically this is what we refer to as a "symbol", following the parlance from the planar theory. However you do raise a good point. In essence we are saying in Section 3 that an $X$ symbol, i.e. $X^{bacd}{jge} = \rho(\sigma)$}^{(1_{b},1_{a},1_{c},2_{d}), is really expressible as a $P$-symbol, via Equation (20) and therefore this is not a new "symbol".

7 - In the last paragraph before Section 3 on page 8 the authors list Rep(G) as an anyon model, but this is a symmetric fusion category and not a modular fusion category. Probably you meant its Drinfeld center? But then later in Table 1 there is a UCC in the column for "Planar?" in the table of graph anyon models. But Rep($D_4$) is not a planar anyon model, so I'm a bit confused what is meant.

Sorry for the ambiguity, we do mean the representation ring. It is a symmetric fusion category with respect to planar braiding, however we were solving our graph braiding hexagon equations for this fusion category and there no reason a priori it would have trivial solutions on a graph. Note that, indeed, we did not demand modularity in our definition of an anyon model. One of the reasons is because the definition of an S-matrix uses planar R-symbols and it is not, a priori, clear how to construct a general S-matrix for a general graph. The use of the word Anyon may thus be a bit of a generalisation from the usual definition but since we are generalising the theory, we thought we might as well call our models anyon models.

8 - In Section 3.1 there is an emphasis on picking good bases for the matrix representations of the graph braid generators. In order to characterize the image of the subgroup of the unitary group (set of quantum gates) generated by the graph braids, don't they all need to be in the same basis? Overall I found the discussion of good vs. bad choices of bases hard to follow.

Yes indeed, they do need to be in the same basis. However, in order to incorporate the no transmutation principle into the theory (see also our reply to the question 2 above) we need to be especially careful when choosing the fusion bases. The no transmutation principle is satisfied very easily in the standard 2D anyon theory, because in 2+1D one has much more freedom to deform the braids. For graphs braids, the existence of vertices causes several surprising and counterintuitive complications and the no transmutation principle is one of them. \newline When we consider the graph braid symbols (corresponding to graph braid generators) involving four particles, there are two choices of the basis of the fusion space i.e. what fusion trees these operators act on: } $( \ (a \times b) \times ( c \times d ) \ ) $ or $ ( \ ( \ (a \times b) \times c \ ) \times d \ ) $. {By $ (a \times b)$ we mean particles $a$ and $b$ are joined by a fusion tree and we order the particles so that $a$ is initially furthest from the junction point, so, under the graph braid action $a$ and $b$ will return, exchanged. We display both of these choices in the first column of Figure 5. The red inner oval is fixed as these are the particles that will be exchanged under the action of $\sigma_{3}$. However from here we still have two choices, given by the two choices of blue oval in subfigure (a). This is important as the choice of fusion tree dictates what topological charges are preserved under the action of an exchange, i.e even though $a \times b$ is exchanged, this does not affect the charge of $a \times b$. Similarly, if we choose these symbols to act on the fusion basis with $c\times d$, then we are fixing the action of the symbols to not change the topological charge of $c \times d$ and thus the matrix corresponding to this symbol must be diagonal in the $c \times d$ basis. However, we could have equally chosen the symbol to preserve the charge $ (a \times b ) \times c$, then it would be diagonal in this basis. The issue or novelty arises that these two choices are necessarily equivalent since there is a non trivial change of basis between these two choices.

9 - This is only an opinion but the notation e.g. $f= b\times c$ and calling $f$ a composite anyon is a little misleading in the anyon graph braids in Figure 3, since in the Q-hexagon equations there is a sum over $f$. Rather $f$ is an anyon in the fusion channel of $b$ and $c$. This is again confusing in Equation 59. Later however I am worried this is important, for example at the top of page 14, where a composite anyon is used as an index for one of the graph braid generators. In the planar setting it is not valid to use composite anyons as labels of fusion trees, so it would be good to see some more explanation of what is meant by this notation.

Thank you for this comment, you make a good point and this was really sloppiness on our part. We have amended the paper to say $b\times c$ instead of $f$ where possible and where we felt it would over-clutter our already cumbersome notation, we made a comment to say: "By $f$ we mean an anyon in the fusion channel of $b$ and $c$".
We have amended the equation (28), at the top of page 14 to say $a \times b$ instead of $f$.

10 - Is the label in the superscript of the graph braid $\sigma_{3}^{(1_{d},2_{c},1_{b},2_{a})}$ supposed to match the one in Figure 5(b)?

Figure 5 has now been updated. The labels of the graph braids $\sigma_{3}^{(1_{d},2_{c},1_{b},2_{a})}$ and $\sigma_{3}^{(2_{d},2_{c},1_{b},2_{a})}$ do match the initial configuration of the anyons before the exchange (the top plots in Figure 5a-d). The labels in the superscripts determine which anyon is shuttled to which branch of the $T$-junction with the rightmost anyon (labelled by '$d$' being shuttled as first and the leftmost anyon (labelled by '$a$') being shuttled as last.

11 - An investigation of what is meant by non-degeneracy of graph-braiding seems to be missing. Depending on the authors' aims it is not so important but it could be good to mention.

Yes indeed, this is a good insight. We also feel this is an interesting direction, however we felt this manuscript was already sufficiently long and technical. We are currently examining the implications of this pursuit and intend on producing a manuscript on this topic in the future.

12 - It is not obvious to me that a given solution to the graph hexagon equations uniquely specifies a wire network anyon model. Does one need to find solutions to graph ribbon equations as well?

In general I was a bit confused about the role of topological twists, independently of the underlying graph topology. This seems to be treated at the end of Section 4, where there is a definition given of topological twist and maybe partially in Appendix D but I was unclear about the big picture from the main body of the paper alone.

This is a very good question. In Appendix G we list solutions to the $N=3$ graph hexagons and the other polygon consistency equations for $N=4$ for Ising anyons. As one can see, the symbol $R^{\sigma\sigma}_1$ remains completely free after solving all the consistency equations. Thus, there is some $U(1)$ freedom left despite having satisfied all the possible graph braiding relations (we discuss this in Sections 3 and 6 and in Table 1). This freedom seems to be inconsequential for the construction of the topological quantum gates, since this $U(1)$ freedom always appears only as a global phase factor in the given quantum gate. However, this example shows that it is not always the case that solving the graph braiding consistency relations uniquely specifies the wire network anyon model (as there may still be some free parameters left). We anticipate that in actual physical models there will exist some physical mechanism that will superselect these free parameters.

We have analysed the topological twist and the related $D$-symbols as an example of the simplest graph braiding anyon theory. Note that the $D$-hexagons are completely universal. In other words, any graph that contains a closed loop will also imply the $D$-hexagon relations for its corresponding graph anyon model. However, if the graph does not contain any loops (i.e. when it is a tree graph), then the $D$-hexagons will not be relevant for its corresponding graph anyon model. The topological twist is also crucial for technical reasons in the proof of the equivalence between graph anyon models on a theta graph and anyon models on the plane. In particular in the step of the proof depicted in Figure 36 and Figure 37.

13 - If Appendix A is meant to be a stand-alone section containing the mathematical background of the graph-braid group it may best best if it did not make any reference to anyons, since the graph braid group is a purely mathematical notion. A presentation of all generators and relations for each type of graph under consideration should be clearly stated.

We changed the terminology from "anyon" to "particle" in that section. For each graph we study in the main body of the paper we typically consider only a sub-complete set of the relations appearing in the geometric presentation of its graph braid group. This is because only a sub-complete set of relations is normally sufficient to prove the respective result. Please note that this is a considerable simplification, since in order to state all the relations concretely, one needs to use discrete Morse theory algorithm from [L. Sabalka, D. Farley Alg \& Geom Topology 5:1075-1109, 2005]. Without the knowledge of the graph at hand, one can write down the relations only abstractly. The relations come from boundary words of the critical 2-cells of the Morse complex corresponding to the $n$-particle configuration space of the given graph. This is in stark contrast to the Artin braid group which possesses a simple list of relators. One of the fortunate simplifications in our work is the fact that that despite this complicated nature of graph braid groups, certain natural physically-motivated quotients of graph braid groups have presentations closely related to the presentations of Artin braid groups. Parts of these results are utilised in Section 7 of our manuscript to prove that the graph braiding on the theta graph is equivalent to braiding in 2D. We also utilise these relations in Section 8 in order to prove that on the stadium graph there exist two types of topologically inequivalent braids (see for instance the caption under Figure 23). As the Referee points out, in principle one needs to know all the relations in order to prove such a statement for the stadium graph. In this particular case, we have refrained from writing all the relators since they are extremely tedious to list -- the relations given by the Morse theory would involve long combinations of the critical $1$-cells of the Morse complex. We have checked that these relations indeed imply that there are two topologically inequivalent sets of braids by a tedious (but standard) calculation involving multiple Tietze transformations of the Morse presentation of the respective graph braid group.

14 - In Appendix D it may be best to avoid the use of the word "morphism" in the description of half-twists in graph-braided anyon models, since (apart from the underlying fusion category that exists by fiat) the authors have not shown that the graph-braided anyon model described a category.

Yes indeed, this is a very good point and we have changed it to say "map". Thank you for this.

15- In Appendix E there are some sentences that allude to demanding that the F-symbols should be gauge invariant but this is not physically justified and it will rule out many interesting theories. I think this was stated unintentionally based on the content of later appendices.

Thank you for this comment. We have amended that section to make it more clear what we meant. The original paragraph was a bit confusing -- we do not demand the $F$-symbols to be gauge-invariant, but rather consider the subgroup of the gauge group which leaves them invariant. We have clarified this point in the following equation:

$$ \frac{u^{af}{d} \hspace{2pt} u^{bc}}} { u^{ab{e} \hspace{2pt} u^{ec}\neq 0. $$}}=1 \quad \mathrm{whenever} \quad [F_{d}^{abc}]_{ef

---

## Round 1 · Referee Report · Anonymous (Referee 2) · 2024-2-12

Strengths

  1. Interesting & novel

  2. Clearly written

Weaknesses

  1. The main weakness is that the authors did not prove 'braid coherence' for their anyon wire network models.

Report

Report on the paper 'Extending the planar theory of anyons to quantum wire networks'

Due to the delay, I give a rather shorter report, than one could expect for a paper this length.
I will focus on the main text and appendix C.

The authors extend the theory of anyons (i.e., (unitary) (modular) tensor categories) to quantum wire networks. This is an important extension for several reasons. When restricting planar anyon models to quantum wire networks, it can happen that 'anyon' models that do not exhibit a consistent braiding on the plain, do have a consistent braiding on a wire network. Moreover, there are proposed models that do fall in this class.

In my opinion, this paper easily fullfils the requirements for publication in scipost. Below, I provide a list of questions and suggestions. The authors can use this list in order to improve the paper further.

Questions and suggestions.

Section 1.

The relation between the current paper and earlier work (such as [6,45]) is not entirely clear.
It could be beneficial if this is stated more clearly.

Section 2.

The authors mention that it could be interesting to consider non-commutative 'fusion' products, as the underlying graph provides a natural ordering. This is of course true. But, it would be good if the authors can mention any system (either experimental or theoretical), that hosts graph anyons that obey a non-commutative 'fusion' product.

A question concerning gauge factors and unitarity. In the multiplicity free case, the authors take u^ab_c in U(1), but the pentagon equations allow for arbitrary (non-zero) gauge coefficients. Are the authors restricting themselves to unitary models? If so, which result in the paper depend on this restriction?

A related question. Which results (if any) depend on the assumption that the fusion rules are multiplicity free?

When discussing Oceanu rigidity (or the number of independent solutions), one typically considers the number of solutions up to fusion automorphisms as well (as their number is finite, this does of course not change the rigidity property).

Section 3.

In section 3.1, the authors state that they did not prove that they actually considered all consistency relations in the case N=4. This is discussed in more detail in appendix C. It was, however, not entirely clear what additional work has to be done to prove braid coherence on wire networks.

In section 3.3, braiding on the H-graph is considered. The authors start with the anyons on one of the legs, and consider various consistency conditions. It was not immediately clear to me that if these consistency conditions are satisfied, it is implied that braiding two anyons on different legs (in the presence of other anyons as well), is als consistent. In other words, is it necessary to show 'coherence in the distribution of the anyons over the wire network'? Or is there a simple reason one does not have to worry about this?

Section 5.

I suggest the authors change 'lollipop' to 'tadpole' (throughout the paper), because tadpole is the established term for this graph/diagram.

Section 6.

It would be interesting to know if the authors think that some of the 'observations' in this section can be upgraded to more general results (and become conjectures), either in a reply or the in the paper.

Section 8.

The extend of the enhancement of 'computational power' on wire networks that occurs for certain models is not entirely clear to me. If a model is already universal, the only enhancement that can occur is that one needs fewer operations to approximate a gate for the same confidence level. However, if a model is not universal (i.e., provides a dense cover of some U(n)), it can happen that the different types of braids are inequivalent, and one can implement more gates. But, if one can still not obtain a sufficient set of gates (universal single qudit and an entangling two-qudit gate), has one really gained (much) in computational power?

A more naive question. In section 7, it is shown that the Theta graph yields effective planar anyon models. In section 8, the authors use the stadium graph, in which the Theta graph can be imbedded in two different ways. So, the braiding on the planar graph should also be like the planar case. So how can one get the inequivalent braidings the authors use in section 8 in the first place. Clearly, I am missing something here.
  • validity: high
  • significance: high
  • originality: high
  • clarity: high
  • formatting: excellent
  • grammar: excellent

Author:  Mia Conlon  on 2024-10-10  [id 4855]

(in reply to Report 2 on 2024-02-12)
Category:
remark
question

Report on the paper 'Extending the planar theory of anyons to quantum wire networks'

Due to the delay, I give a rather shorter report, than one could expect for a paper this length. I will focus on the main text and appendix C.

The authors extend the theory of anyons (i.e., (unitary) (modular) tensor categories) to quantum wire networks. This is an important extension for several reasons. When restricting planar anyon models to quantum wire networks, it can happen that 'anyon' models that do not exhibit a consistent braiding on the plain, do have a consistent braiding on a wire network. Moreover, there are proposed models that do fall in this class.

In my opinion, this paper easily fullfils the requirements for publication in scipost. Below, I provide a list of questions and suggestions. The authors can use this list in order to improve the paper further.

We thank Referee 2 for providing a very useful critical review of our work and providing a list of questions and suggestions to improve our manuscript for publication. We will address each of their points in order and we have updated our manuscript accordingly.

Section 1.

The relation between the current paper and earlier work (such as [6,45]) is not entirely clear. It could be beneficial if this is stated more clearly.

In Refs [6, 45], the authors use Morse theory to derive geometric generators and some of the relations for graph braid groups on a variety of graphs. The authors use these relations to show that certain properties of the graph braid group can be characterised solely by the topological connectivity of the network at hand. They also showed that multiple topologically inequivalent graph braids (exchanging the same pairs of particles) can exist on modular networks. In the submitted manuscript, we utilise the algebraic and combinatorical properties of graph braid groups from [6, 45] and consider the action of these groups on a Hilbert space given by the possible outcomes of fusing anyons, i.e the standard basis for the planar theory of anyons. In the previous works, the fusion rules and many-anyon structure of the Hilbert space is ignored (only the purely topological relations between the braids are studied). In essence our work is trying to adapt the planar category theoretic description of anyons to exchanges governed by graph braid groups. In addition we chose to focus on anyon models which are prominent candidates for topological quantum computing e.g. Ising, toric code models. We use the purely topological relations proved in Refs [6, 45] in order to derive the consistency polygon equations for the graph-braided anyon fusion theory. Examples of such purely topological relations are in equations (22), (30), (35), (56), (55). Each of these relations is re-proved in our submitted manuscript in a pictorial way in order to keep the presentation more accessible and self-contained.}

Section 2.

The authors mention that it could be interesting to consider non-commutative 'fusion' products, as the underlying graph provides a natural ordering. This is of course true. But, it would be good if the authors can mention any system (either experimental or theoretical), that hosts graph anyons that obey a non-commutative 'fusion' product.

A standard way this could arise would be excitations terminating on the boundary of non-Abelian toric code. One notable example where non commutative fusion has been studied is in \textbf{J. Math. Phys. 63, 042306 (2022)}. They study a 1+1D TFT with Haagreup symmetry, wherein the defect lines are labelled by the Haagreup fusion category. This category has a non-commutative fusion product. Physically, this could arise in our context in the case where the 1D edges are described by this TQFT. Experimentally, Non-commutative fusion could arise in quantum computing platforms that utilises gapped edges of a non-Abelian toric code. In our context this could arise where our the edges of our graphs arise as boundaries of a 2D topological order. }

A question concerning gauge factors and unitarity. In the multiplicity free case, the authors take $u^{ab}_c$ in U($1$), but the pentagon equations allow for arbitrary (non-zero) gauge coefficients. Are the authors restricting themselves to unitary models? If so, which result in the paper depend on this restriction?

Yes this is a good observation and one we should have mentioned more clearly. We always assumed unitary solutions of the pentagon equation. However it is worth pointing out we chose this restriction based on physical motivations, i.e. to be able to interpret the diagrams we utilise throughout the paper as space time evolutions. The graph braiding consistency equations we derive could also be studied with not necessarily unitary solutions of the pentagon equation. }

A related question. Which results (if any) depend on the assumption that the fusion rules are multiplicity free?

All of the example models we studied were multiplicity free and our graph braiding consistency equations in the paper are all written for the case of multiplicity free fusion. We made this choice for ease of exposition and the physical models that were of interest to us were all multiplicity free. The construction of the equations did not require the absence of fusion multiplicity and so we forsee that one could extend our results to include fusion multiplicity. }

When discussing Oceanu rigidity (or the number of independent solutions), one typically considers the number of solutions up to fusion automorphisms as well (as their number is finite, this does of course not change the rigidity property).

Thank you for this comment.

Section 3.

In section 3.1, the authors state that they did not prove that they actually considered all consistency relations in the case N=4. This is discussed in more detail in appendix C. It was, however, not entirely clear what additional work has to be done to prove braid coherence on wire networks.

In order to prove a coherence theorem for our graph anyon models we would need essentially two ingredients. Firstly, we need a minimal set of graph braid generators acting on the anyon Hilbert space that generate all other graph braid actions. This requires having a complete list of all generators and all relations amongst them. In Appendix C, we put forward an intuitive argument that we have such a collection on a trijunction (P,Q,R,A/B-symbols). In this appendix we essentially show that new generators ($N > 4$ particles) can be expressed in terms of the three and four particle generators, by sliding a fusion vertex through the braid. However we did not prove that this procedure can be done for any graph braid. This argument could probably be lifted to an N-valent single vertex graph by considering sub-trijunctions. However for general graphs, this is a very difficult problem due to the structure of graph braid groups. Secondly, we need a list of relations that impose the collection of non-degenerate constraints on these generators; our hexagon equations, charge conserving equations and greater particle number consistency equations. Crucially, here we would need to show our methods to generate these constraint equations is exhaustive. This we have been unable to do. We plan to address this in our future work.

In section 3.3, braiding on the H-graph is considered. The authors start with the anyons on one of the legs, and consider various consistency conditions. It was not immediately clear to me that if these consistency conditions are satisfied, it is implied that braiding two anyons on different legs (in the presence of other anyons as well), is als consistent. In other words, is it necessary to show 'coherence in the distribution of the anyons over the wire network'? Or is there a simple reason one does not have to worry about this?

This is a great insight and something we should have mentioned, thank you. By changing the distribution of anyons over the network, without exchanging, this is equivalent to changing the base point of the fundamental group, of the configuration space, which in this case is the initial setup of anyons for the graph braid group. Accordingly, changing the basepoint results in conjugating the braid operators by the same unitary matrix. So, there is always freedom in the description of the graph braid group (via the configuration space) to shuttle anyons around the network so long as no non-contractible loops (which correspond to exchanges) are introduced in the configuration space. However this would not change the resulting braiding exchange operators.}

Section 5.

I suggest the authors change 'lollipop' to 'tadpole' (throughout the paper), because tadpole is the established term for this graph/diagram.

Thank you for your suggestion. We would like to keep lollipop to avoid confusion between our abstract graph network and the tadpole diagram that appears in quantum field theory. Additionally, this terminology is quite standard in the literature we have seen for graph braid groups. }

Section 6.

It would be interesting to know if the authors think that some of the 'observations' in this section can be upgraded to more general results (and become conjectures), either in a reply or the in the paper.

One very general feature we observe is that when a graph contains closed loops, then the solutions of the consistency equations will always be discrete sets. Another quite general feature we observe is that if the graph does not contain loops, then there are often free $U(1)$ parameters in our solutions, however we not always observe this, there are two notable exceptions: Fibonacci and $PSU(2)_5$. Additionally, if there are solutions to the $N=3$ graph hexagon equations on any graph, then there are always solutions to the $N > 3$- consistency equations. Additionally, we conjecture that on the trijunction, the graph anyon model is coherent for $N > 4$ particles, i.e. when one constructs new graph consistency equations for $N > 4$ particles, these equations are already satisfied by the $N=4$ equations and additionally do not constrain the solutions any further.

Section 8.

The extend of the enhancement of 'computational power' on wire networks that occurs for certain models is not entirely clear to me. If a model is already universal, the only enhancement that can occur is that one needs fewer operations to approximate a gate for the same confidence level. However, if a model is not universal (i.e., provides a dense cover of some U(n)), it can happen that the different types of braids are inequivalent, and one can implement more gates. But, if one can still not obtain a sufficient set of gates (universal single qudit and an entangling two-qudit gate), has one really gained (much) in computational power?

There are essentially two aspects here. Firstly, needing fewer operations to approximate a gate can be very beneficial. Each operation in a quantum computer can induce errors, therefore needing to implement less operations for the same computational gate can make this platform a more reliable quantum computing architecture. Secondly, in our setup the increased (albeit still non-universal) computational complexity is found with \textbf{purely} topologically protected operations, i.e. braids. The most currently sought after anyon-based quantum computing platform involves Majorana zero modes, i.e. the Ising anyon theory. The set of gates produced by this anyon model is not universal, so many schemes have been proposed to include non topological operations, i.e. bring the Majorana modes into proximity with each other to allow a non topologically protected phase rotation, to complete the gate set. We view, maybe optimistically, our results in this regard as beneficial to computational platforms. }

A more naive question. In section 7, it is shown that the Theta graph yields effective planar anyon models. In section 8, the authors use the stadium graph, in which the Theta graph can be imbedded in two different ways. So, the braiding on the planar graph should also be like the planar case. So how can one get the inequivalent braidings the authors use in section 8 in the first place. Clearly, I am missing something here.

The core idea here is that each theta sub-graph of the stadium graph supports a solution of the graph hexagon equations, however these solutions need \textbf{not} be equivalent. In the context of the graph braid \textbf{group}, it was was proven in PRB. 102:201407 (Ref.[46]), that graph braids restricted to each distinct sub-graph need not be topologically equivalent. In our work we show this result lifts to the action of the graph braid generators on the fusion space. This necessitated introducing new quantities to our graph anyon models for example the topological twist, which we discuss in Appendix D.

---

## Editorial Decision

resubmitted